# GUARDIANS OF IMAGE QUALITY: BENCHMARKING DEFENSES AGAINST ADVERSARIAL ATTACKS ON IMAGE QUALITY METRICS

## ABSTRACT

Most modern image-quality-assessment (IQA) metrics are based on neural networks, which makes the adversarial robustness of these metrics a critical concern. This paper presents the first comprehensive study of IQA defense mechanisms in response to adversarial attacks on these metrics. We systematically evaluated 29 defense strategies — including adversarial purification, adversarial training, and certified robustness — and applied 14 adversarial attack algorithms in both adaptive and nonadaptive settings to compare these defenses on nine no-reference IQA metrics. Our analysis of the differences between defenses and their applicability to IQA metrics recognizes that a defense technique should preserve IQA scores and image quality. Our proposed benchmark aims to guide the development of IQA defense methods and can evaluate new methods; the latest results are at *link hidden for blind review*.

## 1 INTRODUCTION

Image-quality-assessment (IQA) metrics are essential for developing and evaluating image- and video-processing algorithms. Modern IQA metrics based on neural networks correlate highly with subjective-quality assessments. Neural networks, however, are vulnerable to input perturbations. Recent studies have explored IQA metric robustness (Antsiferova et al. (2024); Meftah et al. (2023); Zhang et al. (2024); Ghildyal & Liu (2023)), revealing that modern neural-networks-based metrics are susceptible to adversarial attacks. Adversarial attacks on IQA metrics are perturbations that increase the metric's score of an adversarial image without improving its real perceptual quality. Such attacks can manipulate image search results, as search engines (e.g., Microsoft's Bing) employ IQA metrics to rank outputs (Bing (2013)). Also, as IQA metrics serve in public benchmarks to evaluate image-/video-processing and compression algorithms, competitors can exploit metric's vulnerabilities to artificially inflate their algorithm's evaluated quality. As it was shown in Comparison (2021), the leaders by VMAF, a learning-based video quality metric by Netflix (vma), differ from the leaders by subjective comparison (Comparison (2021)). The fact that VMAF's vulnerability is being exploited is seen in Google's libaom video-compression codec, which has a "–tune=vmaf" option to increase VMAF scores for compressed videos by applying sharpening filters (Deng et al. (2020)). Several works showed that optimizing image restoration for modern IQA metrics can reduce actual image's quality (Ding et al. (2021)) or generate visual artifacts (Kashkarov et al. (2024)).

Researchers have proposed defense methods to increase neural-network robustness in different applications. Several benchmarks cover object-classification defenses (Croce et al. (2021); Dong et al. (2020)), but few defense methods are developed for IQA metrics, and no benchmarks were proposed for this task. Relative to defenses for classification models, which need only provide a correct class label, IQA metric defenses are more challenging. A successful IQA defense must meet two criteria: restore the original IQA scores and restore the perceptual quality of the original image, which may have suffered distortion after the attack. This paper introduces the first benchmark that systematically evaluates defenses against adversarial attacks on IQA metrics. Our contributions include a new method for measuring and comparing the efficiency of adversarial IQA metric defenses, a dataset of adversarial images for evaluating nonadaptive defenses, a series of comprehensive experiments, an in-depth analysis of the results, and an online leaderboard. Our method is the first to systematize defenses for IQA metrics. We analyzed 30 defense algorithms, including empirical and certified

054
055
056
057
058
059
060
061
062
063
064
065
066
067
068
069
070
071
072

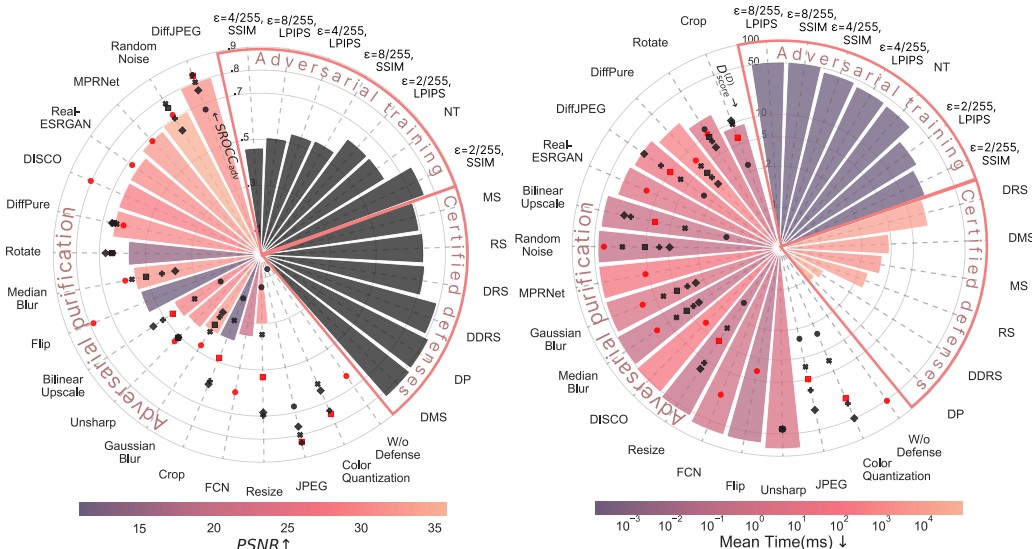

Figure 1: Adversarial defenses efficiency for IQA metrics in terms of $SROCC_{adv}$ (left) and $D_{score}^{(D)}$. Dots and bars are for adaptive and non-adaptive attacks, respectively. Each dot represent result for each preset of defense. Red dots represent a selected preset for adaptive case. We average the results by 9 IQA metrics and 14 attacks.

ones, and evaluated their efficiency against 14 adversarial attacks. We analyzed scenarios of both adaptive and nonadaptive attacks, depending on the awareness of an attack of the defense method. The benchmark is available online [1] along with the dataset of adversarial images that can be used for adversarial training; code for our proposed method, IQA models, adversarial attacks, and defense methods is in the GitHub repository [2]. The benchmark gives developers and researchers a unified framework for measuring and comparing defense efficiency and we encourage submissions of new methods.

## 2 RELATED WORK

Existing comparisons of defense methods efficiency mostly handle object classification (Croce et al. (2021), Dong et al. (2020)). All leaderboards of defense methods on paperswithcode.com are based on image classification datasets: ImageNet, CIFAR, MNIST, etc. However, no benchmark addresses defense methods for IQA metrics attacks. The closest benchmark is Antsiferova et al. (2024), which compares the robustness of existing IQA metrics to adversarial attacks. However, aforementioned benchmark does not analyse defense methods. It can serve to evaluate a new IQA metric, but considering the vast variability of IQA tasks (e.g., measuring the quality of user- or AI-generated content, artificial distortions caused by image-processing algorithms, etc.), a single universal and robust IQA metric can not be created. Instead, we propose a comparison of defense methods' efficiency for IQA task to help the researchers improve the robustness of their existing metrics.

IQA metrics can be categorized as full-reference (FR) or no-reference (NR) depending on the availability of the reference image. This paper evaluates adversarial defenses on NR metrics, which is a more challenging task, because NR metrics show lower robustness than FR ones (Antsiferova et al. (2024)). Moreover, the only three existing robust metrics are FR (VMAF NEG Li (2020), R-LPIPS Ghazanfari et al. (2023), LipSim Ghazanfari et al. (2024)), but, to our knowledge, no robust NR metrics have been proposed so far, which makes it essential to find suitable defenses for them. Adversarial attacks fall into two categories depending on the attacker's knowledge of the model: "white box" or "black box". White-box attacks employ gradients of the attacked models; how-

---
[1]hidden for blind review
[2]hidden for blind review

ever, in some situations, gradients are unavailable, and black-box attacks remain applicable. Several white-box adversarial attacks (Shumitskaya et al. (2024); Korhonen & You (2022); Zhang et al. (2022b); Wang & Simoncelli (2008); Shumitskaya et al. (2023)) and at least two black-box attacks (Ran et al. (2024); Yang et al. (2024)) have emerged for fooling IQA metrics. Defense methods for neural networks come in certified and empirical types. Certified methods provide deterministic or probabilistic robustness guarantees for particular perturbations, datasets, or model architectures. However, these methods are usually computationally complex and reduce the model's general accuracy. One of the most well-known certified methods is randomized smoothing (Cohen et al. (2019)). Later variations appeared in Salman et al. (2020); Chen et al. (2022b), and included a denoiser to improve the defended model's performance. Empirical methods lack robustness guarantees but require fewer computational resources. A widely used empirical defense method is adversarial training (Wong et al. (2020); Singh et al. (2023)), which updates the model weights based on generated adversarial examples during training. Vanilla adversarial training, however, may decrease model performance. Adversarial purification (Nie et al. (2022)) is an empirical method that removes adversarial perturbations by processing input data. Although adversarial purification is model-agnostic and computationally efficient, it may fail to eliminate advanced adversarial perturbations and can sometimes degrade image quality. Examples include compression, spatial transformation, blurring, and denoising.

## 3 METHODOLOGY

### 3.1 PROBLEM DEFINITION

**Adversarial attacks**. This work evaluates adversarial defenses for NR IQA because NR metrics are more susceptible than FR metrics to adversarial attacks. Therefore, an attacked model, represented by an NR IQA metric, takes a single image as input and estimates image quality. Formally, the NR IQA metric is the mapping $f_\omega : X \to \mathbb{R}$, parameterized by the vector of weights $\omega$. Here, $X \in [0,1]^{3 \times H \times W}$ is the set of input images. An adversarial attack $A : X \to X$ is the perturbation of the input image defined as

$$A(x) = \underset{x' : \rho(x', x) \leq \varepsilon}{\arg \max} \; L(f_\omega(x')), \tag{1}$$

where $L$ is a loss function that represents the model's outputs for perturbed images and $\rho(\cdot, \cdot)$ is the distance function defined on $X \times X$. We increase IQA scores during the attack to reflect real-life applications. For IQA metric attacks, we define $L(f_\omega(x')) = \frac{f_\omega(x')}{\text{diam}(f_\omega)}$, where $\text{diam}(f_\omega) = \sup_{x,z \in X} \{|f_\omega(x) - f_\omega(z)|\}$ represents the range of IQA metric values. The Appendix A.3 includes the ranges of IQA metrics in our work.

**Adversarial defenses**. In our work, we consider three types of adversarial defenses.

*Adversarial purification* is an algorithm $P : X \to X$ that aims to transform the input image according to the following optimization problem:

$$\min \; |f_\omega(P(x')) - f_\omega(x)| + \lambda \rho(P(x'), x), \tag{2}$$

where $x'$ is the adversarial image, and $\lambda$ is the regularization term.

*Adversarial training* is formulated as the following min-max problem:

$$\min_\omega \mathbb{E}_{(x,y) \sim \mathcal{D}} \left[ \max_{\|\delta\|_p \leq \varepsilon} \mathcal{L}(f_\omega(x + \delta), y) \right], \tag{3}$$

where $\mathcal{D}$ is the distribution of training data, $\mathcal{L}$ is a training loss function, and $\varepsilon$ is the allowable attack magnitude. In practice, adversarial training uses an adversarial attack rather than internal maximization.

*Certified methods* used in our paper are based on randomized smoothing (Cohen et al. (2019)), denoised randomized smoothing (Salman et al. (2020)), diffusion-based randomized smoothing (Carlini et al. (2022); Chen et al. (2022b)) and median smoothing (Chiang et al. (2020)). The idea behind randomized smoothing is replacement of the original IQA metric with a smoothed version:

$$g(x) = \underset{\epsilon \sim \mathcal{N}(0, \sigma^2)}{\mathbb{E}} f_\omega(x + \epsilon) \tag{4}$$

where $\epsilon$ is a centered Gaussian random variable.

Table 1: Adversarial attacks in our benchmark. We adjusted parameters to align attack strengths and launched each attack as procedures.

| | Adversarial attack | Param. | Short description |
|---|---|---|---|
| White-box | MADC (Wang & Simoncelli (2008)) | lr | Grad. project. onto MSE |
| | I-FGSM (Kurakin et al. (2018)) | lr | Grad. descent to increase IQA metric |
| | Korhonen et al. (Korhonen & You (2022)) | lr | Sobel-filter-masked gradient descent |
| | Zhang et al. (Zhang et al. (2022b)) | lr | Grad. descent with saving DISTS |
| | SSAH (Luo et al. (2022)) | lr | Grad. descent with high-freq. min. |
| | FACPA (Shumitskaya et al. (2023)) | amplit. | Perturb. generated using U-Net |
| | UAP (Shumitskaya et al. (2024)) | amplit. | Universal perturb. via grad. descent |
| | cAdv (Bhattad et al. (2019)) | lr | Grad. descent with recolorization |
| Black-box | NES (Ilyas et al. (2018)) | $\epsilon$ | Grad. descent with approx. gradient |
| | One Pixel (Su et al. (2019)) | #pixels | Perturbs pixels with diff. evolution |
| | Parsimonious (Moon et al. (2019)) | $\epsilon$ | Perturbs using discrete optimization |
| | Square (Andriushchenko et al. (2020)) | $\epsilon$ | Square-like perturb. via rand. search |
| | Patch-RS (Croce et al. (2022)) | $\epsilon$ | Finds adv. patch via random search |

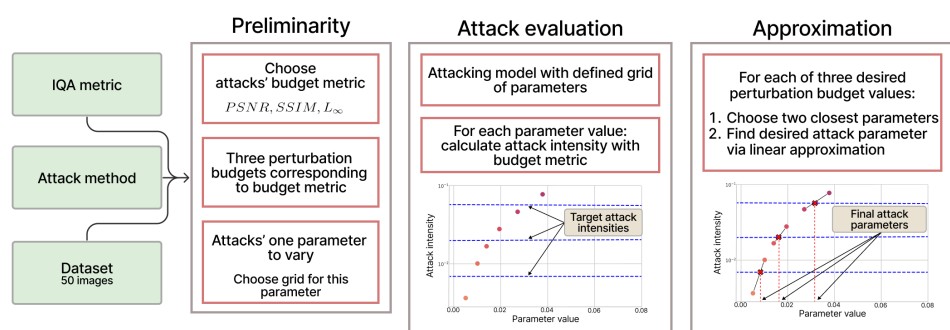

Figure 2: Procedure for selecting adversarial attack parameters.

## 3.2 ADVERSARIAL ATTACKS

This work considers two adversarial attack scenarios: nonadaptive and adaptive. In the first case, we generate a set of adversarial images. In the second, we integrate differentiable defenses into the attacked IQA metric, allowing the adaptive attacks to access the IQA metric and defense-method gradients, thereby simulating a greater challenge. We selected several white- and black-box attacks from the IQA robustness benchmark (Antsiferova et al. (2024)). Table 1 describes these attacks. We executed each method with three distinct hyperparameter sets corresponding to "weak", "medium", and "strong" attacks by perturbation budget to account for different adversarial attack strengths. Figure 2 illustrates the parameter-selection procedure, ensuring all attack hyperparameters yielded perturbations with an equal set of $l_\infty$-bounds. We chose a subset of 50 images used for attack alignment via clustering the KonIQA dataset by spatial complexity (SI), colorfulness, and ground-truth quality labels. All chosen hyperparameter sets are listed in the Appendix A.3.

## 3.3 ADVERSARIAL DEFENSES

**Adversarial purification**. The top part of the Table 2 describes the selected adversarial purification techniques. We used five parameter sets to vary the defense strength. Parameter sets differ by defense strength, e.g., scaling ratio, blurring kernel size, and number of diffusion steps. The Appendix A.5 provides a list of used defense parameters.

**Adversarial training**. IQA presents additional challenges in applying adversarial training since adversarial examples generated in training have lower subjective quality. Manually labeling such images with new scores is impractical, and using ground-truth labels from pre-attack images is inaccurate. To overcome these limitations, our experiments employed six training configurations: two

Table 2: Compared adversarial defense methods in our benchmark.

| | Defense method | Param. | Short description |
|---|---|---|---|
| Adversarial purification | Gaussian blur | kernel size | Smooth with a Gaussian filter |
| | Median blur | kernel size | Smooth with a median filter |
| | JPEG (Guo et al. (2018)) | q | JPEG compression algorithm |
| | Color quantization (Xu et al. (2018)) | npp | Reduce the number of colors |
| | DiffJPEG (Reich et al. (2024)) | q | Differentiable JPEG |
| | Unsharp masking | kernel size | Unsharp mask |
| | FCN (Gushchin et al. (2024)) | — | Neural filter to counter color attack |
| | Flip | — | Mirror the image |
| | Bilinear Upscale | scale | Resize and upscale to original size |
| | Resize (Guo et al. (2018)) | scale | Change the image size |
| | Random Rotate | angle lim. | Image rotation |
| | Random Crop (Guo et al. (2018)) | size | Crop the image |
| | Random noise | — | Add random noise |
| | MPRNet (Zamir et al. (2021)) | — | 3-stage CNN for denoising |
| | Real-ESRGAN (Wang et al. (2021)) | — | GAN-based super-res. denoising |
| | DISCO (Ho & Vasconcelos (2022)) | — | Enc.+loc. implicit module denoising |
| | DiffPure (Nie et al. (2022)) | t | Diffusion denoising |
| | Classic adversarial training | $\epsilon$, FR metric | Model fine-tuning on adv. img. |
| | Gradient Norm optimization Liu et al. (2024) | — | Perform gradient normalisation during training |
| Certified | Random. Smoothing (RS) (Cohen et al. (2019)) | | Noisy samp. →clf.→voting |
| | Denoised RS (DRS) (Salman et al. (2020)) | | Noisy samp.→denoiser→ clf.→voting |
| | Diffusion DRS (DDRS) (Carlini et al. (2022)) | | Noisy samp.→1-step diffus.→clf.→voting |
| | DensePure (DP) (Chen et al. (2022b)) | | Noisy samp.→N-step diffus.→clf.→voting |
| | Median Smoothing (MS) (Chiang et al. (2020)) | | Noisy samp.→reg.→median |
| | Denoised MS (DMS) (Chiang et al. (2020)) | | Noisy samp.→denoiser→reg.→ median |

score-penalizing strategies (using SSIM or LPIPS) with three attack magnitudes $\varepsilon = \{2, 4, 8\}/255$. To create the adversarial images during training, we applied an APGD attack (Croce & Hein (2020)).

**Certified defenses**. This work employed smoothing-based certified defenses that are easy to implement with any NR image-quality metric and avoid restricting the model architecture. Only one study (Chiang et al., 2020) investigated smoothing for regression. Additionally, we added classification-based defenses which we adapted for regression by converting the regression model into a multiclass classification model. All these defense methods generate noisy variations of the input images, which then pass through the model. Before passing them through the model, some of these methods apply denoising to boost accuracy. Table 2 provides further details. For each certified defense method, we generated 1000 noisy images as an input.

To discretize a regression-based quality metric for classification-based methods, we divided the metric range into N segments, each corresponding to a specific class. According to our experiments, 10 segments showed optimal correlation with subjective scores. We also added classes for metric values below or above the calculated range, ensuring every value falls into a class. Doing so yielded an $(N + 2)$-class metric classifier. Note that these classes are ordered, with higher class values indicating better quality. Thus, we can measure classifier-metric quality the same way we measure regression-metric quality, using relative gain and correlation with subjective scores. Given an input image, the results of the classification-based certified method are the quality class and the certified radius $R$. The method guarantees that the class remains unchanged for the input image in an $l_2$ ball of radius $R$. The results of a regression-based certified method are the metric score and the certified delta. The method guarantees that within an $l_2$ ball of radius $\epsilon$, the metric score changes by no more than delta. To make this value comparable across metrics, we define the certified relative delta by dividing the certified delta by $\text{diam}(f_\omega)$.

## 3.4 EXPERIMENTAL SETUP

**Datasets**. Our study used four datasets to evaluate adversarial defenses. KonIQA-10k (Hosu et al. (2020)) and KADID-10k (Lin et al. (2019)) contain various natural images with multiple distortions. NIPS 2017: Adversarial Learning Development Set (2017, Competition Page) is designed for evaluating adversarial attacks against image classifiers.AGIQA-3K dataset (Li et al. (2024)) contains AI-generated images for different quality levels. To balance computational efficiency and dataset diversity, we randomly sampled 1,000 out of 10,073 KonIQA-10k images and 1,000 out of 10,125

KADID-10k images. We included each distortion type and strength and sampled 8 out of 81 original images from KADID-10k, resulting in 1000 distorted images. We analyzed different sample sizes while selecting a subset of images to ensure that our subset represents the whole dataset. Figure 5 in the Appendix A.2 shows that the distribution for the 1,000-image sample is nearly identical to the distribution for the full dataset. Due to high computational complexity, we used a smaller set of 50 images from each dataset for black-box attacks and, for the same reason, a smaller set of 10 images from each dataset when evaluating certified defenses.

**IQA metrics**. We chose nine NR-IQA metrics by the results of the IQA adversarial robustness benchmark (Antsiferova et al. (2024)): META-IQA (Zhu et al. (2020)), MANIQA (Yang et al. (2022)), CLIP-IQA+ (Wang et al. (2023)), TOPIQ (Chen et al. (2024)), KonCept (Hosu et al. (2020)), SPAQ (Fang et al. (2020)), PAQ2PIQ (Ying et al. (2020)), Linearity (Li et al. (2020)), and FPR (Chen et al. (2022a)). These metrics employ different convolutional and transformer-based architectures with a wide robustness $R_{score}$ range. The Appendix A.5 provides a more detailed description. Because of adversarial training's computational complexity, we selected only two CNN-based NR-IQA metrics — Linearity and KonCept — because of their high correlation with subjective quality scores.

### 3.5 EVALUATION METRICS

**Robustness scores**. $R_{score}$ and $R_{score}^{(D)}$ (R robustness Zhang et al. (2022a)) aims to assess model robustness by measuring relative score changes before and after attacks. $R_{score}$ takes into consideration the maximum allowable quality-prediction change:

$$R_{score} = \frac{1}{N} \sum_{i=1}^{N} \log \left( \frac{\max\{\beta_1 - f_\omega(x_i), f_\omega(x_i) - \beta_2\}}{|f_\omega(x_i) - f_\omega(P(x_i'))|} \right), \tag{5}$$

where $N$ is the number of images, $x_i$ is the $i_{th}$ source image, $x_i'$ is the attacked version of $x_i$. $f_\omega(*)$ is the IQA model, and $\beta_1$ and $\beta_2$ are the maximum and minimum mean opinions scores (MOS) in the dataset, which are ground-truth quality labels. In addition, we propose a variation of this metric called $R_{score}^{(D)}$, which differs only in applying purification $P(*)$ to $x_i$ and $x_i'$. These metrics can be calculated only for datasets with subjective scores. A larger value means better robustness.

$D_{score}$ and $D_{score}^{(D)}$ measure adversarial purification's ability to reduce the discrepancy between the IQA scores of the original and attacked images after applying a defense:

$$D_{score} = \frac{100}{n} \sum_{i=1}^{n} \frac{|f_\omega(P(x_i')) - f_\omega(x_i)|}{\text{diam}(f_\omega)}; \; D_{score}^{(D)} = \frac{100}{n} \sum_{i=1}^{n} \frac{|f_\omega(P(x_i')) - f_\omega(P(x_i))|}{\text{diam}(f_\omega)}, \tag{6}$$

where scores denoted with the superscript $^D$ are for purified source images, $P$ represents the purification method. Lower scores correspond to better defense performance. We propose these metrics to quantify how much an IQA metric's predicted values for an adversarial image can be restored to their originals after the defense is applied. The fundamental premise is that a robust defense should minimize the disparity between the IQA scores of the defended and original images.

For certified defense methods, we additionally measured the *certified radius* ($Cert.R \uparrow$), which indicates how much the input image can undergo alteration without changing the class prediction; the *percentage of abstentions* ($Abst. \downarrow$), reported by classification-based methods when their predictions are highly uncertain; and *certified relative delta* ($Cert.RD \uparrow$), which is the certified delta, produced by the defense method, divided by $\text{diam}(f_\omega)$. This parameter characterizes how much a metric score can change in a fixed $l_2$ ball of norm $\epsilon$ around a given image $x$.

**Quality scores**. We use $PSNR$ and $SSIM$ (Wang et al. (2004) ) to measure the perceptual similarity between purified images and their original images, reflecting the preservation of visual quality post-defense. The underlying principle is that the defense mechanism should restore the IQA score and preserve the image's perceptual quality.

**Performance scores**. We use $SROCC$ and $PLCC$ to assess an IQA metric's performance in the presence of adversarial defense. We measured the correlation between ground-truth image-quality scores $y$ and the IQA metric predictions. $SROCC_{clear}$ is a coefficient measured for purified non-adversarial images, which represents a scenario with a detection method that identifies adversarial

Table 3: Comparison of defenses. Evaluated metrics are averaged across all images and attacks for all quality metrics for nonadaptive/adaptive use cases on all datasets.

| | Time (ms) ↓ | $SROCC_{clear}$ ↑ | $SROCC_{adv}$ ↑ | $D_{score}^{(D)}$ ↓ | $R_{score}$ ↑ | $PSNR_{adv}$ ↑ | $SSIM_{adv}$ ↑ | $MSE$ ↓, $\times 10^{-3}$ | $L_{inf}$ ↓ |
|---|---|---|---|---|---|---|---|---|---|
| W/o Defense | — | 0.611/0.511 | 0.447/0.413 | 56.39/66.68 | 0.76/0.56 | **43.89/44.61** | **0.94/0.94** | **2.51**/1.86 | **0.08/0.09** |
| Flip | **0.05** | 0.593/0.587 | 0.555/0.420 | 7.91/67.41 | **1.17**/0.45 | 10.76/10.76 | 0.28/0.29 | 110.47/109.80 | 0.95/0.95 |
| Color Quantization | 0.07 | 0.587/— | 0.532/— | 27.38/— | 0.83/— | 32.54/— | 0.86/— | 2.84/— | 0.11/— |
| Median Blur | 0.11 | 0.551/0.531 | 0.431/0.424 | 15.14/49.95 | 0.92/0.50 | 31.38/31.80 | 0.86/0.87 | 4.48/3.17 | 0.51/0.51 |
| Bilinear Upscale | 0.15 | 0.569/0.479 | 0.452/0.355 | 18.13/40.93 | 0.86/0.58 | 32.82/28.68 | 0.91/0.83 | 3.50/4.23 | 0.35/0.48 |
| Crop | 0.16 | 0.587/0.431 | 0.508/0.385 | 11.68/**6.49** | 0.92/0.78 | 11.53/11.00 | 0.33/0.37 | 89.94/105.34 | 0.94/0.93 |
| Resize | 0.19 | 0.597/0.511 | 0.549/0.353 | 10.56/54.31 | 1.02/0.42 | 32.11/29.38 | 0.90/0.85 | 3.83/3.90 | 0.37/0.45 |
| FCN | 0.52 | 0.571/0.562 | 0.478/0.310 | 23.89/64.32 | 0.80/0.41 | 20.89/20.78 | 0.78/0.77 | 13.24/13.35 | 0.54/0.55 |
| Unsharp | 0.78 | 0.611/0.595 | 0.427/0.370 | 43.22/80.24 | 0.52/0.32 | 30.34/29.77 | 0.87/0.86 | 3.81/3.03 | 0.33/0.35 |
| Gaussian Blur | 0.99 | 0.552/0.522 | 0.423/0.376 | 15.75/45.67 | 0.84/0.53 | 32.22/32.30 | 0.90/0.90 | 3.83/2.72 | 0.34/0.35 |
| Rotate | 2.14 | 0.560/0.585 | 0.501/0.469 | 6.64/16.24 | 1.09/0.89 | 11.56/14.65 | 0.31/0.42 | 96.44/54.03 | 0.97/0.96 |
| Real-ESRGAN | 5.89 | 0.552/0.501 | 0.503/0.436 | 9.47/30.13 | 0.66/0.58 | 30.32/30.47 | 0.89/0.88 | 3.97/2.98 | 0.43/0.44 |
| DiffJPEG | 8.11 | **0.625**/0.610 | **0.608**/0.549 | 12.94/29.81 | 1.07/0.71 | 34.33/31.33 | 0.91/0.87 | 3.04/2.61 | 0.26/0.33 |
| Random Noise | 8.29 | 0.556/0.594 | 0.539/0.508 | 10.14/44.84 | 0.87/0.59 | 25.42/35.87 | 0.54/0.90 | 4.78/**1.79** | 0.30/0.13 |
| MPRNet | 65.79 | 0.565/0.565 | 0.535/0.488 | 12.14/45.00 | 0.97/0.53 | 32.21/32.32 | 0.88/0.89 | 4.23/2.91 | 0.37/0.36 |
| DISCO | 139.60 | 0.585/0.562 | 0.581/0.476 | 3.51/47.91 | 1.14/0.50 | 29.12/29.08 | 0.86/0.86 | 4.34/3.31 | 0.43/0.43 |
| JPEG | 387.34 | 0.622/— | 0.605/— | 13.07/— | 1.07/— | 34.25/— | 0.90/— | 3.03/— | 0.26/— |
| DiffPure | 4291.42 | 0.496/0.487 | 0.485/0.470 | **2.01**/22.96 | 0.79/0.75 | 27.59/30.11 | 0.79/0.86 | 5.34/3.44 | 0.48/0.43 |

attacks before passing them to the purification defense; and $SROCC_{attacked}$ is measured for purified adversarial images:

$$SROCC_{clear} = SROCC(\vec{y}, f_\omega(P(\vec{x}))); \ SROCC_{adv} = SROCC(\vec{y}, f_\omega(P(\vec{x}'))). \quad (7)$$

### 3.6 Implementation Details

We used a sophisticated end-to-end automated training and evaluation pipeline with a unified interface using GitLab CI/CD tools to ensure all our results are reproducible. Calculations were made with Slurm Workload Manager with $120 \times$ NVIDIA Tesla A100 80 Gb GPU, Intel Xeon Processor (Ice Lake) 32-Core Processor @ 2.60 GHz. All calculations took a total of about 25,000 GPU hours.

We used original open-source implementations with default parameters for all adversarial attacks, defenses, and IQA metrics when available. For each attack and defense, we varied one main parameter — commonly associated with the attack strength — while keeping the remaining parameters consistent with their original implementations (see 2). However, because the exact relationship between the varied parameters and attack strength is sometimes poorly defined, the impact on attack intensity may be somewhat unpredictable. The Appendix A.4, A.5 provides a list of parameters for attacks and defenses, links to the original repositories, and a list of the applied patches necessary to enable gradients in some metrics.

## 4 Results

This section presents defense efficiency against adaptive and nonadaptive attack scenarios. In all tables and figures, for nonadaptive cases, we report the results of defenses with a hyperparameter set that provides the best $SROCC_{adv}$ on the KonIQA dataset.

**Overall performance-robustness trade-off**. Our comparison considers three aspects of defense method performance: improving the robustness of the defended model ($D_{score}$, $R_{score}^{(D)}$), and preserving correlation with human perception ($SROCC$, $PLCC$) along with the quality of the image ($PSNR$, $SSIM$). By varying defense parameters, its performance can be balanced with other characteristics. Strong defenses may significantly alter input images and erase adversarial perturbation, but they often degrade image quality and correlation with human perception. Figure 3 demonstrates how adversarial purification parameters influence the robustness-performance ratio. The strong defenses in the lower left corner almost completely returned IQA metrics scores to the original ones before the attacks but significantly lowered correlations with subjective quality, making them unsuitable for real-life application scenarios. The red line shows the Pareto-optimal front of tested methods; it includes strong JPEG compression, weak DiffPure and Gaussian blur, and DISCO. Flipping the image shows a high correlation, but only for nonadaptive attacks. Table 3 shows overall results for adversarial purification defenses, and Table 4 — for adversarial training and certified methods. In all tables and figures, we show only SROCC; PLCC results are similar and are provided in the Appendix A.8. DiffJPEG leads in several categories, demonstrating the best $SROCC_{adv}$, $D_{score}$

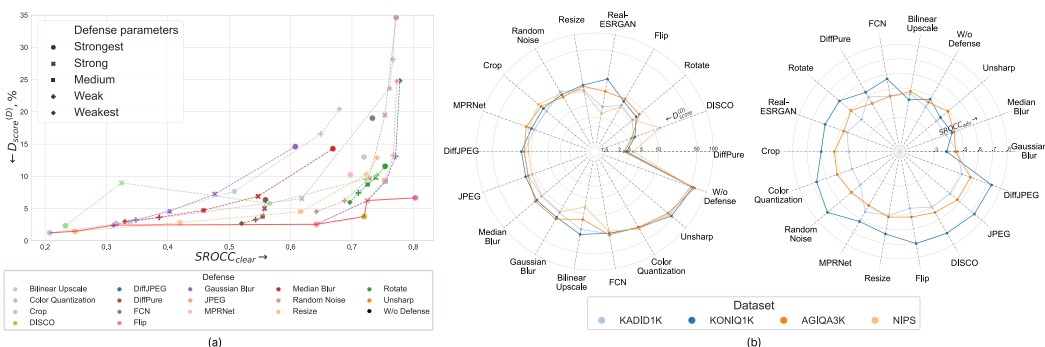

(a)  (b)

Figure 3: Scatter plot (a) depicts results for different presets of parameters of purification defenses in nonadaptive cases, where red line indicated Pareto-optimal defenses. (b) illustrates the results for different test datasets in terms of $D_{score}^{(D)}$ (left) and $SROCC_{adv}$ (right).

Table 4: Comparison of adversarial training (left) and certified defenses (right). **C** is classification-based methods, **R** — regression-based. For AT methods, APGD is an attack used during fine-tuning, 2/4/8 is an attack perturbation budget. LPIPS/SSIM is an FR metric used for adjusting MOS values.

| | Adaptive attacks, 1000 images KonIQ, KONCEPT+LINEARITY | | | | Nonadaptive attacks, 10 images from KonIQ, 9 IQA metrics | | | | | |
|---|---|---|---|---|---|---|---|---|---|---|
| AT Defense | $SROCC_{clear}$ ↑ | $SROCC_{adv}$ ↑ | $D_{score}^{(D)}$↓ | $R_{score}$ ↑ | Cert Defense | Time(ms)↓ | $SROCC_{clear}$ ↑ | $SROCC_{adv}$ ↑ | $D_{score}^{(D)}$↓ | $R_{score}$ ↑ | Cert.R ↑ / Cert.RD ↓ |
| APGD-LPIPS-2 | 0.840±0.00 | 0.651±0.14 | 20.70±13.92 | 1.10±0.82 | RS (**C**) | 11080 | 0.747 | 0.706 | 2.70 | 5.61 | **0.183** / ∞ |
| APGD-LPIPS-4 | 0.866±0.00 | 0.576±0.25 | 36.59±29.97 | 0.77±0.80 | DRS (**C**) | 15320 | **0.882** | 0.712 | 16.57 | 2.01 | 0.175 / ∞ |
| APGD-LPIPS-8 | 0.867±0.00 | 0.547±0.22 | 45.61±37.10 | 0.79±0.87 | DDRS (**C**) | 39800 | 0.819 | 0.792 | 1.20 | 6.21 | 0.174 / ∞ |
| APGD-SSIM-2 | 0.830±0.00 | 0.763±0.05 | 17.11±11.41 | 1.23±0.94 | DP (**C**) | 82130 | 0.823 | 0.815 | 1.09 | 6.20 | 0.162 / ∞ |
| APGD-SSIM-4 | 0.852±0.00 | 0.505±0.27 | 39.53±33.33 | 0.80±0.89 | MS (**R**) | **2830** | 0.753 | 0.694 | 3.80 | 1.92 | 0 / 1.707 |
| APGD-SSIM-8 | **0.873±0.00** | 0.582±0.20 | 45.38±37.21 | 0.64±0.81 | DMS (**R**) | 5970 | 0.875 | **0.822** | 4.70 | 1.89 | 0 / **1.440** |
| NT Liu et al. (2024) | 0.815±0.00 | 0.649±0.14 | 35.42±23.97 | 0.805±0.82 | | | | | | | |

and $R_{score}$, while being in the top-3 methods according to $SROCC_{clear}$ and $PSNR$. IQA metrics with adversarial training showed lower robustness than ones with purification defenses. For adversarial training, smaller $\epsilon$ restrictions for generating attacks while training led to better performance, but not higher $SROCC_{clear}$. The best AT method (APGD-SSIM-2) ranks the top-3 methods according to $R_{score}$, but is worse for other performance metrics. In our experiments, certified methods performed well even in empirical settings when perturbations with an $l_2$-norm significantly larger than the certified radius (for classification-based) or $\epsilon$ (for regression-based). The best method in terms of empirical robustness $D_{score}^{(D)}$ is DensePure, largely because diffusion-based denoising is highly effective at removing Gaussian noise with a known $\sigma$. Additionally, converting regression into a classification task by quantizing metric values helps achieve zero gain on more than half of the data. **Inference computational complexity**. Tables 3 and 4 show time measurements for compared methods. Certified methods show the best efficiency in adversarial defense but have the highest computational complexity. The fastest certified defense is median smoothing, which requires about 26 seconds for one image, while the slowest — DensePure — runs for 116 seconds. Conversely, adversarial training comes without additional computational cost during inference, making it the fastest defense method in our benchmark. The computational complexity of purification-based defenses largely depends on the specifics of the particular method. Simple image preprocessing methods (blurring, rotation, random noise) add almost no additional computational cost. The most computationally intensive method is Diffpure. Due to the diffusion-based model with multiple denoising steps, it is slower than other purification defenses by 1-3 orders of magnitude. The second slowest purification-based defense is JPEG, the only method that requires running on the CPU. However, its differentiable approximation, DiffJPEG, demonstrates significantly better speed as it runs on GPU.

**Defenses against adaptive and non-adaptive attacks**. Figure 1 compares defense performance against non-adaptive and adaptive attacks. Note that adversarial training was measured only for adaptive attacks, while certified defenses were only for nonadaptive scenarios: it is possible to construct an attack on a certifiably smoothed model, but such a procedure requires extremely high computations with a potentially low attack success. By design, adaptive attacks are significantly more successful than non-adaptive ones, and so $D_{score}^{(D)}$ robustness bars are higher than markers. At the same time, $SROCC_{adv}$ of defended IQA metrics against adaptive attacks is lower, which could be related to the more unpredictable behavior of adaptive attacks. Simple spatial transformations

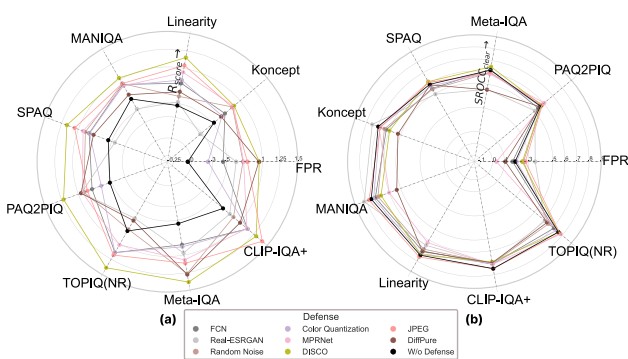

Figure 4: $R$ $score$ (a) and $SROCC_{clear}$ (b) on different IQA metrics of some purification defenses.

(Flip, Resize) and frequency filtering (Gaussian Blur, Median Blur) are effective in the nonadaptive case but insufficient for adaptive attacks. More complex methods that incorporate randomness, such as Random Rotate, Random Crop, and DiffPure, perform better for adaptive cases as it is more complex for an attack to account for random transformations.

**Defenses against weak and strong attacks** The comparison results for different variations of attack parameters, as presented in the Appendix A.8 (Table 14), show that increasing attack strength generally leads to decreased defense success. Although this decrease is not significant for most defenses in the case of nonadaptive attacks, it is more prominent for Random Noise, DISCO, and FCN in the adaptive scenario.

**Different types of defenses** In the non-adaptive scenario, robustness scores' most effective defense methods incorporated compression and spatial transformations. Compression-, denoising-, and filtering-based (like FCN, unsharp masking) provide the best visual quality preservation. Methods based on random transforms, compression, and denoising were the most resistant against adaptive attacks. Therefore, spatial transforms and advanced defenses designed for adversarial noise removal (DISCO, DiffPure) are the least robust against adaptive attacks: adversarial perturbations can be easily adapted to these methods. The table with the results for defenses grouped by categories is in the Appendix A.8.

**Defenses for different IQA metrics' architectures**. The chosen IQA metrics fall into categories by their backbones: CNN-based (META-IQA, KonCept, SPAQ, PAQ2PIQ, Linearity), transformer-based (Maniqa, CLIP-IQA+), and custom (FPR, TOPIQ). We show $R$ $score$ and $SROCC_{clear}$ of purification defenses applied for different IQA metric types in a non-adaptive scenario in Figure 4. Transformer-based metrics have greater $R$ $score$ robustness even without any defense. The high robustness of such metrics yielded a low robustness increase after applying defenses. For other metrics, most defenses increased the robustness. DISCO managed to improve the robustness of all metrics, but the effect was much stronger on CNN-based metrics. JPEG defense was one of the best to improve the $R$ $score$ of transformer-based metrics. Defended transformer-based metrics showed a higher correlation decrease than metrics of other architectures. It should be noted that custom architecture can be highly vulnerable. FPR model shows the worst $R$ $score$. This vulnerability is likely caused by its atypical architecture for the NR-IQA task, which includes a Siamese network and an attempt to "hallucinate" the features of the pseudo-reference image from a distorted one. These results correlate with the previous researchAntsiferova et al. (2024)

**Defenses for regular/AI-gen image content** Figure 3 (b) compares $D_{score}$ (left) and $SROCC_{adv}$ (right) for purification methods by different datasets. Three datasets contained natural scene images, and AGIQA-3K contained AI-generated images. There is no significant difference in defense efficiency for most methods between datasets, but some advanced defenses based on neural networks (Real-ESRGAN, DISCO) have larger discrepancies. As shown in Figure 3 (right), correlations depend highly on the dataset. On average, $SROCC_{adv}$ on KonIQA1k is significantly higher than on KADID and AGIQA-3K, similar to results in Table 25 in the Appendix regarding $SROCC_{clear}$. This can be due to two factors: a) Several IQA models (e.g., TOPIQ and CLIP-IQA+) were trained on the KonIQ-10k dataset or its subsets, giving them a natural advantage on KonIQ-1k. b) Certain IQA models, such as MetaIQA and PAQ2PIQ, generally achieve higher correlation values on

KonIQ-10k, as reported in their respective studies, suggesting an inherent dataset bias. This figure also reveals that defenses can boost the correlation coefficients compared to scenarios W/o Defense.

**Guarantees of the defenses.** Among all the methods compared, only certified methods provide theoretically reliable predictions. Table 4 presents the results for certified defenses. Classification-based methods (RS, DRS, DDRS, and DP) guarantee that perturbations with an $l_2$-norm less than the certified radius will not alter the metric score. Similarly, regression-based methods (MS and DMS) assure that any perturbation with an $l_2$-norm less than $\epsilon = 0.05$ will cause the metric score to change by no more than the certified delta. Compared to more sophisticated methods, simple randomized smoothing showed the highest certified radius. Among regression-based methods, Denoised Median Smoothing showed the lowest certified relative delta.

**Perceptual quality of defended images** The perceptual quality of defended images changes only after purification approaches. Most presented purification defenses aim to restore the original content of the image. However, the restored images differ from the original and may have flaws in the form of artifacts. The examples of defenses' performance are in the Appendix A.8. The most noticeable artifacts caused by purification defenses include removing details of the original image (DISCO, MPRNet), altering the image content (Real-ESRGAN, DiffPure), reducing the image clarity (Diff-Pure, blur defenses), and compression artifacts (JPEG/DiffJPEG, Color Quantization). Figures 12 and 11 in the Appendix A.10 show examples of images after applying different purification methods. Table 3 shows perceptual quality metrics for purification defenses. Adversarial images without defenses were closer to the original images than purified ones. Color Quantization and Bilinear Upscale are more successful in restoring the original image in non-adaptive cases than other defenses. Flip, Rotate, and Crop show the worst results because they transform the attacked image without changing its content so that the $PSNR$ and $SSIM$ do not apply to them.

**Statistical tests** We used a one-sided Wilcoxon Signed Rank Test with Bonferroni correction to reduce the risk of false positives due to the many tested hypotheses. The defenses were compared pairwise, with each pair yielding a percentage indicating how often one defense statistically outperformed the other under adversarial attack conditions. Tables in the Appendix A.9 present test results on KonIQA, NIPS, and KADID1K datasets. The results show that the top performers include DiffJPEG, DISCO, and DiffPure, which show high superiority percentages against most other defenses. The results from these tables intuitively make sense when considering the design and complexity of each defense. Further statistical tests, including evaluations of quality scores, expand the analysis to both adaptive and non-adaptive scenarios in the Appendix A.9.

## 5    CONCLUSION

This paper analyzed the efficiency of 29 adversarial defense methods against a wide range of attacks for IQA metrics. We showed that defending IQA metrics is more challenging than object classification models due to the additional requirement of preserving an image's quality. According to the results, adversarial training is the best defense in three out of four criteria: it offers a zero inference computational overhead, no distortions and may provide high correlations. However, for our task of defending IQA metrics, it showed lower robustness than purification methods. Among purification defenses, DiffJPEG, DiffPure, and Real-ESRGAN offer good performance-robustness trade-offs, but the latter two methods are more vulnerable to adaptive attacks. Certified defenses are also efficient in all criteria but one, which is inference computational time. Suprisingly, some of the purification approaches showed comparable robustness to randomized smoothing in our settings. We published the dataset of adversarial images used in nonadaptive scenarios and the results as an online leaderboard. This dataset and the benchmark can be helpful for researchers and companies who want to make their IQA metrics more robust to potential attacks. Although the proposed benchmark can identify the most effective defense methods against adversarial attacks on IQA metrics, it can also pinpoint attacks most resistant to these defenses, which can be considered to have a potential negative social impact. By publishing our findings, we highlight the necessity of further research on incorporating defense methods in image quality assessment. The limitations of our comparison listed in the Appendix A.1 are mostly related to parameters of attacks and defenses. Due to the extreme measurement complexity, we varied only one parameter for most defenses, and a more in-depth evaluation is a subject of our future work.

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

# A APPENDIX

## A.0 STRUCTURE

- Section A.1 outlines the limitations of our work, including attack parameters, transferability, and ranking methodology.
- Section A.2 gives a more in-depth analysis of used datasets, including their characteristics and relevance to the study.
- Section A.3 presents a list of parameters of IQA metrics and explains the methodology for calculating their ranges.
- In section A.4, we describe evaluated attacks with their parameters.
- Section A.5 describes each defense method, evaluated in our study, including more in-depth principles behind certified defense.
- Section A.6 provides quantitative results for non-adaptive attacks. It proves that alignment for attack strength is correct.
- In section A.7 we provide more in-depth analysis of purification defenses.
- Section A.8 includes Figures and Tables with additional results. It also provides an analysis of the difference between SROCC and PLCC.
- Section A.9 presents statistical tests (in particular, one-sided Wilcoxon Signed Rank Tests) for observed results and supports the points we made in the main paper.
- Section A.10 provides some visual examples of attacked and defended images to illustrate the impact of the evaluated methods.

## A.1 LIMITATIONS

While the proposed framework for benchmarking defenses against adversarial attacks on IQA metrics offers significant contributions, we acknowledge the existence of the following limitations to be addressed in future work:

1. **Handling Multiple Parameter Attacks**: The current framework deals mainly with attacks that have a single parameter. However, Some attacks might have multiple parameters to control their strength, complicating the evaluation process. Moreover, a group of Boundary Attacks adapts their parameters according to the response of the attacked model, which poses an additional challenge in a fixed-parameter setting. Future versions will include methods for dealing with different types of evolving attacks, possibly through dynamic parameter optimization techniques

2. **Transferability of Adversarial Attacks**: There might be defenses that better generalize to attacks produced on other defenses. Currently, the framework does not evaluate the transferability of adversarial attacks among different defenses. Future versions should provide insights into the generalizability and robustness of the defense.

3. **Simplified Ranking Methodology**: The current framework employs a straightforward ranking methodology that may not fully capture the complexity and the existence of different evaluation metrics with varying importance levels depending on the attack used for testing. Different evaluation measures can be assigned different weights based on their importance and relevance to the attack. This system allows for a composite score that reflects the overall performance of a defense mechanism. To provide a nuanced assessment of metric robustness, a more rigorous statistical framework for ranking metrics will be employed in future versions.

Addressing these limitations in future work will ensure the framework's robustness and adaptability in diverse and realistic scenarios.

## A.2   DATASETS

In Table 5 we provide information about the datasets used in our study.

Table 5: List of datasets used in our benchmark.

| Dataset | Size | Resolution | Subjective ratings | Short description |
| --- | --- | --- | --- | --- |
| KonIQA-10k | 1,000 (out of 10,073) | $512 \times 384$ | 120,000 | Provides wide range of real-world photos with authentic distortions |
| KADID-10k | 8 out of 81 original images | $512 \times 384$ | 30,000 | Large-scale dataset with wide variety of content and artificial distortions |
| NIPS 2017 | 1,000 | $299 \times 299$ | — | Competition on adversarial examples and defenses in the NIPS 2017 |
| AGIQA-3K | 2,982 | $512 \times 512$ | 125,244 | AGIs from GAN-/auto-regression-/diffusion-based model with subjective scores |

We use a subset of 1000 images at $512 \times 384$ resolution from the freely available to the research community KonIQA-10K IQA dataset Hosu et al. (2020) to evaluate adversarial defense methods. We chose it due to its large, diverse collection of real-world images with subjective quality scores. The original set was partitioned into 10 clusters using K-Means Lloyd (1982) based on 3 parameters: Spatial Information (SI), Colorfulness (CF), and Mean Opinion Scores (MOS). We selected 100 random images from each cluster, resulting in a diverse set of test images regarding quality and content. Using these images as the source, we generated an adversarial dataset of over 215,000 images (1000 × number of attacks × number of attacked NR IQA metrics × number of attack hyperparameter sets). Figure 5 shows that the distribution for the 1,000-image sample is nearly identical to the distribution for the full dataset. Due to significantly higher computational complexity, we used a smaller set of 50 images (5 per cluster) for black-box attacks. For the same reason, we used a smaller set of 10 images for certified defenses to generate attacks. To evaluate the impact of sampling this procedure was repeated 10 times, focusing on purification methods and black-box attacks to accelerate calculations. We used default parameters for defense methods and 3 presets from the main paper for attacks. For each IQA model, attack and defense method, we calculated four scores per sample: $D_{score}$, $SROCC_{clear}$, $SROCC_{adv}$, and $SSIM$.

Figure 6 illustrates the distribution of these scores for each sample. The results show that the distributions are nearly identical across all samples and metrics, with consistent mean values. To assess the differences between the means of distributions, we computed the mean for each distribution and score, yielding a list of 10 mean values per score. Then, we calculated the mean and variance of these values across the 10 samples. These values can be found in Table 6. To verify these findings statistically, we performed a Kruskal-Wallis test Kruskal & Wallis (1952) for each metric across the 10 samples. The p-values are shown in Table 6. These p-values indicate no significant differences between the samples, confirming that the sampling procedure does not introduce variability into the evaluation results. This consistency strengthens our conclusions and ensures that the findings are robust across different random subsets of the dataset.

## A.3   IQA METRICS

We define metric range as $\text{diam}(f_\omega) = \sup_{x,z \in X} \{|f_\omega(x) - f_\omega(z)|\} = \text{upper} - \text{lower}$, where upper is called the upper metric bound and lower - lower metric bound. To calculate these bounds, we used the DIV2K_valid_HR subset from the DIV2K dataset (Agustsson & Timofte (2017)). The upper bound is set to the highest metric value across the chosen subset, while the lower bound is set to

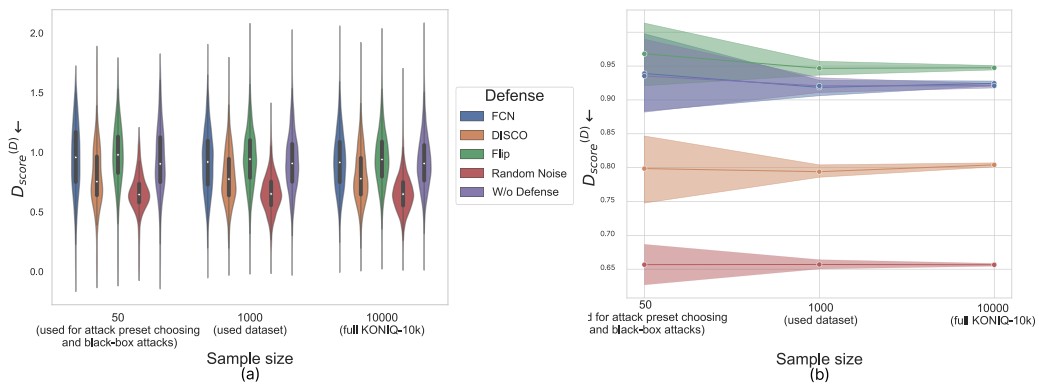

Figure 5: (a): Distribution of $D_{score}^D$ on different source dataset sizes. (b): Mean $D_{score}^D$ and corresponding confidence intervals.

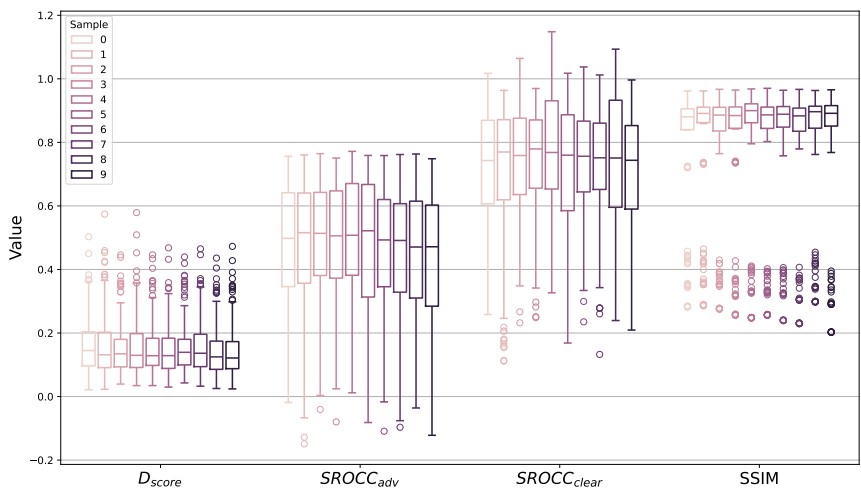

Figure 6: The effect of sampling 50 images on results.

the minimum value between the lowest metric value on subset images compressed with JPEG with quality of 10 and sampled random noise of the image subset size.

We report metric ranges in Table 7 and metric parameters in Table 8. The $R_{score}$ is taken from Antsiferova et al. (2024).

## A.4 USED ADVERSARIAL ATTACKS

The early methods for attacking IQA metrics were initially developed to stress-test their performance. In 2008, Wang & Simoncelli (2008) introduced the **MADC** method, which utilizes gradient projection onto a proxy FR quality metric. This method generates an image with the same quality level according to proxy metrics, used to compare the metrics' accuracy. Many years later, Kurakin et al. (2018) proposed creating an adversarial image by iteratively adding the model's gradient with respect to the image (**I-FGSM**). However, this approach produces highly visible distortions in low-frequency areas. To address this issue, Korhonen & You (2022) proposed multiplying the gradients by a weights map produced using the Sobel filter and morphological operations to add distortions only in high-textured regions (**Korhonen et al.**). In another work by Zhang et al. (2022b), the authors suggested incorporating a FR IQA component into the loss function to control visual quality.

Table 6: Results across 10 samplings of 50 images.

| Score | Mean | Variance of means for each sample | p-value after Kruskal-Wallis test |
|---|---|---|---|
| $D_{score}$ | 0.1533 | 0.000044 | 0.5425 |
| $SROCC_{adv}$ | 0.4681 | 0.00043 | 0.1449 |
| $SROCC_{clear}$ | 0.7343 | 0.00066 | 0.1958 |
| $SSIM$ | 0.7953 | 0.000034 | 0.1138 |

Table 7: List of NR IQA metric ranges in our benchmark.

| Metric | Lower bound | High bound | Metric range |
|---|---|---|---|
| META-IQA (Zhu et al. (2020)) | 0.000 | 1.000 | 1.000 |
| MANIQA (Yang et al. (2022)) | 0.000 | 1.000 | 1.000 |
| KonCept (Hosu et al. (2020)) | 26.403 | 66.869 | 40.466 |
| SPAQ (Fang et al. (2020)) | 21.749 | 77.755 | 56.006 |
| PAQ2PIQ (Ying et al. (2020)) | 58.380 | 84.171 | 25.791 |
| Linearity (Li et al. (2020)) | 25.780 | 83.226 | 57.446 |
| FPR (Chen et al. (2022a)) | 47.225 | 77.047 | 29.822 |
| CLIP-IQA+ (Wang et al. (2023)) | 0.000 | 1.000 | 1.000 |
| TOPIQ (Chen et al. (2024)) | 0.215 | 0.822 | 0.607 |

Table 8: List of NR IQA metrics used in our benchmark.

| Metric | $R_{score} \uparrow$ | Backbone | Number of parameters | Input transformations | Code |
|---|---|---|---|---|---|
| META-IQA (Zhu et al. (2020)) | 1.168 | ResNet-18 | 13.2M | ImageNet Normalization | Github |
| MANIQA (Yang et al. (2022)) | 0.986 | ViT-B/8 | 135.62M | $224 \times 224$ crop | Github |
| KonCept (Hosu et al. (2020)) | 0.584 | InceptionResNetV2 | 59.82M | Normalization (0.5, 0.5) | Github |
| SPAQ (Fang et al. (2020)) | 0.493 | ResNet-50 | 23.5M | $224 \times 224$ crop | Github |
| PAQ2PIQ (Ying et al. (2020)) | 0.449 | ResNet-18 | 11M | — | Github |
| Linearity (Li et al. (2020)) | 0.267 | ResNeXt-101 | 90M | ImageNet Normalization | Github |
| FPR (Chen et al. (2022a)) | -0.229 | Custom | 16.6M | Splitting into fixed-sized patches | Github |
| CLIP-IQA+ (Wang et al. (2023)) | | CLIP | 244M | — | Github |
| TOPIQ (Chen et al. (2024)) | | Custom | 45M | — | Github |

They used metrics such as DISTS, SSIM, and LPIPS (**Zhang et al.**). Another approach to reducing attack visibility is **SSAH** proposed by Luo et al. (2022), which decomposes the image into low and high frequencies and generates an attack only in high frequencies. Bhattad et al. (2019) introduced the **cAdv** method, which operates in the LAB color space instead of the pixel space. Some methods, such as those proposed by Moosavi-Dezfooli et al. (2017) and Baluja & Fischer (2017), work much faster since they do not require a backpropagation step during inference. Other fast-working methods are based on creating universal adversarial perturbations (**UAP**) (Shumitskaya et al., 2022; 2024), that can also be applied to the task of video quality assessment. As an improvement of this idea, Shumitskaya et al. (2023) proposed the **FACPA** method, which requires initial training on low-resolution data and can then be efficiently applied to high-resolution data.

For black-box attacks, we chose to adopt several approaches that were designed for image classifiers.

The **NES** attack, proposed by Ilyas et al. (2018), estimates gradient using the natural evolutionary strategy (NES) and then performs gradient descent with the approximation to minimize the objective. The **Parsimonious attack**, proposed by Moon et al. (2019), searches for perturbations consisting of pixels with $\pm\varepsilon$. It defines the working set as the set of pixels in the perturbation that have pixel value of $+\varepsilon$, and updates it in Lazy Greedy Insert and Lazy Greedy Deletion. This method also uses hierarchical lazy evaluation and starts from coarse blocks and finishes with pixel updates. Andriushchenko et al. (2020) introduced the **Square attack** that updates perturbation according to a random search algorithm. The update is generated as a square patch which is applied to existing perturbation. If the update improves the objective, the patch is applied; otherwise, the perturbation remains unchanged. Parsimonious and Square attacks were chosen as they are among the most efficient score-based black-box attacks and can be easily converted to attacking IQA models. **Patch-RS** was chosen to represent black-box sparse attacks. NES was chosen to represent gradient estimation

Table 9: List of adversarial attacks used in our benchmark. WB and BB are white-box and black-box attack types. We adjust varied parameters to align attacks' strengths.

| Adversarial attack | Type | Restriction | Varied parameter | Fixed parameters |
|---|---|---|---|---|
| I-FGSM (Kurakin et al. (2018)) | WB | $l_\infty$ | lr | eps = 10 / 255, iters = 10 |
| Optimised-UAP (Shumitskaya et al. (2024)) | WB | $l_\infty$ | amplitude | eps = 0.1, lr = 0.001, n_epoch = 5 |
| Korhonen et al. (Korhonen & You (2022)) | WB | $l_\infty$ | lr | iters = 10 |
| Zhang et al. (Zhang et al. (2022b)) | WB | $l_\infty$ | lr | iters = 10 |
| MADC (Wang & Simoncelli (2008)) | WB | $l_\infty$ | lr | eps = 10 / 255, iters = 8 |
| cAdv (Bhattad et al. (2019)) | WB | SSIM | lr | — |
| SSAH (Luo et al. (2022)) | WB | $l_\infty$ | lr | lambda_lf=0.1 |
| FACPA (Shumitskaya et al. (2023)) | WB | $l_\infty$ | amplitude | n_epoch=5, lr = 0.001, $\varepsilon = 10/255$ |
| NES (Ilyas et al. (2018)) | BB | $l_\infty$ | $\epsilon$ | sigma=0.001, N=32, n=20, eta=0.1, max_iters=250 |
| Parsimonious (Moon et al. (2019)) | BB | PSNR | $\epsilon$ | max_queries=10000, batch_size=64, block_size=32, max_iters=10000 |
| One Pixel (Su et al. (2019)) | BB | $l_0$ | pixel count | POPSIZE=300, batch_size=32, iters=5 |
| Square (Andriushchenko et al. (2020)) | BB | $l_\infty$ | $\epsilon$ | p_init=0.05, max_queries=10000 |
| Patch-RS (Croce et al. (2022)) | BB | PSNR | $\epsilon$ | max_queries=10000, p_init=0.8, n_restarts=1 |

Table 10: List of varied parameters' values of adversarial attacks.

| Adversarial attack | Type | Varied parameter | Parameter values |
|---|---|---|---|
| I-FGSM (Kurakin et al. (2018)) | WB | lr | 4e-04, 7e-04, 1e-03, 2e-03, 3e-03, 5e-03, 8e-03, 14e-03, 2e-02, 4e-02 |
| Optimised-UAP (Shumitskaya et al. (2024)) | WB | amplitude | 0.1, 0.189, 0.278, 0.367, 0.456, 0.544, 0.633, 0.722, 0.811, 0.9 |
| Korhonen et al. (Korhonen & You (2022)) | WB | lr | 5e-05, 1e-04, 2e-04, 5e-04, 1e-03, 2e-03, 5e-03, 1e-02, 2e-02, 5e-02 |
| Zhang et al. (Zhang et al. (2022b)) | WB | lr | 5e-05, 1e-04, 2e-04, 5e-04, 1e-03, 2e-03, 5e-03, 1e-02, 2e-02, 5e-02 |
| MADC (Wang & Simoncelli (2008)) | WB | lr | 1e-05, 2e-05, 5e-05, 1e-04, 2e-04, 5e-04, 1e-03, 2e-03, 5e-03, 0.01 |
| cAdv (Bhattad et al. (2019)) | WB | lr | 5e-05, 1e-04, 2e-04, 5e-04, 1e-03, 2e-03, 5e-03, 1e-02, 2e-02, 5e-02 |
| SSAH (Luo et al. (2022)) | WB | lr | 1e-4, 2e-4, 3e-4, 5e-4, 8e-4, 13e-4, 22e-4, 36e-4, 6e-3, 0.01 |
| FACPA (Shumitskaya et al. (2023)) | WB | amplitude | 0.1, 0.189, 0.278, 0.367, 0.456, 0.544, 0.633, 0.722, 0.811, 0.9 |
| NES (Ilyas et al. (2018)) | BB | $\epsilon$ | 0.01, 0.014, 0.02, 0.027, 0.038, 0.053, 0.074, 0.103, 0.143, 0.2 |
| Parsimonious (Moon et al. (2019)) | BB | $\epsilon$ | 0.01, 0.014, 0.02, 0.027, 0.038, 0.053, 0.074, 0.103, 0.143, 0.2 |
| One Pixel (Su et al. (2019)) | BB | $l_0$ | 5.0, 10.0, 15.0, 20.0, 25.0, 30.0, 35.0, 40.0, 45.0, 50.0 |
| Square (Andriushchenko et al. (2020)) | BB | $\epsilon$ | 0.01, 0.014, 0.02, 0.027, 0.038, 0.053, 0.074, 0.103, 0.143, 0.2 |
| Patch-RS (Croce et al. (2022)) | BB | $\epsilon$ | 49, 64, 81, 121, 169, 225, 289, 361, 484, 625 |

methods. **Patch-RS** from Sparse-RS, proposed by Croce et al. (2022), also uses random search to create a square patch and search for its place on the image. Su et al. (2019) suggested to change a predefined number of pixels to make their **One Pixel** attack successful using Differential Evolution algorithm.

We report the list of parameters for attacks and fixed values for non-varied parameters in Table 9 and the list of varied parameter values in Table 10.

### A.5 EVALUATED DEFENSES

#### A.5.1 PURIFICATION

According to Guo et al. (2018), several standard preprocessing techniques for images can be used as defenses against additive adversarial noise. These methods include compression (**JPEG**, **Diff JPEG** (Reich et al. (2024)) **color quantization** (Xu et al. (2018))), spatial transformations (**Resize**, **Rotate**, **Crop**, **Flip**), blurring (**Median blur**, **Gaussian blur**, etc.), **Unsharp masking**, and others. Although not originally designed as defenses, studies have shown these methods can be effective. Since adversarial perturbations are often high-frequency noise, denoising techniques can help remove them. The Multi-Stage Progressive Image Restoration Network (**MPRNet** (Zamir et al. (2021))) is a three-stage convolutional neural network for image deblurring, deraining, and denoising. The first two stages use an encoder-decoder architecture for multi-scale contextual information, while the final stage operates at the original resolution to preserve details. MPRNet features supervised attention modules and cross-stage feature fusion for effective information transfer. **Real-ESRGAN** (Wang et al. (2021)), a GAN-based model with several residual dense blocks for super-resolution, is trained with synthetic data and can be used for adversarial denoising. Another idea for adversarial denoising is based on applying diffusion models. **DiffPure** (Nie et al. (2022)) employs diffusion models to purify

Table 11: List of compared adversarial Purification methods.

| Defense method | Type | Varied parameter | Fixed parameters | Parameter values | Code |
|---|---|---|---|---|---|
| JPEG (Guo et al. (2018)) | Compression | q | — | 10, 30, 50, 70, 90 | — |
| DiffJPEG (Reich et al. (2024)) | Compression | q | — | 10, 30, 50, 70, 90 | Github |
| Color quantization (Xu et al. (2018)) | Compression | npp | — | 2, 5, 16, 20, 25 | — |
| Resize (Guo et al. (2018)) | Spat. transform. | scale | — | 0.1, 0.25, 0.5, 0.75, 0.9 | — |
| Bilinear Upscale | Spat. transform. | scale | — | 0.1, 0.25, 0.5, 0.75, 0.9 | — |
| Rotate | Spat. transform. | angle lim. | — | 10, 15, 20, 30, 50 | — |
| Crop (Guo et al. (2018)) | Spat. transform. | size | — | 32, 64, 128, 256, 288 | — |
| Flip | Spat. transform. | — | — | — | — |
| Gaussian blur | Blurring | kernel size | sigma=0.15*kernel_size+ 0.35 | 3, 5, 7, 9, 11 | — |
| Median blur | Blurring | kernel size | — | 3, 5, 7, 9, 11 | — |
| Unsharp masking | Preprocessing | kernel size | sigma=1, amount=1 | 3, 5, 7, 9, 11 | — |
| MPRNet (Zamir et al. (2021)) | Denoising | — | — | — | Github |
| Real-ESRGAN (Wang et al. (2021)) | Denoising | — | denoise_strength=0.2, outscale=1, tile=0, tile_pad=10, pre_pad=0 | — | Github |
| DiffPure (Nie et al. (2022)) | Defense | t | t_delta=15, diffusion_type=ddpm, sample_step=1 | 5, 10, 20, 30, 50 | Github |
| DISCO (Ho & Vasconcelos (2022)) | Defense | — | — | — | Github |
| FCN (Gushchin et al. (2024)) | Defense | — | — | — | Github |
| Random noise | Adding noise | — | — | — | — |

adversarial images by introducing a small amount of noise through forward diffusion, then reversing the process to recover a clean image. **DISCO** (Ho & Vasconcelos (2022)) is an image purification method aimed at enhancing classification robustness. It employs local implicit functions to ensure small perturbations do not significantly alter local data representations. By maintaining these robust local representations, DISCO effectively resists adversarial perturbations that do not align with the data's local structure. Some adversarial attacks, such as color-based modifications, are not bounded. Standard denoising approaches are ineffective against these. Gushchin et al. (2024) proposed neural filter **FCN**, to counter color attack cAdv on image quality metrics. FCN features a compact, fully convolutional architecture with three hidden layers of 64, 32, and 3 filters, optimized with Adam and MSE loss.

We report the list of parameters for defenses and fixed values for non-varied parameters in Table 11.

### A.5.2 ADVERSARIAL TRAINING

We fine-tuned Linearity and KonCept IQA models using the original images and attacked images in a 1:1 ratio from the original KonIQA-10K training dataset for 30 epochs. During the training process, we used a 2-step APGD attack Croce & Hein (2020) to generate the attacked images. This method uses an adaptive step that allows a small number of iterations to achieve strong adversarial examples and reduce computational time. The goal of the attack during the training process is to increase model loss. We adjusted the MOS values based on the FR metric scores. For a given original image $x$ with MOS $y$, we obtain the adjusted MOS for the attacked image $x'$ as follows:

$$y' = y - M(x, x') \tag{8}$$

We have considered LPIPS and 1 - SSIM as M. To evaluate the impact of attack magnitude during training we chose 3 different attack magnitudes $\varepsilon = \{2, 4, 8\}/255$.

### A.5.3 CERTIFIED METHODS

**Description.** Cohen et al. (2019) proposed the **Randomized Smoothing (RS)** method to transform any classifier that performs well under Gaussian noise into a new classifier that is certifiably robust to adversarial perturbations under the $l_2$ norm. The overall process of this defense can be described as follows: given an input image, the algorithm samples $N$ noisy variations of this image using a Gaussian noise model with a certain $\sigma$. These images are then passed through the backbone classification model, and the most frequently predicted class is given as the final answer. This approach results in an algorithm that provides a provable answer for the model within a $l_2$ ball. The radius of this ball is calculated based on the difference between the most popular and the second most popular classes across the sampled images used for answer selection. The main disadvantage of the previous approach is that running the classifier on noisy data causes a drop in model accuracy, as it was not trained to handle such data. To address this issue, Salman et al. (2020) extended randomized smoothing to **Denoised Randomized Smoothing (DRS)** by denoising the noisy image before passing it to the model. Since the noise model is known, training an effective denoiser for a given $\sigma$ is relatively straightforward. Carlini et al. (2022) extended the approach of Salman et al. (2020) by replacing the denoiser with a pre-trained denoising diffusion probabilistic model

Table 12: Experiment to determine the optimal number of classes $N$ for regression metric discretization.

| $N$ | $SROCC_{clear} \uparrow$ (no Monte-Carlo sampling) | $Cert.R \uparrow$ (with Monte-Carlo sampling) |
|---|---|---|
| 3 | 0.49 | 0.249 |
| 5 | 0.53 | 0.248 |
| **10** | **0.56** | 0.206 |
| 15 | 0.56 | 0.160 |
| 20 | 0.56 | 0.142 |
| $\infty$ | 0.56 | 0 |

(**Diffusion Denoised Randomized Smoothing (DDRS)**). They used only one diffusion step because it demonstrated high speed and relatively good quality. Chen et al. (2022b) proposed the **DensePure (DP)** method that involves multiple runs of denoising via the reverse process of the diffusion model (using different random seeds) to generate multiple samples. These samples are then passed through the classifier, and the final prediction is made using majority voting.

Chiang et al. (2020) proposed a method to certify regression models. Instead of using the most popular class within the $l_2$ ball, they utilize the median of function values. They also theoretically demonstrated that using the median is better than the mean. We denote this method as **Median Smoothing (MS)**. They further extended the method to **Denoised Median Smoothing (DMS)** by adding a denoising step before model prediction to improve accuracy.

**Parameters selection.** Given an input image, the results of the classification-based certified method are the metric score and the certified radius $R$. The method guarantees that the class remains unchanged for the input image within a $l_2$ ball of radius $R$. All classification-based certified methods were run with the following parameters: $\sigma = 0.12, N_0 = 100, N = 1000, \alpha = 0.001$. Here, $\sigma$ is the standard deviation of the Gaussian noise used for sampling, $N_0$ is the number of samples for class selection, $N$ is the number for class certification, and alpha is the probability of class change within the $l_2$ ball of the predicted certified radius $R$.

Given an input image, the results of a regression-based certified method are the metric score and the certified delta. The method guarantees that, within a $l_2$ ball of radius $\epsilon$, the metric score changes by no more than delta. To make this value comparable across metrics, we define the certified relative delta by dividing the certified delta by the metric range. All regression-based certified methods were run with the following parameters: $\sigma = 0.12, \epsilon = 0.05, N = 1000, \alpha = 0.001$.

Scripts for running all these methods are available on GitHub.

**Classifier-based methods application.** To discretize a regression quality metric for classification-based methods, we divided the metric range into $N$ segments, each corresponding to a specific class. We also added additional classes for metric values that fall below or above the calculated range, ensuring that every metric value is assigned to a class. This resulted in a $(N + 2)$-class metric-classifier. Note that these classes are ordered, with higher class values indicating better quality. Thus, we can measure the quality of the classifier metric in the same way as the regression metric – using relative gain and correlations with subjective scores.

We conducted additional experiments to determine the optimal value of $N$ on PAQ2PIQ NR metric. The main challenge is balancing the trade-off between $SROCC_{clear}$ and $Cert.R$. As the number of classes increases, $SROCC_{clear}$ also increases, but $Cert.R$ decreases. This occurs because a higher number of classes makes it easier to cross class borders during Monte Carlo sampling. Table 12 presents the results of our experiment, indicating that when the number of classes is set to ten, $SROCC_{clear}$ on the discrete metric without Monte Carlo sampling is optimal. Additionally, we measured $Cert.R$ for this number of classes and discovered that $Cert.R$ does not significantly decrease for $N = 10$. Therefore, we chose $N = 10$ to discretize NR metric values in the main experiments of this paper.

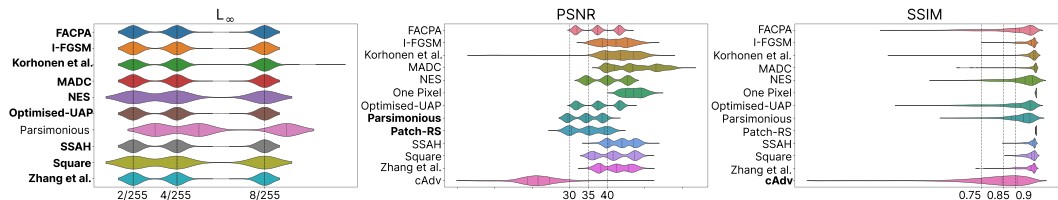

Figure 7: The intensity of adversarial attacks averaged across all IQA metrics. The dotted vertical lines show the targeted strength values for which the preset parameters of the bolded attack methods were selected.

## A.6 NON-ADAPTIVE CASE DATASET

In figure 7, we show the intensity of the applied attacks, averaged over all IQA metrics. For each attack method, a metric of intensity was selected among $L_\infty$, PSNR, and SSIM. Marked with dotted lines, the values represent specific levels of attack intensity that we aimed to achieve with the presets for the highlighted attack methods. One Pixel and Patch-RS attacks are hidden in the first plot, since these methods allow any change in the $L_\infty$ norm.

## A.7 MORE IN-DEPTH ANALYSIS

Here we provide some findings to better understand the underlying reasons for defense performance. First, adversarial perturbations generally consist of high-frequency noise. For this reason, defenses that employ compression in some way are effective. JPEG and DiffJPEG remove high-frequency noise alongside adversarial perturbations while maintaining the structural information of a clean image, as the perturbation has a far more complex and unnatural representation than the original high-frequency parts of an image. This conclusion suggests that developers of purification methods should analyze how the high-frequency components of an image and perturbations differ. DISCO uses an encoder-decoder architecture, which is a similar approach to compression. The learned features of clean images help DISCO project images back onto the natural image manifold. Denoising methods, such as MPRNet and Real-ESRGAN, show average results compared to other techniques since they were trained on noise of a simpler nature, while adversarial perturbations possess more complex high-frequency structures. Fine-tuning on adversarial perturbations is a subject for future research.

Diffusion-based models offer high variability in strength due to their architecture. They can be precisely tuned for the desired adversarial attack of a particular budget. On the other hand, DiffPure introduces its own processing artifacts, causing the worst correlations and low PSNR and SSIM compared to original images. This reveals a significant difference between applying diffusion-based defenses in classification tasks (where they are state-of-the-art methods) and quality assessment tasks.

However, to effectively mitigate adaptive attacks, methods need to employ high randomness. Particularly effective is the combination of randomness and geometric transformations, as perturbations are vulnerable to them. Flip and Random Rotate are great examples. The first lacks randomness, and adaptive attacks easily surpass it, while Random Rotate significantly reduces attack effectiveness since the angle differs between the calculation of an attack and inference.

## A.8 ADDITIONAL RESULTS

We show performances for evaluated defenses in tables below. Confidence intervals in some table are large due to the fact that we calculate average score across large pool of IQA metrics/attacks/datasets. To statistically check what defense is better we provide results of statistical tests A.9.

We analyse how much does $SROCC$ and $PLCC$ correlations are differ in table 26. It reports that ranks in both cases are identical. Thus, in other tables we report only $SROCC$.

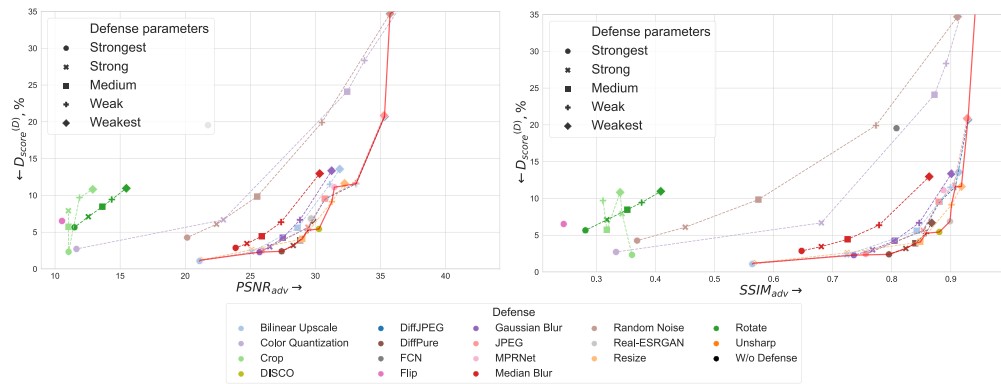

Figure 8: $D_{score}^{(D)}(\downarrow)/PSNR(\uparrow)$ (left) and $D_{score}^{(D)}(\downarrow)/SSIM(\uparrow)$ tradeoffs for Purification-based defenses in non-adaptive scenario averaged across KonIQA, KADID and AGIQA-3K datasets. Red line denotes the Pareto Optimal front.

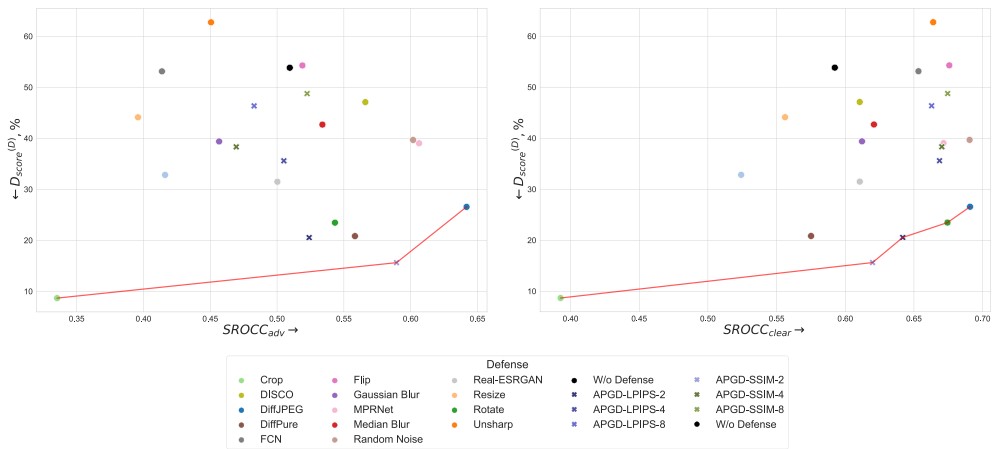

Figure 9: $D_{score}^{(D)}(\downarrow)/SROCC_{adv}(\uparrow)$ (left) and $D_{score}^{(D)}(\downarrow)/SROCC_{clear}(\uparrow)$ tradeoffs for Purification-based and Adversarial Training defenses in adaptive scenario averaged across KonIQA, KADID and AGIQA-3K datasets. Red line denotes the Pareto Optimal front.

Table 13 reports results for purification defenses on KonIQA and KADID datasets for adaptive and non-adaptive cases. The results show, that DiffJPEG and JPEG show best correlations for metrics, while Rotate and Diffpure increase $R\ score$ the best.

Table 14 shows how well defenses can respond to attack of different strength. In summary, results do not change much across strength.

Tables 15 and 16 present results of purification defenses on different IQA metrics. We can see that correlations are highly dependent on IQA metric, while top methods in terms of robustness ($D_{score}$, $R_{score}$) are similar accross different IQA models.

Table 17 shows scores for different attacks types. The experiment showed that undefended images are more close to the original than defended by any defense. The results are the same as on all attack types. JPEG and DiffJPEG show greater $SROCC_{adv}$, while Color Quantization has better PSNR with the original images.

Table 24 reports results for grouped purification defenses. The results demonstrate that compression is one of the best in all metrics except $R_{score}^{(D)}$.

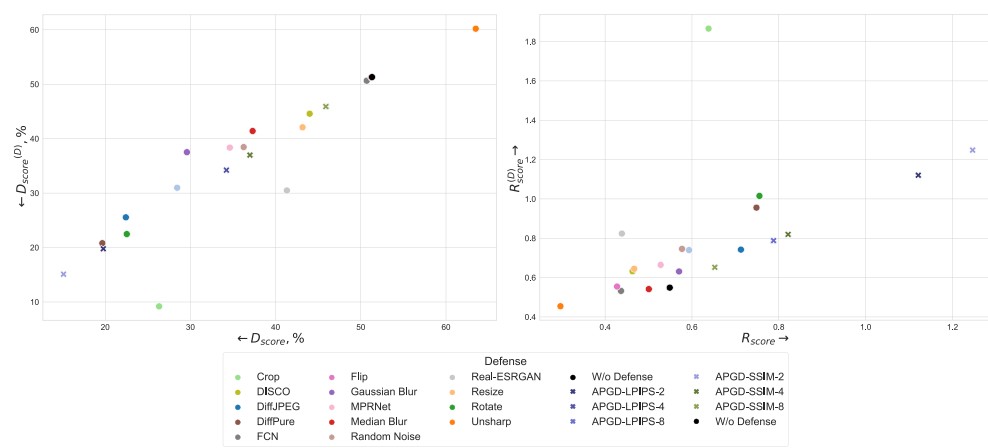

Figure 10: Comparison of $D_{score}$ and $D_{score}^{(D)}$ (left) and $R_{score}/R_{score}^{(D)}$ (right) for Purification-based and Adversarial Training defenses in adaptive scenario. Results are averaged across KonIQA, KA-DID and AGIQA-3K datasets.

Table 25 compares results on different datasets. The results differ much, but simple defenses like Flip and Rotate are effective on all datasets.

Table 27 report some score for certified methods. It shows that RS has bigger $Cert.R$ and $Abst.$ among all classification-based defenses, while MS is the best among regression-based defenses.

Figure 8 pictures tradeoffs for Purification-based defenses. It shows the importance of tuning defense parameters for defenses with different parameters.

Figure 9 compares correlation coefficients with $D_{score}^{(d)}$. The picture demonstrates the efficiency of the simple transforms.

Figure 10 illustrates difference between $D_{score}$ $D_{score}^{(D)}$ (left) and $R_{score}$ and $R_{score}^{(D)}$ (right). The main finding is that there is high correlation within these pairs of scores with rare exceptions like Random Crop.

A.9   STATISTICAL TESTS

We used the one-sided Wilcoxon Signed Rank Test as a statistical test since it is appropriate for comparing paired samples, as it is non-parametric and does not assume normality of the underlying data, making it ideal for adversarial robustness scenarios, where the distribution of data (IQA metrics, in our case) may not follow normal patterns. This test provides insights into whether a defense mechanism shows a statistically significant improvement over another in terms of $D_{score}$, a metric that reflects image quality after adversarial perturbations have been applied. We also conducted tests for other scores used in this work. Results of these comparisons for different datasets are presented in tables 18, 19, 20 for non-adaptive scenario and in tables 21, 22 and 23 for adaptive case. The defenses are compared in a pairwise fashion, with each pair yielding a percentage indicating how often one defense statistically outperforms the other under adversarial attack conditions. Intuitively, the Wilcoxon Signed Rank Test results can be understood as a way to rank one defense in terms of how frequently it outperforms others. Defenses that exhibit statistical superiority across most tests (i.e., a higher percentage of tests where they outperform others) can be considered more reliable across a broader range of conditions. The tests also underscore the importance of choosing the right defense for specific types of attacks and image quality metrics.

Furthermore, the application of the Bonferroni correction across these tests further strengthens the reliability of the results, controlling the family-wise error rate due to the large number of comparisons and, therefore, reducing the risk of false positives. The Bonferroni correction applied across these tests is crucial because of the number of comparisons made: 13 attacks, three intensity scales,

Table 13: Comparison of purification defenses. Evaluated metrics are averaged across all images, attacks and quality metrics on KonIQ and KADID datasets.

| Defense | Common | | Non-adaptive case | | | | Adaptive case | | | |
|---|---|---|---|---|---|---|---|---|---|---|
| | $SROCC_{clear}\uparrow$ | Mean Time(ms)$\downarrow$ | $D_{score}^{(D)}\downarrow$ | $R_{score}^{(D)}\uparrow$ | $PSNR\uparrow$ | $SROCC_{adv}\uparrow$ | $D_{score}^{(D)}\downarrow$ | $R_{score}^{(D)}\uparrow$ | $PSNR\uparrow$ | $SROCC_{adv}\uparrow$ |
| W/o Defense | 0.632±0.02 | **0.05±0.01** | 48.61±45.75 | 0.655±0.65 | **39.47±8.07** | 0.464±0.13 | 67.35±44.93 | 0.416±0.57 | **39.37±9.31** | 0.371±0.06 |
| Unsharp | 0.615±0.02 | 1.11±0.57 | 40.42±36.40 | 0.742±0.65 | 28.92±3.48 | 0.426±0.15 | 85.28±55.54 | 0.317±0.52 | 28.61±3.80 | 0.319±0.13 |
| Color Quantization | 0.618±0.03 | 0.09±0.02 | 25.69±24.31 | 0.892±0.63 | 32.26±4.05 | 0.551±0.11 | — | — | — | — |
| FCN | 0.599±0.04 | 0.76±0.15 | 21.12±19.39 | 1.001±0.63 | 20.53±1.10 | 0.519±0.11 | 72.34±45.45 | 0.351±0.56 | 20.11±1.09 | 0.258±0.14 |
| Bilinear Upscale | 0.595±0.02 | 0.23±0.05 | 19.89±17.44 | 0.975±0.67 | 31.32±4.08 | 0.457±0.13 | 44.83±32.01 | 0.568±0.65 | 27.65±3.25 | 0.350±0.10 |
| Gaussian Blur | 0.560±0.02 | 1.41±0.29 | 14.10±12.20 | 1.136±0.67 | 30.41±3.81 | 0.434±0.12 | 49.02±30.86 | 0.453±0.59 | 30.94±4.34 | 0.349±0.12 |
| Median Blur | 0.554±0.02 | 0.16±0.04 | 13.55±10.92 | 1.235±0.87 | 29.40±3.69 | 0.433±0.11 | 53.69±32.14 | 0.428±0.59 | 30.30±4.19 | 0.405±0.12 |
| JPEG | 0.648±0.03 | 579.41±135.20 | 13.37±11.21 | 1.132±0.58 | 31.15±3.75 | 0.626±0.05 | — | — | — | — |
| DiffJPEG | **0.651±0.03** | 12.17±2.00 | 13.26±11.08 | 1.135±0.58 | 31.18±3.76 | **0.629±0.05** | 31.86±20.12 | 0.661±0.57 | 30.20±4.06 | **0.540±0.08** |
| Random Noise | 0.600±0.04 | 12.45±0.80 | 11.06±9.80 | 1.264±0.58 | 25.37±2.04 | 0.571±0.07 | 48.72±32.98 | 0.489±0.56 | 35.31±6.22 | 0.493±0.13 |
| MPRNet | 0.588±0.05 | 72.57±3.46 | 10.57±9.62 | 1.362±0.67 | 29.38±3.88 | 0.554±0.06 | 48.71±30.97 | 0.494±0.65 | 30.88±4.29 | 0.492±0.09 |
| Resize | 0.629±0.03 | 0.27±0.05 | 10.05±7.60 | 1.344±0.55 | 30.37±3.80 | 0.579±0.07 | 59.39±39.28 | 0.458±0.54 | 28.13±3.44 | 0.335±0.11 |
| Crop | 0.596±0.02 | 0.23±0.05 | 10.02±7.51 | 1.484±0.85 | 11.47±0.40 | 0.518±0.08 | **6.86±6.57** | **1.593±0.54** | 11.08±0.20 | 0.389±0.10 |
| Real-ESRGAN | 0.601±0.06 | 7.41±1.80 | 9.34±7.25 | 1.579±0.65 | 29.47±3.45 | 0.541±0.10 | 34.04±21.04 | 0.643±0.61 | 29.61±4.00 | 0.439±0.08 |
| Flip | 0.570±0.03 | 0.07±0.01 | 6.71±4.88 | 1.491±0.54 | 10.53±0.37 | 0.543±0.06 | 72.30±46.54 | 0.371±0.57 | 10.78±0.16 | 0.392±0.13 |
| Rotate | 0.574±0.01 | 3.22±0.38 | 5.75±3.88 | 1.586±0.50 | 11.14±0.35 | 0.512±0.06 | 16.52±8.65 | 0.889±0.40 | 14.21±0.57 | 0.458±0.11 |
| DISCO | 0.621±0.04 | 193.56±9.74 | 3.87±3.46 | **1.760±0.44** | 27.80±2.91 | 0.611±0.05 | 51.97±36.63 | 0.591±0.80 | 28.10±3.35 | 0.454±0.05 |
| DiffPure | 0.537±0.03 | 1432.56±70.26 | **3.76±3.72** | 1.749±0.53 | 27.54±2.94 | 0.513±0.06 | 26.58±19.75 | 0.734±0.64 | 29.08±3.70 | 0.487±0.05 |

Table 14: Comparison of purification defenses by different attack strength. Evaluated metrics are averaged across all images, attacks and quality metrics on KonIQ and KADID dataset.

| | Weak | | | Medium | | | Strong | | |
|---|---|---|---|---|---|---|---|---|---|
| | $D_{score}\downarrow$ | $R_{score}\uparrow$ | $SROCC_{adv}\uparrow$ | $D_{score}\downarrow$ | $R_{score}\uparrow$ | $SROCC_{adv}\uparrow$ | $D_{score}\downarrow$ | $R_{score}\uparrow$ | $SROCC_{adv}\uparrow$ |
| W/o Defense | 33.11 / — | 0.822 / — | 0.522 / — | 45.88 / — | 0.639 / — | 0.470 / — | 66.83 / — | 0.502 / — | 0.401 / — |
| Bilinear Upscale | 14.51 / 28.97 | 0.909 / 0.674 | 0.499 / 0.385 | 18.44 / 38.96 | 0.822 / 0.556 | 0.463 / 0.354 | 24.16 / 53.77 | 0.748 / 0.461 | 0.411 / 0.311 |
| Gaussian Blur | 15.10 / 27.56 | 0.878 / 0.655 | 0.465 / 0.421 | 16.52 / 38.85 | 0.836 / 0.518 | 0.437 / 0.371 | 20.13 / 58.06 | 0.761 / 0.361 | 0.400 / 0.256 |
| Resize | 9.99 / 43.42 | 1.118 / 0.525 | 0.594 / 0.411 | 11.78 / 58.27 | 1.053 / 0.404 | 0.582 / 0.347 | 14.15 / 81.16 | 0.995 / 0.270 | 0.561 / 0.247 |
| MPRNet | 12.67 / 30.77 | 0.993 / 0.602 | 0.561 / 0.530 | 14.13 / 42.40 | 0.946 / 0.490 | 0.556 / 0.513 | 16.87 / 60.60 | 0.885 / 0.360 | 0.546 / 0.432 |
| DiffJPEG | 9.71 / 20.72 | 1.138 / 0.774 | **0.634** / 0.573 | 11.96 / 37.29 | 1.057 / 0.660 | **0.632 / 0.555** | 16.89 / 37.29 | 0.967 / 0.547 | **0.619 / 0.491** |
| JPEG | 9.72 / — | 1.139 / — | 0.631 / — | 11.99 / — | 1.054 / — | 0.629 / — | 16.97 / — | 0.963 / — | 0.616 / — |
| Unsharp | 30.91 / 59.45 | 0.652 / 0.388 | 0.484 / 0.403 | 42.38 / 83.78 | 0.530 / 0.232 | 0.419 / 0.304 | 60.27 / 122.51 | 0.439 / 0.097 | 0.376 / 0.248 |
| Median Blur | 13.02 / 34.98 | 0.981 / 0.574 | 0.462 / 0.468 | 15.05 / 47.84 | 0.915 / 0.427 | 0.434 / 0.424 | 18.09 / 67.41 | 0.856 / 0.282 | 0.404 / 0.322 |
| Real-ESRGAN | 21.48 / 26.51 | 0.689 / 0.621 | 0.564 / 0.476 | 23.15 / 31.73 | 0.665 / 0.566 | 0.548 / 0.461 | 26.67 / 41.73 | 0.627 / 0.467 | 0.509 / 0.380 |
| Color Quantization | 15.46 / — | 1.016 / — | 0.586 / — | 21.08 / — | 0.897 / — | 0.568 / — | 36.81 / — | 0.726 / — | 0.499 / — |
| DISCO | 8.72 / 40.31 | 1.176 / 0.514 | 0.607 / 0.479 | 8.30 / 52.05 | 1.193 / 0.438 | 0.612 / 0.453 | 8.29 / 64.85 | 1.190 / 0.406 | 0.613 / 0.429 |
| DiffPure | 17.92 / 15.97 | 0.780 / 0.869 | 0.501 / 0.502 | 17.33 / 20.49 | 0.800 / 0.766 | 0.515 / 0.492 | 17.57 / 32.05 | 0.797 / 0.593 | 0.521 / 0.467 |
| FCN | 15.38 / 47.23 | 0.973 / 0.471 | 0.566 / 0.344 | 20.74 / 67.76 | 0.885 / 0.318 | 0.529 / 0.248 | 30.85 / 100.34 | 0.771 / 0.194 | 0.463 / 0.182 |
| Random Noise | 14.72 / 26.84 | 0.907 / 0.688 | 0.576 / **0.578** | 14.58 / 42.29 | 0.921 / 0.490 | 0.572 / 0.517 | 17.43 / 72.84 | 0.869 / 0.272 | 0.566 / 0.382 |
| Crop | 11.51 / 18.44 | 1.045 / 0.788 | 0.557 / 0.435 | 13.47 / 18.26 | 0.982 / 0.791 | 0.529 / 0.403 | 16.93 / **19.89** | 0.899 / **0.762** | 0.468 / 0.330 |
| Rotate | 9.20 / **10.65** | 1.153 / **1.066** | 0.533 / 0.543 | 9.98 / 15.27 | 1.110 / **0.886** | 0.520 / 0.474 | 11.21 / 23.80 | 1.072 / 0.683 | 0.485 / 0.355 |
| Flip | **6.38** / 47.67 | **1.318** / 0.508 | 0.557 / 0.480 | **7.62** / 67.31 | **1.255** / 0.347 | 0.553 / 0.395 | 9.81 / 99.51 | 1.166 / 0.182 | 0.520 / 0.300 |

and 7 IQA models, resulting in thousands of pairwise comparisons. The conservative nature of the Bonferroni correction makes the results highlighted as significant and reliable.

The results from these tables intuitively make sense when considering the design and complexity of each defense. Advanced methods like DiffPure and DISCO incorporate more sophisticated techniques to address adversarial perturbations, resulting in higher effectiveness. These methods are tailored to different input image modifications, making them more robust than straightforward approaches. Basic defenses, on the other hand, apply straightforward transformations like blurring or resizing, which may remove some adversarial noise but perform poorly, especially against stronger methods. This highlights the limitations of more straightforward approaches in adversarial scenarios, where sophisticated attacks require more nuanced defenses.

## A.10 EXAMPLES OF ATTACKS AND DEFENSES

We show examples of attacks and defenses with corresponding metric values in Figure 11. We chose the PAQ2PIQ metric and several types of defenses. The central part of the image is zoomed to show the effects of the defenses and attacks.

We show image artifacts of presented defenses in Figure 12. The attacks were performed on MANIQA metric. We demonstrate that most defenses have artifacts. Most of them include: removing details of the original image (DISCO, MPRNet), altering the image content (Real-ESRGAN, DiffPure), reducing the image clarity (DiffPure, blur defenses), changing image color (FCN), and compression artifacts (JPEG/DiffJPEG, Color Quantization).

Table 15: Per-metric comparison of purification defenses in adaptive use case ($SROCC_{clear}/SROCC_{adv}$). Evaluated metrics are averaged across all images and attacks on KonIQ and KADID dataset.

| Defense | Linearity | KonCept | PAQ2PIQ | MANIQA | Meta-IQA | SPAQ | FPR | TOPIQ(NR) | CLIP-IQA+ |
|---|---|---|---|---|---|---|---|---|---|
| W/o Defense | 0.526 / 0.436 | 0.477 / 0.405 | 0.449 / 0.349 | 0.497 / 0.465 | 0.617 / 0.456 | 0.355 / 0.251 | -0.133 / 0.070 | 0.494 / 0.440 | 0.653 / 0.464 |
| Crop | 0.611 / 0.501 | 0.236 / 0.178 | 0.404 / 0.376 | 0.522 / 0.461 | 0.458 / 0.386 | — / — | 0.173 / 0.154 | 0.611 / 0.540 | 0.592 / 0.518 |
| Real-ESRGAN | 0.613 / 0.461 | 0.708 / 0.544 | 0.510 / 0.414 | 0.786 / 0.616 | 0.278 / 0.303 | 0.438 / 0.343 | 0.295 / 0.265 | 0.576 / 0.506 | 0.681 / 0.495 |
| Unsharp | 0.631 / 0.274 | 0.706 / 0.501 | 0.510 / 0.261 | 0.783 / 0.549 | 0.624 / 0.247 | 0.558 / 0.298 | 0.230 / -0.087 | 0.717 / 0.443 | 0.693 / 0.381 |
| DISCO | 0.662 / 0.554 | 0.519 / 0.486 | 0.555 / 0.423 | 0.630 / 0.583 | 0.643 / 0.523 | 0.571 / 0.407 | 0.108 / -0.075 | 0.719 / 0.655 | 0.653 / 0.530 |
| Resize | 0.676 / 0.372 | 0.481 / 0.328 | 0.510 / 0.314 | 0.565 / 0.490 | 0.575 / 0.331 | — / — | 0.209 / -0.109 | 0.738 / 0.563 | 0.555 / 0.390 |
| Bilinear Upscale | 0.677 / 0.522 | 0.433 / 0.289 | 0.540 / 0.414 | 0.481 / 0.407 | 0.429 / 0.280 | 0.569 / 0.304 | 0.193 / -0.044 | 0.644 / 0.561 | 0.607 / 0.417 |
| DiffPure | 0.680 / 0.643 | 0.515 / 0.499 | 0.564 / 0.483 | 0.558 / 0.584 | 0.424 / 0.404 | 0.577 / 0.467 | 0.061 / 0.149 | 0.611 / 0.577 | 0.685 / 0.576 |
| FCN | 0.700 / 0.306 | 0.624 / 0.376 | 0.546 / 0.207 | 0.684 / 0.450 | 0.595 / 0.167 | 0.515 / 0.222 | 0.190 / -0.141 | 0.675 / 0.396 | 0.693 / 0.340 |
| Gaussian Blur | 0.706 / 0.439 | 0.534 / 0.379 | 0.593 / 0.415 | 0.586 / 0.508 | 0.485 / 0.228 | 0.582 / 0.297 | 0.040 / -0.085 | 0.663 / 0.496 | 0.695 / 0.466 |
| Rotate | 0.713 / 0.484 | 0.674 / 0.546 | 0.557 / 0.420 | 0.731 / 0.710 | 0.506 / 0.268 | 0.582 / 0.406 | 0.274 / 0.228 | 0.734 / 0.594 | 0.700 / 0.470 |
| Median Blur | 0.722 / 0.535 | 0.548 / 0.480 | 0.549 / 0.425 | 0.598 / 0.570 | 0.460 / 0.302 | 0.572 / 0.365 | 0.174 / -0.036 | 0.668 / 0.580 | 0.652 / 0.421 |
| Flip | 0.743 / 0.424 | 0.678 / 0.546 | 0.515 / 0.348 | 0.743 / 0.607 | 0.550 / 0.312 | 0.570 / 0.330 | 0.220 / -0.086 | 0.731 / 0.564 | 0.776 / 0.481 |
| Random Noise | 0.745 / 0.580 | 0.683 / 0.592 | 0.572 / 0.429 | 0.732 / 0.654 | 0.611 / 0.444 | 0.574 / 0.432 | 0.238 / 0.172 | 0.707 / 0.601 | 0.712 / 0.531 |
| DiffJPEG | 0.748 / 0.664 | 0.673 / 0.608 | 0.586 / 0.467 | 0.747 / 0.712 | 0.583 / 0.490 | 0.593 / 0.473 | 0.307 / 0.186 | 0.738 / 0.649 | 0.734 / 0.608 |
| MPRNet | 0.755 / 0.619 | 0.665 / 0.600 | 0.589 / 0.456 | 0.653 / 0.621 | 0.574 / 0.455 | 0.569 / 0.394 | 0.157 / 0.089 | 0.751 / 0.667 | 0.712 / 0.525 |

Table 16: Per-metric comparison of purification defenses in adaptive use case ($D_{score}/R_{score}$). Evaluated metrics are averaged across all images and attacks on KonIQ, KADID and NIPS datasets.

| Defense | Linearity | KonCept | PAQ2PIQ | MANIQA | Meta-IQA | SPAQ | FPR | TOPIQ(NR) | CLIP-IQA+ |
|---|---|---|---|---|---|---|---|---|---|
| W/o Defense | 63.66 / 0.31 | 41.80 / 0.47 | 41.61 / 0.49 | 25.61 / 0.62 | 42.62 / 0.39 | 60.63 / 0.46 | 281.18 / -0.28 | 21.57 / 0.70 | 21.91 / 0.68 |
| Unsharp | 71.57 / 0.21 | 60.29 / 0.23 | 56.56 / 0.26 | 36.70 / 0.41 | 48.48 / 0.26 | 84.57 / 0.20 | 376.46 / -0.45 | 26.08 / 0.56 | 24.13 / 0.60 |
| Resize | 66.23 / 0.25 | 21.52 / 0.67 | 26.15 / 0.64 | 19.13 / 0.68 | 45.11 / 0.29 | — / — | 223.38 / -0.20 | 21.74 / 0.64 | 21.28 / 0.67 |
| Flip | 62.20 / 0.31 | 45.41 / 0.38 | 42.11 / 0.44 | 28.26 / 0.57 | 40.12 / 0.43 | 66.02 / 0.36 | 264.75 / -0.15 | 22.44 / 0.67 | 22.63 / 0.65 |
| FCN | 61.60 / 0.29 | 44.00 / 0.41 | 43.61 / 0.41 | 26.67 / 0.54 | 43.05 / 0.37 | 62.42 / 0.37 | 256.72 / -0.27 | 23.11 / 0.63 | 23.31 / 0.60 |
| DISCO | 56.08 / 0.40 | 30.57 / 0.53 | 33.90 / 0.57 | 24.56 / 0.63 | 36.75 / 0.55 | 57.28 / 0.43 | 151.53 / 0.10 | 20.22 / 0.73 | 20.72 / 0.72 |
| Median Blur | 49.19 / 0.36 | 28.51 / 0.51 | 34.85 / 0.49 | 21.77 / 0.58 | 33.52 / 0.50 | 48.64 / 0.42 | 174.94 / -0.12 | 19.21 / 0.67 | 22.31 / 0.63 |
| MPRNet | 43.85 / 0.42 | 27.66 / 0.54 | 32.94 / 0.52 | 21.32 / 0.61 | 33.94 / 0.52 | 41.87 / 0.49 | 151.72 / -0.04 | 16.67 / 0.73 | 20.77 / 0.66 |
| Random Noise | 42.43 / 0.46 | 34.61 / 0.52 | 37.16 / 0.51 | 24.32 / 0.60 | 35.74 / 0.50 | 45.79 / 0.52 | 165.95 / -0.10 | 17.36 / 0.76 | 18.95 / 0.74 |
| Gaussian Blur | 32.26 / 0.56 | 23.82 / 0.62 | 30.48 / 0.53 | 20.24 / 0.64 | 27.93 / 0.59 | 36.96 / 0.55 | 149.81 / -0.12 | 14.57 / 0.84 | 21.51 / 0.65 |
| Bilinear Upscale | 31.03 / 0.57 | 20.91 / 0.68 | 22.89 / 0.68 | 17.66 / 0.73 | 22.14 / 0.71 | 34.90 / 0.58 | 135.84 / 0.03 | 17.10 / 0.77 | 20.30 / 0.66 |
| Real-ESRGAN | 21.71 / 0.73 | 66.37 / 0.08 | 39.43 / 0.44 | 36.74 / 0.32 | 19.19 / 0.76 | 22.89 / 0.75 | 67.99 / 0.20 | 13.21 / 0.89 | 17.36 / 0.72 |
| Rotate | 21.39 / 0.78 | 23.80 / 0.64 | 13.23 / 0.98 | 9.10 / 1.01 | 14.16 / 0.95 | 26.92 / 0.70 | 12.24 / 0.98 | 10.20 / 1.03 | 10.42 / 1.02 |
| DiffJPEG | 20.07 / 0.74 | 22.80 / 0.64 | 22.70 / 0.66 | 18.04 / 0.71 | 17.54 / 0.79 | 25.02 / 0.70 | 84.61 / 0.15 | 10.94 / 0.95 | 15.59 / 0.81 |
| DiffPure | 19.63 / 0.79 | 17.53 / 0.76 | 19.73 / 0.74 | 14.65 / 0.82 | 14.74 / 0.90 | 21.66 / 0.79 | 66.03 / 0.27 | 13.12 / 0.87 | 13.17 / 0.88 |
| Crop | 16.87 / 0.89 | 31.32 / 0.45 | 20.08 / 0.71 | 15.68 / 0.77 | 19.38 / 0.78 | — / — | 17.24 / 0.85 | 8.17 / 1.14 | 10.96 / 0.96 |

Table 17: Comparison of purification defenses by attack type. Evaluated metrics are averaged across all images, attacks and quality metrics for nonadaptive use case on KonIQ and KADID datasets.

| Defense | Restricted WB | | | Unrestricted WB | | | Black-Box | | |
|---|---|---|---|---|---|---|---|---|---|
| | $R_{score} \uparrow$ | $SROCC_{adv} \uparrow$ | $PSNR \uparrow$ | $R_{score} \uparrow$ | $SROCC_{adv} \uparrow$ | $PSNR \uparrow$ | $R_{score} \uparrow$ | $SROCC_{adv} \uparrow$ | $PSNR \uparrow$ |
| W/o Defense | 0.36±0.65 | 0.387±0.29 | 42.12±5.69 | 2.06±1.67 | 0.535±0.26 | 52.31±32.77 | 1.19±0.76 | 0.590±0.31 | 38.87±7.00 |
| Unsharp | 0.30±0.47 | 0.329±0.28 | 30.64±2.20 | 0.89±0.43 | 0.525±0.25 | 26.39±7.31 | 0.87±0.37 | 0.578±0.27 | 28.37±3.96 |
| Real-ESRGAN | 0.62±0.34 | 0.474±0.23 | 31.62±1.37 | 0.70±0.29 | 0.494±0.24 | 25.64±6.49 | 0.75±0.28 | 0.658±0.17 | 28.31±3.86 |
| FCN | 0.62±0.44 | 0.455±0.21 | 20.49±0.33 | 1.07±0.32 | 0.532±0.25 | 18.83±1.87 | 1.21±0.28 | 0.632±0.22 | 21.20±0.46 |
| Color Quantization | 0.63±0.49 | 0.500±0.25 | 33.62±1.63 | 1.08±0.40 | 0.533±0.25 | 27.30±7.77 | 1.21±0.51 | 0.632±0.24 | 32.38±2.45 |
| Bilinear Upscale | 0.68±0.33 | 0.389±0.27 | 33.69±1.94 | 0.92±0.25 | 0.493±0.24 | 27.30±7.99 | 1.01±0.24 | 0.555±0.26 | 29.92±4.70 |
| Gaussian Blur | 0.78±0.24 | 0.374±0.26 | 32.70±1.78 | 0.85±0.22 | 0.443±0.24 | 26.53±7.37 | 0.87±0.27 | 0.522±0.27 | 29.00±4.75 |
| DiffPure | 0.81±0.25 | 0.514±0.19 | 29.46±1.32 | 0.78±0.21 | 0.424±0.22 | 24.39±5.58 | 0.78±0.24 | 0.529±0.22 | 26.26±4.20 |
| Random Noise | 0.82±0.27 | 0.533±0.24 | 26.00±0.69 | 0.83±0.24 | 0.523±0.23 | 22.44±4.03 | 0.99±0.32 | 0.633±0.19 | 25.74±0.65 |
| Crop | 0.83±0.30 | 0.476±0.22 | 11.83±0.13 | 1.05±0.33 | 0.540±0.24 | 11.44±0.50 | 1.19±0.34 | 0.588±0.23 | 11.04±0.74 |
| Median Blur | 0.84±0.28 | 0.385±0.25 | 31.70±1.92 | 0.99±0.31 | 0.455±0.23 | 26.07±7.12 | 1.03±0.28 | 0.503±0.24 | 27.83±5.42 |
| JPEG | 0.91±0.42 | 0.612±0.20 | 33.23±1.72 | 1.14±0.37 | 0.572±0.23 | 26.99±7.34 | 1.23±0.34 | 0.655±0.23 | 30.02±3.37 |
| DiffJPEG | 0.91±0.42 | 0.614±0.20 | 33.29±1.72 | 1.14±0.37 | 0.575±0.23 | 27.02±7.36 | 1.22±0.33 | 0.660±0.22 | 30.03±3.39 |
| MPRNet | 0.94±0.28 | 0.557±0.16 | 32.36±1.61 | 0.98±0.26 | 0.511±0.20 | 26.14±7.20 | 0.99±0.26 | 0.629±0.18 | 27.93±4.92 |
| Resize | 0.94±0.35 | 0.545±0.21 | 32.65±1.78 | 1.08±0.33 | 0.555±0.22 | 26.55±7.39 | 1.23±0.27 | 0.639±0.21 | 28.96±4.73 |
| Rotate | 1.08±0.20 | 0.486±0.20 | 11.41±0.38 | 1.11±0.24 | 0.513±0.23 | 11.09±0.70 | 1.18±0.21 | 0.558±0.24 | 10.87±1.26 |
| Flip | 1.11±0.28 | 0.564±0.23 | 10.85±0.21 | 1.29±0.27 | 0.553±0.27 | 10.53±0.41 | 1.43±0.25 | 0.655±0.21 | 10.21±0.52 |
| DISCO | 1.20±0.22 | 0.594±0.22 | 29.62±1.29 | 1.07±0.27 | 0.544±0.22 | 24.21±5.35 | 1.20±0.25 | 0.651±0.20 | 26.64±3.57 |

Table 18: Wilcoxon tests in nonadaptive use case of purification defenses on KonIQ dataset for $D_{score}^{(D)}$. Each cell value represents the percentage of experiments in which defense denoted in row statistically performs better in terms of $D_{score}^{(D)}$ than the defense in corresponding column with $p_{value}$=0.05.

| Defense | DiffJPEG | Bilinear Upscale | Unsharp | Resize | Rotate | Crop | Median Blur | JPEG | Gaussian Blur | Color Quantization | DiffPure | Random Noise | Flip | MPRNet | FCN | Real-ESRGAN | DISCO | W/o Defense |
|---|---|---|---|---|---|---|---|---|---|---|---|---|---|---|---|---|---|---|
| DiffJPEG | — | 65.24% | 76.92% | 25.36% | 8.55% | 33.05% | 38.46% | 9.69% | 39.32% | 58.12% | 0.85% | 3.70% | 15.95% | 31.05% | 45.87% | 27.07% | 1.42% | 88.03% |
| Bilinear Upscale | 6.84% | — | 62.39% | 13.39% | 6.27% | 9.97% | 5.70% | 4.84% | 0.00% | 28.77% | 0.00% | 1.99% | 8.26% | 21.08% | 16.81% | 7.12% | 0.00% | 83.48% |
| Unsharp | 0.00% | 1.71% | — | 5.13% | 0.28% | 0.85% | 0.85% | 0.00% | 1.14% | 2.28% | 0.00% | 0.00% | 8.83% | 0.00% | 0.00% | 0.00% | 0.00% | 48.15% |
| Resize | 41.88% | 54.42% | 79.20% | — | 5.70% | 40.17% | 50.43% | 40.17% | 45.87% | 57.83% | 8.26% | 20.23% | 11.68% | 42.74% | 47.58% | 35.33% | 0.00% | 81.20% |
| Rotate | 56.41% | 70.94% | 90.60% | 46.15% | — | 49.29% | 63.53% | 62.68% | 72.36% | | 13.96% | 38.75% | 27.92% | 49.00% | 62.11% | 51.00% | 10.26% | 89.17% |
| Crop | 33.05% | 60.40% | 80.91% | 17.38% | 8.55% | — | 44.44% | 36.18% | 42.17% | 58.97% | 10.83% | 29.34% | 11.68% | 37.04% | 49.00% | 41.60% | 10.26% | 86.32% |
| Median Blur | 22.79% | 58.69% | 79.49% | 26.78% | 15.38% | 29.34% | — | 19.09% | 36.18% | 60.97% | 7.98% | 15.10% | 15.95% | 27.35% | 47.29% | 36.18% | 8.83% | 90.03% |
| JPEG | 0.00% | 58.12% | 79.77% | 21.94% | 8.83% | 31.91% | 33.33% | — | 34.47% | 59.54% | 0.85% | 3.99% | 15.10% | 23.65% | 45.58% | 25.64% | 0.85% | 88.89% |
| Gaussian Blur | 19.66% | 79.20% | 75.50% | 25.93% | 11.97% | 23.08% | 26.78% | 16.81% | — | 55.56% | 1.14% | 8.55% | 12.25% | 24.22% | 41.03% | 28.77% | 0.85% | 87.46% |
| Color Quantization | 2.28% | 23.65% | 66.10% | 14.25% | 2.85% | 6.55% | 9.12% | 1.99% | 6.84% | — | 0.28% | 0.00% | 5.41% | 1.14% | 7.69% | 5.70% | 0.57% | 74.64% |
| DiffPure | 80.91% | 85.75% | 92.59% | 68.38% | 47.86% | 60.97% | 80.91% | 84.33% | 75.50% | 86.61% | — | 58.40% | 44.73% | 66.67% | 75.21% | 67.52% | 39.89% | 95.16% |
| Random Noise | 61.54% | 72.93% | 80.06% | 49.29% | 26.50% | 44.16% | 65.53% | 62.39% | 64.67% | 73.22% | 7.98% | — | 28.49% | 52.99% | 60.68% | 46.72% | 9.12% | 82.91% |
| Flip | 35.61% | 51.57% | 66.10% | 34.76% | 8.55% | 36.18% | 45.58% | 34.19% | 43.87% | 54.42% | 9.69% | 22.79% | — | 43.30% | 47.58% | 33.90% | 5.41% | 65.24% |
| MPRNet | 28.49% | 53.28% | 54.13% | 23.93% | 10.26% | 27.64% | 39.89% | 27.07% | 34.47% | 44.16% | 2.85% | 10.26% | 15.38% | — | 39.60% | 25.93% | 2.28% | 62.68% |
| FCN | 9.97% | 42.45% | 82.91% | 18.23% | 4.27% | 7.69% | 21.94% | 9.97% | 24.22% | 39.32% | 1.42% | 5.41% | 4.56% | 24.79% | — | 15.38% | 2.85% | 80.34% |
| Real-ESRGAN | 29.91% | 59.54% | 85.19% | 42.17% | 25.93% | 33.05% | 37.61% | 32.48% | 36.75% | 62.39% | 10.26% | 21.94% | 19.37% | 42.17% | 51.00% | — | 18.80% | 83.19% |
| DISCO | 78.92% | 82.91% | 87.75% | 70.37% | 51.28% | 59.54% | 78.63% | 79.49% | 76.35% | 81.48% | 23.36% | 60.40% | 43.02% | 65.53% | 72.08% | 54.42% | — | 94.59% |
| W/o Defense | 0.00% | 0.00% | 13.68% | 0.00% | 0.28% | 2.85% | 0.00% | 0.00% | 0.00% | 0.00% | 0.00% | 0.00% | 0.28% | 1.99% | 0.00% | 0.00% | 0.00% | — |

Table 19: Wilcoxon tests in nonadaptive use case of purification defenses on KADID dataset for $D_{score}^{(D)}$.

| Defense | DiffJPEG | Bilinear Upscale | Unsharp | Resize | Rotate | Crop | Median Blur | JPEG | Gaussian Blur | Color Quantization | DiffPure | Random Noise | Flip | MPRNet | FCN | Real-ESRGAN | DISCO | W/o Defense |
|---|---|---|---|---|---|---|---|---|---|---|---|---|---|---|---|---|---|---|
| DiffJPEG | — | 54.70% | 70.94% | 22.79% | 10.83% | 30.20% | 22.79% | 14.53% | 26.78% | 69.23% | 2.28% | 9.40% | 14.81% | 3.13% | 52.14% | 5.41% | 0.85% | 78.35% |
| Bilinear Upscale | 6.27% | — | 63.82% | 15.95% | 5.98% | 8.26% | 2.28% | 3.13% | 0.57% | 51.00% | 0.57% | 3.70% | 5.41% | 0.00% | 30.77% | 3.99% | 0.00% | 77.78% |
| Unsharp | 0.00% | 0.85% | — | 1.99% | 0.28% | 0.85% | 0.00% | 0.00% | 0.00% | 5.13% | 0.00% | 0.00% | 0.28% | 0.00% | 0.28% | 0.00% | 0.00% | 54.70% |
| Resize | 43.02% | 58.40% | 75.78% | — | 5.98% | 44.44% | 41.88% | 41.60% | 66.95% | 10.83% | 35.04% | 18.52% | 27.07% | 54.99% | 3.70% | 11.68% | | 81.77% |
| Rotate | 60.97% | 58.40% | 80.91% | 18.80% | — | 45.30% | 58.97% | 65.24% | 56.41% | 83.48% | 19.94% | 57.26% | 38.75% | 42.45% | 56.98% | 15.10% | 11.68% | 88.60% |
| Crop | 38.75% | 58.40% | 80.91% | 18.80% | 9.40% | — | 31.62% | 39.60% | 33.05% | 74.07% | 12.82% | 35.04% | 17.38% | 24.22% | 56.98% | 15.10% | 11.68% | 87.46% |
| Median Blur | 25.36% | 48.72% | 81.48% | 17.66% | 10.54% | 25.93% | — | 25.93% | 29.91% | 75.21% | 3.42% | 21.08% | 13.68% | 9.69% | 56.98% | 8.83% | 5.41% | 87.46% |
| JPEG | 0.57% | 49.86% | 69.52% | 19.94% | 7.98% | 29.06% | 23.08% | — | 24.50% | 66.38% | 2.28% | 7.12% | 14.81% | 1.99% | 50.71% | 5.13% | 0.28% | 79.20% |
| Gaussian Blur | 22.22% | 65.53% | 73.79% | 24.50% | 9.40% | 22.22% | 15.67% | 19.09% | — | 72.08% | 1.99% | 17.95% | 15.67% | 4.56% | 56.13% | 7.41% | 0.28% | 89.17% |
| Color Quantization | 1.71% | 6.27% | 51.57% | 8.55% | 3.13% | 3.70% | 3.70% | 1.99% | 1.42% | — | 0.28% | 0.00% | 3.42% | 0.00% | 11.11% | 1.42% | 0.00% | 66.10% |
| DiffPure | 79.49% | 81.77% | 91.74% | 61.54% | 43.87% | 56.13% | 75.21% | 81.77% | 73.22% | 93.16% | — | 72.65% | 55.84% | 55.27% | 83.19% | 28.21% | 36.47% | 94.30% |
| Random Noise | 35.04% | 59.54% | 69.23% | 29.91% | 17.09% | 37.61% | 37.89% | 35.33% | 43.02% | 74.93% | 6.55% | — | 21.94% | 18.23% | 56.70% | 12.25% | 7.12% | 83.48% |
| Flip | 49.57% | 64.67% | 87.46% | 37.04% | 11.11% | 37.89% | 53.28% | 49.29% | 54.99% | 78.35% | 11.97% | 49.29% | — | 38.46% | 76.92% | 9.40% | 6.27% | 93.73% |
| MPRNet | 45.01% | 62.11% | 81.77% | 31.91% | 16.24% | 37.32% | 51.00% | 45.58% | 51.85% | 83.76% | 6.84% | 30.77% | 27.35% | — | 69.23% | 14.81% | 2.56% | 97.44% |
| FCN | 3.42% | 17.95% | 66.67% | 10.54% | 1.99% | 3.70% | 7.12% | 3.99% | 6.27% | 68.38% | 0.28% | 3.99% | 3.70% | 3.99% | — | 1.71% | 1.14% | 86.04% |
| Real-ESRGAN | 68.09% | 68.95% | 87.18% | 58.12% | 42.74% | 49.00% | 64.10% | 68.38% | 65.24% | 80.34% | 35.04% | 65.24% | 52.71% | 52.99% | 79.49% | — | 32.76% | 92.88% |
| DISCO | 77.21% | 81.48% | 88.60% | 66.10% | 48.15% | 55.27% | 70.94% | 76.92% | 69.23% | 90.60% | 23.93% | 72.36% | 54.70% | 56.70% | 82.05% | 25.36% | — | 96.30% |
| W/o Defense | 0.00% | 0.00% | 11.68% | 0.00% | 0.57% | 0.85% | 0.57% | 0.00% | 0.28% | 0.00% | 0.00% | 1.14% | 0.00% | 4.27% | 0.28% | 0.00% | 0.00% | — |

Table 20: Wilcoxon tests in nonadaptive use case of purification defenses on NIPS dataset for $SSIM$ scores.

| Defense | DiffJPEG | Bilinear Upscale | Unsharp | Resize | Rotate | Crop | Median Blur | JPEG | Gaussian Blur | Color Quantization | DiffPure | Random Noise | Flip | MPRNet | FCN | Real-ESRGAN | DISCO | W/o Defense |
|---|---|---|---|---|---|---|---|---|---|---|---|---|---|---|---|---|---|---|
| DiffJPEG | — | 7.12% | 39.03% | 7.41% | 100.00% | 100.00% | 63.25% | 33.33% | 7.69% | 62.11% | 65.53% | 100.00% | 100.00% | 0.00% | 17.66% | 2.28% | 3.13% | 1.99% |
| Bilinear Upscale | 52.71% | — | 44.16% | 37.04% | 100.00% | 100.00% | 79.77% | 53.28% | 43.59% | 62.11% | 82.05% | 100.00% | 100.00% | 0.00% | 53.85% | 1.42% | 0.00% | 4.84% |
| Unsharp | 29.91% | 10.54% | — | 16.24% | 100.00% | 100.00% | 54.99% | 32.76% | 19.37% | 45.58% | 60.68% | 98.58% | 100.00% | 2.28% | 24.22% | 3.42% | 5.41% | 3.99% |
| Resize | 19.66% | 0.00% | 35.04% | — | 88.89% | 88.89% | 63.53% | 35.04% | 21.94% | 49.29% | 53.56% | 88.89% | 88.89% | 0.00% | 45.30% | 0.57% | 0.00% | 3.99% |
| Rotate | 0.00% | 0.00% | 0.00% | 0.00% | — | 0.00% | 0.00% | 0.00% | 0.00% | 0.00% | 0.00% | 0.00% | 62.39% | 0.00% | 0.00% | 0.00% | 0.00% | 0.00% |
| Crop | 0.00% | 0.00% | 0.00% | 0.00% | 86.32% | — | 0.00% | 0.00% | 0.00% | 0.00% | 0.00% | 0.00% | 100.00% | 0.00% | 0.00% | 0.00% | 0.00% | 0.00% |
| Median Blur | 0.00% | 0.00% | 21.94% | 0.00% | 100.00% | 100.00% | — | 0.00% | 2.28% | 0.00% | 0.00% | 100.00% | 100.00% | 0.00% | 0.00% | 0.00% | 0.00% | 0.00% |
| JPEG | 0.00% | 6.27% | 36.18% | 7.12% | 100.00% | 100.00% | 62.68% | — | 7.69% | 60.40% | 62.39% | 100.00% | 100.00% | 0.00% | 11.40% | 0.85% | 1.99% | 1.71% |
| Gaussian Blur | 18.80% | 0.00% | 37.89% | 8.83% | 100.00% | 100.00% | 65.81% | 27.64% | — | 55.84% | 63.53% | 100.00% | 100.00% | 0.00% | 45.01% | 0.57% | 0.00% | 4.56% |
| Color Quantization | 0.00% | 0.00% | 21.08% | 0.00% | 100.00% | 100.00% | 14.25% | 0.00% | 0.00% | — | 16.52% | 100.00% | 100.00% | 0.00% | 0.00% | 0.00% | 0.00% | 0.00% |
| DiffPure | 0.57% | 0.00% | 23.93% | 0.00% | 100.00% | 100.00% | 17.66% | 0.85% | 0.00% | 17.09% | — | 100.00% | 100.00% | 0.00% | 1.71% | 0.00% | 0.00% | 1.14% |
| Random Noise | 0.00% | 0.00% | 0.00% | 0.00% | 99.43% | 84.62% | 0.00% | 0.00% | 0.00% | 0.00% | 0.00% | — | 100.00% | 0.00% | 0.00% | 0.00% | 0.00% | 0.00% |
| Flip | 0.00% | 0.00% | 0.00% | 0.00% | 0.00% | 0.00% | 0.00% | 0.00% | 0.00% | 0.00% | 0.00% | 0.00% | — | 0.00% | 0.00% | 0.00% | 0.00% | 0.00% |
| MPRNet | 52.71% | 16.24% | 42.74% | 41.31% | 100.00% | 100.00% | 63.53% | 54.13% | 45.87% | 61.54% | 62.96% | 100.00% | 100.00% | — | 54.13% | 9.40% | 2.85% | 5.41% |
| FCN | 1.14% | 4.56% | 29.06% | 4.84% | 100.00% | 100.00% | 51.28% | 1.42% | 5.98% | 36.18% | 42.45% | 100.00% | 100.00% | 0.85% | — | 2.28% | 1.42% | 0.85% |
| Real-ESRGAN | 72.08% | 9.97% | 47.86% | 51.00% | 100.00% | 100.00% | 74.93% | 59.83% | 64.67% | 81.20% | 100.00% | 100.00% | 1.99% | 53.85% | | — | 1.14% | 4.27% |
| DISCO | 71.23% | 54.70% | 62.96% | 63.25% | 100.00% | 100.00% | 81.20% | 72.36% | 72.36% | 88.00% | 95.73% | 100.00% | 100.00% | 0.00% | 45.87% | 67.24% | — | 18.52% |
| W/o Defense | 90.03% | 88.03% | 100.00% | 81.48% | 100.00% | 100.00% | 97.44% | 90.03% | 91.17% | 95.44% | 92.88% | 100.00% | 100.00% | 85.19% | 87.75% | 86.04% | 65.81% | — |

Table 21: Wilcoxon tests in adaptive use case of purification-based and Adversarial Training defenses on KADID dataset and **Linearity**, **Koncept** IQA metrics for $D_{score}^{(D)}$ values.

| Defense | FCN | MPRNet | Median Blur | DISCO | Bilinear Upscale | Flip | DiffPure | Crop | DiffJPEG | Real-ESRGAN | Gaussian Blur | Resize | Unsharp | Rotate | Random Noise | W/o Defense | APGD-LPIPS-2 | APGD-LPIPS-4 | APGD-LPIPS-8 | APGD-SSIM-2 | APGD-SSIM-4 | APGD-SSIM-8 |
|---|---|---|---|---|---|---|---|---|---|---|---|---|---|---|---|---|---|---|---|---|---|---|
| FCN | — | 6.25% | 4.17% | 10.42% | 8.33% | 0.00% | 8.33% | 6.25% | 0.00% | 4.17% | 16.67% | 75.00% | 2.08% | 0.00% | 12.50% | 6.25% | 12.50% | 35.42% | 2.08% | 10.42% | 22.92% | |
| MPRNet | 87.50% | — | 79.17% | 75.00% | 22.92% | 87.50% | 8.33% | 20.83% | 12.50% | 0.00% | 41.67% | 60.42% | 87.50% | 16.67% | 41.67% | 87.50% | 12.50% | 35.42% | 54.17% | 6.25% | 45.83% | 56.25% |
| Median Blur | 77.08% | 10.42% | — | 66.67% | 8.33% | 75.00% | 0.00% | 12.50% | 8.33% | 0.00% | 20.83% | 39.58% | 81.25% | 25.00% | 29.17% | 77.08% | 8.33% | 25.00% | 39.58% | 2.08% | 33.33% | 41.67% |
| DISCO | 68.75% | 16.67% | 66.95% | — | 18.75% | 45.83% | 14.58% | 25.00% | 25.00% | 0.00% | 20.83% | 39.58% | 81.25% | 25.00% | 31.25% | 52.08% | 22.92% | 29.17% | 54.17% | 10.42% | 41.67% | 52.08% |
| Bilinear Upscale | 79.17% | 75.00% | 75.00% | 68.75% | — | 81.25% | 6.25% | 25.00% | 25.00% | 0.00% | 79.17% | 91.67% | 89.58% | 16.67% | 66.67% | 83.33% | 14.58% | 50.00% | 58.33% | 6.25% | 52.08% | 64.58% |
| Flip | 60.42% | 6.25% | 12.50% | 54.17% | 10.42% | — | 2.08% | 12.50% | 10.42% | 0.00% | 12.50% | 25.00% | 87.50% | 10.42% | 4.17% | 8.33% | 12.50% | 14.58% | 45.83% | 6.25% | 18.75% | 37.50% |
| DiffPure | 97.92% | 79.17% | 97.92% | 79.17% | 89.58% | 93.75% | — | 25.00% | 68.75% | 0.00% | 97.92% | 97.92% | 93.75% | 31.25% | 91.67% | 91.67% | 33.33% | 89.58% | 95.83% | 8.33% | 75.00% | 93.75% |
| Crop | 85.42% | 75.00% | 81.25% | 75.00% | 75.00% | 83.33% | 75.00% | — | 72.92% | 0.00% | 79.17% | 77.08% | 79.17% | 79.17% | 83.33% | 79.17% | 75.00% | 79.17% | 72.92% | 77.08% | 70.83% | |
| DiffJPEG | 91.67% | 79.17% | 89.58% | 79.17% | 75.00% | 60.42% | 89.58% | 22.92% | — | 0.00% | 89.58% | 77.08% | 93.75% | 25.00% | 83.33% | 91.67% | 14.58% | 66.67% | 87.50% | 8.33% | 87.50% | 87.50% |
| Real-ESRGAN | 0.00% | 0.00% | 0.00% | 0.00% | 0.00% | 0.00% | 0.00% | 0.00% | 0.00% | — | 0.00% | 0.00% | 0.00% | 0.00% | 0.00% | 0.00% | 0.00% | 0.00% | 0.00% | 0.00% | 0.00% | 0.00% |
| Gaussian Blur | 81.25% | 25.00% | 60.42% | 66.67% | 10.42% | 81.25% | 0.00% | 66.67% | 0.00% | 39.58% | — | 50.00% | 89.58% | 8.33% | 45.83% | 81.25% | 20.83% | 45.83% | 41.67% | 4.17% | 41.67% | 41.67% |
| Resize | 72.92% | 29.17% | 47.92% | 52.08% | 0.00% | 66.67% | 0.00% | 39.58% | 0.00% | 50.00% | 79.17% | — | 91.67% | 2.08% | 41.67% | 47.92% | 4.17% | 41.67% | 41.67% | | | |
| Unsharp | 14.58% | 4.17% | 8.33% | 6.25% | 8.33% | 4.17% | 0.00% | 6.25% | 2.08% | 0.00% | 6.25% | 14.58% | — | 2.08% | 14.58% | 35.42% | 0.00% | 0.00% | 10.42% | 12.50% | | |
| Rotate | 89.58% | 79.17% | 83.33% | 75.00% | 75.00% | 85.42% | 58.33% | 70.83% | 0.00% | 81.25% | 77.08% | 97.92% | 97.92% | — | 79.17% | 29.17% | 83.33% | 77.08% | 8.33% | 77.08% | 77.08% | |
| Random Noise | 89.58% | 33.33% | 66.67% | 66.67% | 14.58% | 85.42% | 2.08% | 14.58% | 8.33% | 31.25% | 47.92% | 93.75% | 10.42% | | — | 91.67% | 8.33% | 12.50% | 45.83% | 6.25% | 35.42% | 45.83% |
| W/o Defense | 54.17% | 6.25% | 8.33% | 6.25% | 8.33% | 17.08% | 0.00% | 6.25% | 2.08% | 0.00% | 33.33% | 37.50% | 56.25% | 33.33% | 47.92% | — | 4.17% | 10.42% | 2.08% | 4.17% | | |
| APGD-LPIPS-2 | 45.83% | 33.33% | 33.33% | 50.00% | 0.00% | 0.00% | 0.00% | 0.00% | 0.00% | 33.33% | 37.50% | 56.25% | 33.33% | 33.33% | 47.92% | 4.17% | — | 0.00% | 35.42% | 4.17% | 0.00% | 4.17% |
| APGD-LPIPS-4 | 79.17% | 35.42% | 64.58% | 60.42% | 39.58% | 75.00% | 4.17% | 16.67% | 18.75% | 43.75% | 45.83% | 81.25% | 12.50% | 52.08% | 77.08% | 0.00% | 87.50% | — | 31.25% | 72.92% | | |
| APGD-LPIPS-8 | 45.83% | 33.33% | 33.33% | 50.00% | 0.00% | 0.00% | 0.00% | 0.00% | 0.00% | 33.33% | 37.50% | 50.00% | 0.00% | 33.33% | 47.92% | 4.17% | 0.00% | 2.08% | — | 0.00% | 2.08% | 4.17% |
| APGD-SSIM-2 | 93.75% | 91.67% | 93.75% | 81.25% | 85.42% | 91.67% | 83.33% | 20.83% | 83.33% | 0.00% | 95.83% | 95.83% | 93.75% | 87.50% | 89.58% | 93.75% | 91.67% | 97.92% | 95.83% | — | 91.67% | 95.83% |
| APGD-SSIM-4 | 68.75% | 41.67% | 50.00% | 45.83% | 41.67% | 58.33% | 6.25% | 14.58% | 27.08% | 0.00% | 45.83% | 47.92% | 52.08% | 33.33% | 47.92% | 47.92% | 4.17% | 47.92% | 4.17% | 12.50% | — | 77.08% |
| APGD-SSIM-8 | 52.08% | 37.50% | 39.58% | 37.50% | 25.00% | 45.83% | 2.08% | 8.33% | 4.17% | 0.00% | 37.50% | 39.58% | 56.25% | 4.17% | 37.50% | 47.92% | 4.17% | 12.50% | 62.50% | 2.08% | 4.17% | — |

Table 22: Wilcoxon tests in adaptive use case of purification-based and Adversarial Training defenses on AGIQA dataset and **Linearity**, **Koncept** IQA metrics for $R_{score}$ values.

| | FCN | MPRNet | Median Blur | DISCO | Bilinear Upscale | Flip | DiffPure | Crop | DiffJPEG | Real-ESRGAN | Gaussian Blur | Resize | Unsharp | Rotate | Random Noise | W/o Defense | APGD-LPIPS-2 | APGD-LPIPS-4 | APGD-LPIPS-8 | APGD-SSIM-2 | APGD-SSIM-4 | APGD-SSIM-8 |
|---|---|---|---|---|---|---|---|---|---|---|---|---|---|---|---|---|---|---|---|---|---|---|
| FCN | — | 10.42% | 12.50% | 50.00% | 16.67% | 39.58% | 18.75% | 22.92% | 4.17% | 16.67% | 14.58% | 47.92% | 87.50% | 6.25% | 10.42% | 39.58% | 2.08% | 4.17% | 41.67% | 0.00% | 10.42% | 25.00% |
| MPRNet | 87.50% | — | 16.67% | 77.08% | 58.33% | 27.08% | 89.58% | 25.00% | 39.58% | 27.08% | 45.83% | 22.92% | 64.58% | 87.50% | 14.58% | 50.00% | 83.33% | 6.25% | 41.67% | 52.08% | 4.17% | 43.75% |
| Median Blur | 83.33% | 16.67% | — | 35.42% | 62.50% | 25.00% | 39.58% | 22.92% | 43.75% | 47.92% | 27.08% | 62.50% | 87.50% | 50.00% | 18.75% | 45.83% | 41.67% | 43.75% | 12.50% | 45.83% | 6.25% | 45.83% |
| DISCO | 43.75% | 39.58% | 35.42% | — | 25.00% | 43.75% | 31.25% | 45.83% | 33.33% | 47.92% | 20.83% | 62.50% | 87.50% | 18.75% | 45.83% | 43.75% | 12.50% | 4.17% | 45.83% | 6.25% | 41.67% | 45.83% |
| Bilinear Upscale | 81.25% | 62.50% | 68.75% | 68.75% | — | 79.17% | 14.58% | 56.25% | 39.58% | 47.92% | 37.50% | 58.33% | 81.25% | 35.42% | 47.92% | 81.25% | 33.33% | 37.50% | 50.00% | 10.42% | 37.50% | 54.17% |
| Flip | 39.58% | 8.33% | 12.50% | 54.17% | 18.75% | — | 18.75% | 25.00% | 4.17% | 16.67% | 18.75% | 47.92% | 87.50% | 25.00% | 4.17% | 41.67% | 2.08% | 4.17% | 41.67% | 0.00% | 10.42% | 31.25% |
| DiffPure | 79.17% | 72.92% | 75.00% | 66.67% | 77.08% | 79.17% | — | 16.67% | 64.58% | 66.67% | 66.67% | 64.58% | 72.92% | 83.33% | 72.92% | 75.00% | 81.25% | 66.67% | 72.92% | 77.08% | 72.92% | 77.08% |
| Crop | 72.92% | 52.08% | 58.33% | 47.92% | 37.50% | 70.83% | 16.67% | — | 47.92% | 43.75% | 45.83% | 77.08% | 47.92% | 58.33% | 72.92% | 33.33% | 58.33% | 72.92% | 18.75% | 60.42% | 68.75% |
| DiffJPEG | 89.58% | 62.50% | 68.75% | 60.42% | 58.33% | 93.75% | 33.33% | 41.67% | — | 60.42% | 62.50% | 66.67% | 89.58% | 29.17% | 66.67% | 87.50% | 18.75% | 58.33% | 83.33% | 10.42% | 62.50% | 81.25% |
| Real-ESRGAN | 70.83% | 47.92% | 47.92% | 43.75% | 43.75% | 66.67% | 14.58% | 18.75% | 31.25% | — | 43.75% | 50.00% | 83.33% | 37.50% | 43.75% | 72.92% | 27.08% | 39.58% | 79.17% | 2.08% | 54.17% | 68.75% |
| Gaussian Blur | 81.25% | 75.00% | 68.75% | 68.75% | 56.25% | 81.25% | 33.33% | 43.75% | 35.42% | 33.33% | — | 68.75% | 83.33% | 29.17% | 64.58% | 81.25% | 20.83% | 43.75% | 77.08% | 8.33% | 43.75% | 79.17% |
| Resize | 47.92% | 35.42% | 37.50% | 54.17% | 35.42% | 50.00% | 16.67% | 45.83% | 33.33% | 39.58% | 27.08% | — | 66.67% | 31.25% | 37.50% | 52.08% | 29.17% | 35.42% | 37.50% | 6.25% | 35.42% | 37.50% |
| Unsharp | 4.17% | 12.50% | 12.50% | 31.25% | 12.50% | 12.50% | 16.67% | 22.92% | 4.17% | 10.42% | 14.58% | 27.08% | — | 6.25% | 10.42% | 0.00% | 2.08% | 4.17% | 31.25% | 0.00% | 4.17% | 10.42% |
| Rotate | 89.58% | 83.33% | 79.17% | 75.00% | 58.33% | 93.75% | 22.92% | 43.75% | 60.42% | 50.00% | 68.75% | 62.50% | 91.67% | — | 12.50% | 83.33% | 75.00% | 81.25% | 64.58% | 4.17% | 75.00% | 81.25% |
| Random Noise | 85.42% | 43.75% | 54.17% | 58.33% | 41.67% | 97.92% | 37.50% | 29.17% | 29.17% | 60.42% | 89.58% | 12.50% | 83.33% | 6.25% | — | 66.67% | 6.25% | 8.33% | 41.67% | 0.00% | 14.58% | 31.25% |
| W/o Defense | 41.67% | 16.67% | 16.67% | 52.08% | 18.75% | 20.83% | 18.75% | 22.92% | 10.42% | 18.75% | 16.67% | 47.92% | 93.75% | 16.67% | 12.50% | — | 87.50% | 8.33% | 41.67% | 0.00% | 14.58% | 31.25% |
| APGD-LPIPS-2 | 93.75% | 81.25% | 85.42% | 70.83% | 60.42% | 93.75% | 27.08% | 54.17% | 81.25% | 58.33% | 64.58% | 95.83% | 62.50% | 89.58% | 89.58% | 87.50% | — | 87.50% | 89.58% | 0.00% | 85.42% | 87.50% |
| APGD-LPIPS-4 | 87.50% | 56.25% | 58.33% | 56.25% | 52.08% | 89.58% | 25.00% | 37.50% | 33.33% | 50.00% | 50.00% | 64.58% | 93.75% | 16.67% | 58.33% | 83.33% | 6.25% | — | 81.25% | 4.17% | 41.67% | 81.25% |
| APGD-LPIPS-8 | 54.17% | 41.67% | 47.92% | 50.00% | 22.92% | 58.33% | 18.75% | 22.92% | 14.58% | 10.42% | 12.50% | 56.25% | 64.58% | 14.58% | 20.83% | 52.08% | 4.17% | 8.33% | — | 0.00% | 4.17% | 14.58% |
| APGD-SSIM-2 | 100.00% | 91.67% | 93.75% | 93.75% | 79.17% | 100.00% | 54.17% | 68.75% | 83.33% | 75.00% | 89.58% | 81.25% | 100.00% | 95.83% | 100.00% | 89.58% | 93.75% | 93.75% | 95.83% | — | 91.67% | 93.75% |
| APGD-SSIM-4 | 75.00% | 52.08% | 56.25% | 56.25% | 50.00% | 72.92% | 25.00% | 33.33% | 27.08% | 50.00% | 64.58% | 91.67% | 16.67% | 50.00% | 68.75% | 8.33% | 18.75% | 81.25% | 6.25% | — | 79.17% |
| APGD-SSIM-8 | 62.50% | 41.67% | 41.67% | 52.08% | 22.92% | 58.33% | 18.75% | 27.08% | 12.50% | 16.67% | 14.58% | 60.42% | 87.50% | 14.58% | 18.75% | 54.17% | 6.25% | 14.58% | 45.83% | 0.00% | 6.25% | — |

Table 23: Wilcoxon tests in adaptive use case of purification-based and Adversarial Training defenses on NIPS dataset and **Linearity**, **Koncept** IQA metrics for $PSNR$ values.

| | FCN | MPRNet | Median Blur | DISCO | Bilinear Upscale | Flip | DiffPure | Crop | DiffJPEG | Real-ESRGAN | Gaussian Blur | Resize | Unsharp | Rotate | Random Noise | W/o Defense | APGD-LPIPS-2 | APGD-LPIPS-4 | APGD-LPIPS-8 | APGD-SSIM-2 | APGD-SSIM-4 | APGD-SSIM-8 |
|---|---|---|---|---|---|---|---|---|---|---|---|---|---|---|---|---|---|---|---|---|---|---|
| FCN | — | 0.00% | 0.00% | 0.00% | 0.00% | 100.00% | 0.00% | 100.00% | 0.00% | 0.00% | 0.00% | 2.08% | 0.00% | 100.00% | 0.00% | 0.00% | 0.00% | 0.00% | 0.00% | 0.00% | 0.00% | 0.00% |
| MPRNet | 97.92% | — | 93.75% | 0.00% | 100.00% | 100.00% | 95.83% | 100.00% | 89.58% | 0.00% | 91.67% | 100.00% | 100.00% | 100.00% | 0.00% | 0.00% | 4.17% | 2.08% | 2.08% | 6.25% | 2.08% | 2.08% |
| Median Blur | 97.92% | 0.00% | — | 0.00% | 100.00% | 100.00% | 2.08% | 100.00% | 2.08% | 0.00% | 0.00% | 100.00% | 100.00% | 100.00% | 0.00% | 0.00% | 4.17% | 2.08% | 2.08% | 4.17% | 2.08% | 2.08% |
| DISCO | 100.00% | 100.00% | 100.00% | — | 100.00% | 100.00% | 100.00% | 100.00% | 100.00% | 0.00% | 100.00% | 100.00% | 100.00% | 100.00% | 25.00% | 14.58% | 20.83% | 16.67% | 16.67% | 22.92% | 20.83% | 16.67% |
| Bilinear Upscale | 93.75% | 0.00% | 0.00% | 0.00% | — | 100.00% | 0.00% | 100.00% | 0.00% | 0.00% | 0.00% | 39.58% | 4.17% | 100.00% | 0.00% | 0.00% | 2.08% | 2.08% | 0.00% | 2.08% | 0.00% | 0.00% |
| Flip | 0.00% | 0.00% | 0.00% | 0.00% | 0.00% | — | 0.00% | 0.00% | 0.00% | 0.00% | 0.00% | 0.00% | 0.00% | 0.00% | 0.00% | 0.00% | 0.00% | 0.00% | 0.00% | 0.00% | 0.00% | 0.00% |
| DiffPure | 97.92% | 0.00% | 93.75% | 0.00% | 100.00% | 100.00% | — | 100.00% | 89.58% | 0.00% | 91.67% | 100.00% | 97.92% | 100.00% | 0.00% | 0.00% | 4.17% | 2.08% | 2.08% | 6.25% | 2.08% | 2.08% |
| Crop | 0.00% | 0.00% | 0.00% | 0.00% | 0.00% | 100.00% | 0.00% | — | 0.00% | 0.00% | 0.00% | 0.00% | 0.00% | 100.00% | 0.00% | 0.00% | 0.00% | 0.00% | 0.00% | 0.00% | 0.00% | 0.00% |
| DiffJPEG | 95.83% | 2.08% | 91.67% | 0.00% | 97.92% | 100.00% | 100.00% | 100.00% | — | 0.00% | 0.00% | 100.00% | 100.00% | 100.00% | 0.00% | 0.00% | 4.17% | 2.08% | 2.08% | 6.25% | 2.08% | 2.08% |
| Real-ESRGAN | 0.00% | 0.00% | 0.00% | 0.00% | 0.00% | 0.00% | 0.00% | 0.00% | 0.00% | — | 0.00% | 0.00% | 0.00% | 0.00% | 0.00% | 0.00% | 0.00% | 0.00% | 0.00% | 0.00% | 0.00% | 0.00% |
| Gaussian Blur | 97.92% | 0.00% | 95.83% | 0.00% | 100.00% | 100.00% | 4.17% | 100.00% | 22.92% | 0.00% | — | 100.00% | 100.00% | 100.00% | 0.00% | 0.00% | 4.17% | 2.08% | 2.08% | 6.25% | 2.08% | 2.08% |
| Resize | 91.67% | 0.00% | 0.00% | 0.00% | 45.83% | 100.00% | 0.00% | 100.00% | 0.00% | 0.00% | 0.00% | — | 6.25% | 100.00% | 0.00% | 0.00% | 2.08% | 0.00% | 0.00% | 2.08% | 0.00% | 0.00% |
| Unsharp | 93.75% | 0.00% | 0.00% | 0.00% | 91.67% | 100.00% | 0.00% | 100.00% | 0.00% | 0.00% | 0.00% | 91.67% | — | 100.00% | 0.00% | 0.00% | 2.08% | 2.08% | 0.00% | 4.17% | 2.08% | 0.00% |
| Rotate | 0.00% | 0.00% | 0.00% | 0.00% | 0.00% | 100.00% | 0.00% | 100.00% | 0.00% | 0.00% | 0.00% | 0.00% | 0.00% | — | 0.00% | 0.00% | 0.00% | 0.00% | 0.00% | 0.00% | 0.00% | 0.00% |
| Random Noise | 97.92% | 97.92% | 100.00% | 75.00% | 100.00% | 100.00% | 100.00% | 100.00% | 97.92% | 0.00% | 97.92% | 100.00% | 100.00% | 100.00% | — | 0.00% | 6.25% | 6.25% | 2.08% | 12.50% | 4.17% | 2.08% |
| W/o Defense | 100.00% | 100.00% | 100.00% | 81.25% | 100.00% | 100.00% | 100.00% | 100.00% | 97.92% | 0.00% | 100.00% | 100.00% | 100.00% | 100.00% | 97.92% | — | 35.42% | 35.42% | 41.67% | 50.00% | 37.50% | 41.67% |
| APGD-LPIPS-2 | 95.83% | 87.50% | 91.67% | 72.92% | 95.83% | 100.00% | 87.50% | 100.00% | 87.50% | 0.00% | 89.58% | 95.83% | 93.75% | 100.00% | 87.50% | 27.08% | — | 47.92% | 50.00% | 31.25% | 35.42% | 50.00% |
| APGD-LPIPS-4 | 95.83% | 89.58% | 93.75% | 75.00% | 97.92% | 100.00% | 89.58% | 100.00% | 87.50% | 0.00% | 89.58% | 93.75% | 93.75% | 100.00% | 87.50% | 27.08% | 39.58% | — | 31.25% | 35.42% | 25.00% |
| APGD-LPIPS-8 | 97.92% | 87.50% | 93.75% | 72.92% | 97.92% | 100.00% | 89.58% | 100.00% | 89.58% | 0.00% | 97.92% | 97.92% | 100.00% | 87.50% | 47.92% | 43.75% | 47.92% | — | 66.67% | 50.00% | 43.75% |
| APGD-SSIM-2 | 95.83% | 87.50% | 89.58% | 91.67% | 100.00% | 100.00% | 87.50% | 100.00% | 87.50% | 0.00% | 89.58% | 91.67% | 100.00% | 85.42% | 45.83% | 29.17% | 50.00% | 20.83% | — | 41.67% | 43.75% |
| APGD-SSIM-4 | 95.83% | 87.50% | 91.67% | 75.00% | 95.83% | 100.00% | 89.58% | 100.00% | 87.50% | 0.00% | 89.58% | 97.92% | 93.75% | 100.00% | 87.50% | 29.17% | 29.17% | 18.75% | 25.00% | 41.67% | — | 31.25% |
| APGD-SSIM-8 | 97.92% | 89.58% | 93.75% | 72.92% | 97.92% | 100.00% | 91.67% | 100.00% | 89.58% | 0.00% | 91.67% | 97.92% | 97.92% | 100.00% | 87.50% | 22.92% | 39.58% | 33.33% | 8.33% | 45.83% | 22.92% | — |

Table 24: Comparison of defenses by defense type. Evaluated metrics are averaged across all images, attacks and quality metrics for nonadaptive/adaptive use cases on KonIQ and KADID datasets.

| Defense | $D_{score}^{(D)} \downarrow$ | $D_{score} \downarrow$ | $R_{score}^{(D)} \uparrow$ | $R_{score} \uparrow$ | $SROCC_{adv} \uparrow$ | $SROCC_{clear} \uparrow$ | $PSNR \uparrow$ |
|---|---|---|---|---|---|---|---|
| Filtering | 21.13 / 27.17 | 20.39 / 22.34 | 0.63 / 0.49 | 0.72 / 0.68 | 0.499 / 0.545 | 0.631 / 0.628 | 19.53 / 20.14 |
| Compression | 21.86 / **15.60** | **18.20** / **11.29** | 0.65 / 0.75 | 0.81 / **0.99** | 0.561 / **0.635** | 0.687 / **0.697** | **19.96** / **20.46** |
| Spatial Transforms | 21.20 / 29.95 | 20.27 / 26.53 | 0.64 / 0.46 | 0.69 / 0.62 | **0.578** / 0.508 | 0.684 / 0.604 | 19.62 / 16.57 |
| Denoising | 17.26 / 19.90 | 26.05 / 25.95 | 0.80 / 0.71 | 0.59 / 0.60 | 0.533 / 0.569 | 0.664 / 0.672 | 19.66 / 20.09 |
| With Randomness | 14.93 / 16.71 | 19.17 / 22.29 | 0.83 / **0.84** | 0.77 / 0.69 | 0.523 / 0.528 | 0.634 / 0.596 | 18.81 / 14.81 |
| Adv. Defenses | **8.15** / 26.86 | 23.14 / 22.35 | **1.11** / 0.50 | 0.63 / 0.69 | 0.474 / 0.538 | 0.583 / 0.626 | 19.09 / 19.16 |
| Adv. Training | — / 22.41 | — / 22.41 | — / 0.68 | — / 0.68 | — / 0.552 | — / 0.667 | — / — |

Table 25: Comparison of purification defenses by dataset. Evaluated metrics are averaged across all images, attacks and quality metrics for nonadaptive/adaptive use cases.

| | KonIQA1K | | | KADID1K | | | AGIQA-3K | | | NIPS | |
|---|---|---|---|---|---|---|---|---|---|---|---|
| | $D_{score} \downarrow$ | $R_{score} \uparrow$ | $SROCC_{clear} \uparrow$ | $D_{score} \downarrow$ | $R_{score} \uparrow$ | $SROCC_{clear} \uparrow$ | $D_{score} \downarrow$ | $R_{score} \uparrow$ | $SROCC_{clear} \uparrow$ | $D_{score} \downarrow$ | $R_{score} \uparrow$ |
| W/o Defense | 51.32 / — | 0.57 / — | **0.778** / — | 45.90 / — | 0.74 / — | 0.487 / — | 55.89 / — | 0.57 / — | 0.586 / — | 47.73 / — | 0.60 / — |
| Unsharp | 47.09 / 92.16 | 0.48 / 0.21 | 0.767 / 0.766 | 41.96 / 84.67 | 0.60 / 0.27 | 0.462 / 0.423 | 38.49 / 102.72 | 0.55 / 0.16 | **0.625** / **0.596** | 45.11 / 85.18 | 0.50 / 0.28 |
| Color Quantization | 24.43 / — | 0.83 / — | 0.760 / — | 24.47 / — | 0.93 / — | 0.475 / — | 24.67 / — | 0.86 / — | 0.546 / — | 25.36 / — | 0.85 / — |
| Bilinear Upscale | 19.66 / 45.22 | 0.82 / 0.52 | 0.679 / 0.587 | 18.42 / 36.11 | 0.84 / 0.61 | 0.512 / 0.420 | 12.59 / 47.36 | 1.02 / 0.50 | 0.542 / 0.432 | 20.83 / 26.88 | 0.68 / 0.68 |
| FCN | 22.93 / 73.59 | 0.84 / 0.31 | 0.733 / 0.746 | 21.72 / 70.03 | 0.92 / 0.34 | 0.465 / 0.391 | 26.29 / 74.13 | 0.79 / 0.26 | 0.541 / 0.548 | 18.38 / 50.78 | 0.91 / 0.46 |
| Gaussian Blur | 16.80 / 42.72 | 0.83 / 0.49 | 0.607 / 0.615 | 17.70 / 40.12 | 0.82 / 0.53 | 0.512 / 0.461 | 16.45 / 50.24 | 0.90 / 0.42 | 0.568 / 0.490 | 15.83 / 36.36 | 0.82 / 0.60 |
| Median Blur | 15.17 / 51.43 | 0.93 / 0.41 | 0.668 / 0.678 | 15.60 / 48.65 | 0.99 / 0.44 | 0.440 / 0.402 | 15.98 / 57.85 | 0.95 / 0.36 | 0.571 / 0.512 | 11.82 / 44.05 | 0.98 / 0.49 |
| Real-ESRGAN | 26.36 / 33.41 | 0.59 / 0.54 | 0.719 / 0.682 | 21.18 / 33.15 | 0.73 / 0.57 | 0.484 / 0.385 | 18.49 / 30.48 | 0.61 / 0.57 | 0.464 / 0.457 | 25.38 / 35.13 | 0.61 / 0.53 |
| JPEG | 13.34 / — | 1.02 / — | 0.767 / — | 12.45 / — | 1.09 / — | 0.530 / — | 12.97 / — | 1.08 / — | 0.593 / — | 10.35 / — | 1.06 / — |
| DiffJPEG | 13.27 / 28.33 | 1.02 / 0.65 | 0.770 / 0.765 | 12.43 / 28.46 | 1.09 / 0.67 | 0.532 / 0.484 | 12.96 / 34.54 | 1.08 / 0.57 | 0.593 / 0.579 | 10.33 / 22.22 | 1.06 / 0.73 |
| Resize | 12.35 / 67.49 | 1.00 / 0.31 | 0.722 / 0.628 | 11.60 / 54.19 | 1.00 / 0.31 | 0.536 / 0.439 | 13.03 / 66.14 | 1.00 / 0.31 | 0.561 / 0.478 | 9.73 / 44.73 | 1.10 / 0.55 |
| MPRNet | 11.35 / 46.26 | 1.02 / 0.47 | 0.697 / 0.699 | 15.64 / 42.56 | 0.91 / 0.51 | **0.569** / **0.499** | 14.68 / 51.79 | 1.00 / 0.41 | 0.504 / 0.501 | 12.43 / 41.16 | 0.98 / 0.52 |
| Crop | 14.38 / 19.30 | 1.02 / 0.47 | 0.740 / 0.575 | 13.56 / 18.29 | 0.92 / 0.79 | 0.452 / 0.310 | 17.05 / 21.35 | 0.90 / 0.79 | 0.576 / 0.409 | 16.26 / 46.33 | 0.97 / **0.90** |
| Random Noise | 16.43 / 47.75 | 0.83 / 0.46 | 0.727 / 0.781 | 14.73 / 46.72 | 0.97 / 0.51 | 0.473 / 0.435 | 14.89 / 52.67 | 0.90 / 0.40 | 0.498 / 0.566 | 16.26 / 46.33 | 0.84 / 0.54 |
| Flip | **8.89** / 74.64 | 1.16 / 0.31 | 0.762 / 0.774 | **7.41** / 67.97 | 1.28 / 0.34 | 0.477 / 0.431 | 9.46 / 81.95 | **1.16** / 0.26 | 0.556 / 0.558 | **7.64** / 54.15 | **1.20** / 0.94 |
| Rotate | 10.29 / 17.72 | 1.09 / **0.84** | 0.696 / 0.749 | 9.97 / **15.34** | **1.13** / **0.91** | 0.452 / 0.447 | 12.08 / 22.70 | 1.05 / 0.74 | 0.549 / 0.560 | 9.37 / **13.97** | 1.10 / **0.94** |
| DISCO | **7.77** / 54.65 | **1.20** / 0.38 | 0.720 / 0.735 | 9.10 / 52.58 | **1.17** / 0.45 | 0.522 / 0.459 | 11.26 / 61.65 | 1.05 / 0.36 | 0.543 / 0.542 | 11.28 / 41.30 | 1.15 / 0.61 |
| DiffPure | 17.53 / 23.15 | 0.76 / 0.73 | 0.550 / 0.554 | 19.29 / 22.54 | 0.77 / 0.77 | 0.522 / 0.477 | 17.67 / 21.91 | 0.83 / **0.76** | 0.463 / 0.436 | 12.35 / 20.82 | 0.89 / 0.78 |

Table 26: Comparison of $SROCC$ and $PLCC$ scores averaged across KonIQ, KADID and AGIQA-3K datasets for purification-based and adversarial training defenses.

| Defense | Common | | Non-adaptive case | | Adaptive case | |
|---|---|---|---|---|---|---|
| | $SROCC_{clear}$ ↑ | $PLCC_{clear}$ ↑ | $PLCC_{adv}$ ↑ | $SROCC_{adv}$ ↑ | $SROCC_{adv}$ ↑ | $PLCC_{adv}$ ↑ |
| W/o Defense | 0.617±0.01 | 0.648±0.02 | 0.484±0.13 | 0.464±0.12 | 0.402±0.08 | 0.432±0.08 |
| Unsharp | 0.604±0.02 | 0.631±0.02 | 0.452±0.14 | 0.433±0.14 | 0.345±0.12 | 0.366±0.12 |
| Color Quantization | 0.594±0.02 | 0.616±0.02 | 0.574±0.09 | 0.542±0.09 | — | — |
| FCN | 0.580±0.02 | 0.591±0.01 | 0.522±0.10 | 0.498±0.10 | 0.282±0.13 | 0.299±0.12 |
| Bilinear Upscale | 0.577±0.02 | 0.614±0.03 | 0.499±0.12 | 0.468±0.11 | 0.347±0.09 | 0.376±0.09 |
| Gaussian Blur | 0.543±0.03 | 0.572±0.03 | 0.450±0.13 | 0.426±0.12 | 0.360±0.11 | 0.390±0.10 |
| Median Blur | 0.546±0.02 | 0.579±0.02 | 0.458±0.11 | 0.430±0.11 | 0.412±0.11 | 0.444±0.11 |
| JPEG | 0.630±0.02 | 0.655±0.02 | 0.637±0.05 | 0.610±0.04 | — | — |
| DiffJPEG | **0.632±0.02** | **0.658±0.02** | **0.639±0.04** | **0.613±0.04** | **0.548±0.07** | **0.584±0.06** |
| MPRNet | 0.560±0.03 | 0.595±0.04 | 0.570±0.06 | 0.533±0.05 | 0.483±0.08 | 0.513±0.08 |
| Crop | 0.589±0.01 | 0.624±0.02 | 0.553±0.08 | 0.518±0.07 | 0.381±0.10 | 0.389±0.09 |
| Random Noise | 0.566±0.03 | 0.593±0.04 | 0.573±0.05 | 0.547±0.05 | 0.501±0.12 | 0.534±0.11 |
| Resize | 0.606±0.02 | 0.640±0.03 | 0.588±0.07 | 0.561±0.06 | 0.335±0.10 | 0.370±0.10 |
| Real-ESRGAN | 0.570±0.05 | 0.585±0.04 | 0.537±0.08 | 0.523±0.09 | 0.435±0.08 | 0.467±0.07 |
| Flip | 0.598±0.01 | 0.631±0.02 | 0.597±0.05 | 0.564±0.05 | 0.403±0.12 | 0.423±0.11 |
| Rotate | 0.566±0.01 | 0.595±0.02 | 0.530±0.07 | 0.511±0.06 | 0.459±0.10 | 0.488±0.09 |
| DISCO | 0.595±0.02 | 0.626±0.03 | 0.617±0.05 | 0.589±0.03 | 0.466±0.05 | 0.494±0.05 |
| DiffPure | 0.512±0.04 | 0.547±0.04 | 0.531±0.06 | 0.497±0.06 | 0.472±0.04 | 0.511±0.04 |
| APGD-LPIPS-2 | 0.642±0.00 | — | — | — | 0.510±0.09 | 0.449±0.12 |
| APGD-LPIPS-4 | 0.669±0.00 | — | — | — | 0.485±0.14 | 0.475±0.16 |
| APGD-LPIPS-8 | 0.663±0.00 | — | — | — | 0.461±0.11 | 0.420±0.12 |
| APGD-SSIM-2 | 0.620±0.00 | — | — | — | 0.586±0.02 | 0.504±0.07 |
| APGD-SSIM-4 | 0.670±0.00 | — | — | — | 0.445±0.14 | 0.428±0.17 |
| APGD-SSIM-8 | 0.675±0.00 | — | — | — | 0.504±0.10 | 0.509±0.12 |

Table 27: Comparison of guarantees and computational complexity of certified defenses. For classification-based methods (C) we measured certified radius and number of abstains, for regression-based methods (R) – certified relative delta.

| Defense | $Cert.R$ ↑ for **C** $Cert.RD$ ↑ for **R** | $Abst.$ ↓,% | $Time.$ ↓, sec |
|---|---|---|---|
| Random. Smoothing (RS) (**C**) | **0.20±0.03** / 0.16±0.04 | **4.68±2.85** / 10.61±5.51 | 36.70±18.68 |
| Denoised RS (**C**) | 0.19±0.03 / 0.16±0.04 | 6.30±3.42 / 7.43±4.51 | 44.43±20.97 |
| Diffusion DRS (**C**) | 0.18±0.02 / 0.17±0.03 | 10.55±5.50 / 8.33±5.47 | 71.85±18.94 |
| DensePure (**C**) | 0.17±0.02 / 0.16±0.02 | 7.91±3.95 / 9.21±6.81 | 116.37±19.04 |
| Median Smoothing (MS) (**R**) | **1.47±0.42** / 2.25±0.79 | - | **26.08±17.05** |
| Denoised MS (**R**) | 1.41±0.34 / 1.88±0.49 | - | 29.35±17.06 |

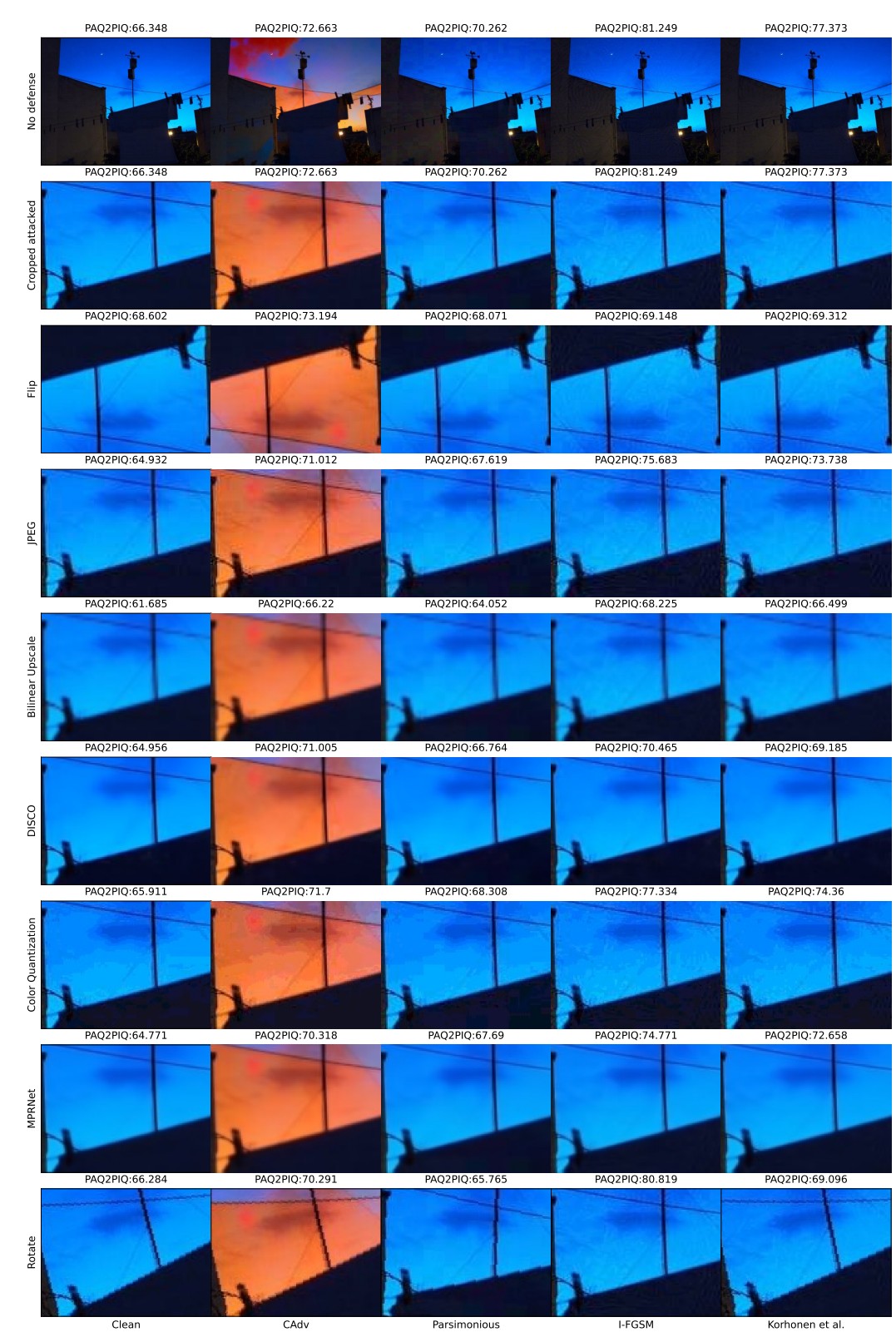

Figure 11: Examples of attacks and defenses on PAQ2PIQ metric. The central part of the image is zoomed to show defense effects.

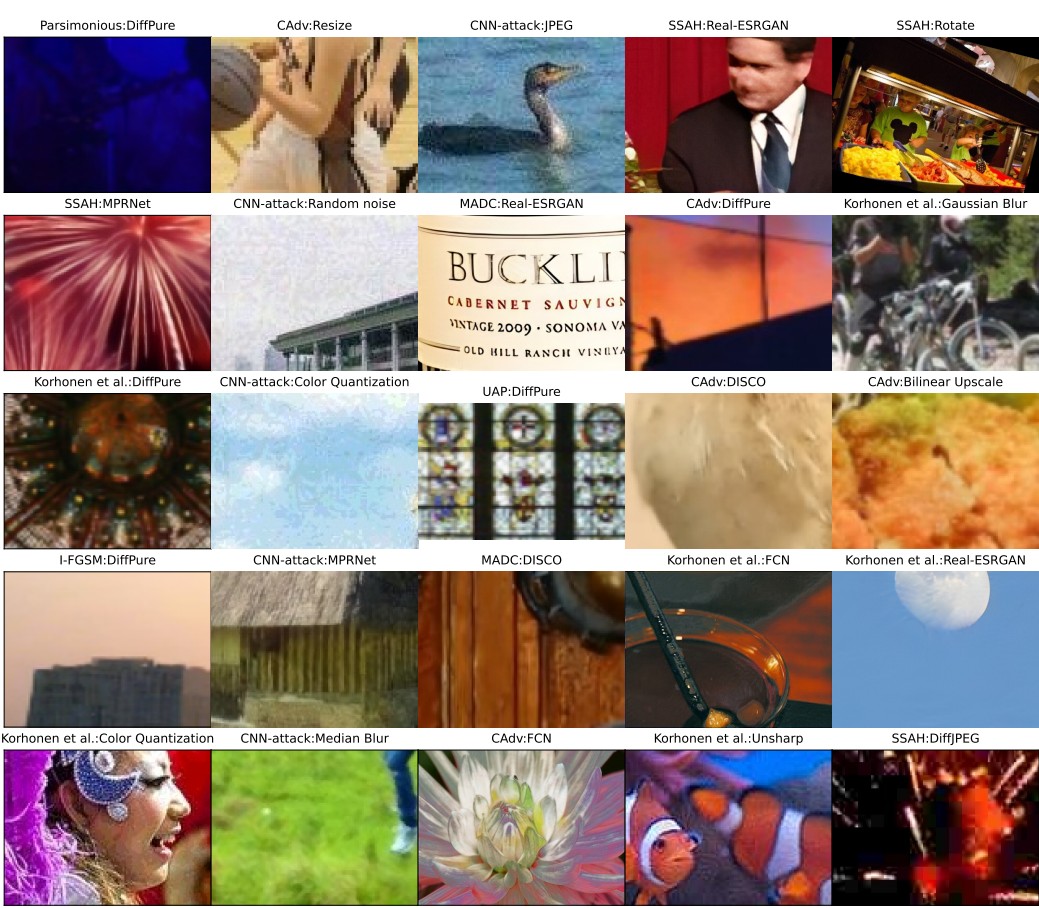

Figure 12: Examples of artifacts caused by various defenses when MANIQA metric is attacked. We selectively zoom in on key parts of the images to highlight the details.

