# OpenReview forum: "Guardians of Image Quality: Benchmarking Defenses Against Adversarial Attacks on Image-Quality Metrics"
_ICLR.cc/2025/Conference — Submitted to ICLR 2025_

### Official Review · Reviewer_vZ76 · 2024-10-31

**Soundness:** 2
**Presentation:** 3
**Contribution:** 2
**Rating:** 5
**Confidence:** 5

**Summary:**

This paper proposes an empirical investigation of the effectiveness of various defense techniques against adversarial attacks on image quality metrics.

**Strengths:**

1. The paper is decently written.
2. The problem under investigation is of practice importance as well described in the Introduction.
3. This empirical study is comprehensive.

**Weaknesses:**

1. A primary technical concern is label preservation (quality preservation in our context) amidst adversarial perturbations. This research evaluates various adversarial attacks, including those like MADC by Wang and Simoncelli, which may not preserve image quality after attacks. In such instances, the model is expected to provide a different quality prediction for the manipulated image, necessitating ground-truth quality assessment through human evaluation.

2. From a conceptual perspective, the reviewer wonders how to understand certified defenses that involve voting and medianing. In classification, it is not hard to comprehend that the robustness is gained through prediction ensembling, which is again under the assumption of label preservation (see Eq. (4)). However, the quality preservation assumption is clearly not true for quality assessment. For example, consider a test image with some Gaussian blur, to perform random smoothing, we shall add Gaussian noise to it according to Eq. (4). Then, the final score is the average quality estimates of a Gaussian blurred image and a Gaussian blurred and noised image (which may be of different quality), which makes less sense to the reviewer.

3. The observed effectiveness of the adversarial attacks (evidenced by an SRCC decrease from 0.611 to 0.477 in Table 3) appears inconsistent with prior research such as [Zhang et al., 2022a] (which reduces to random guessing). Given the limited success of these attacks, interpreting the defense results with similar SRCC values (ranging between 0.5 and 0.6) becomes challenging.

4. Recent NR-IQA models that integrate visual and language components have not been evaluated in this study.

5. The focus of this empirical study aligns more closely with image processing journals rather than a machine learning conference like ICLR, given that no new theories and algorithms are developed.

**Questions:**

The authors should work more on Points 1, 2, and 3 in an attempt to raise the reviewer's rating.

---

> ### Author Response · Authors · 2024-11-18
>
> We would like to thank the reviewer for valuable comments and thoughtful concerns. We address them below:
> ### Concerns
> 1. Indeed, adversarial perturbations can affect image quality, so we carefully selected perturbation budgets to ensure that any changes in the quality of attacked images remain barely noticeable to human observers. To illustrate this, we provide examples of attacked, and defended images at the end of the appendix (the images in visualizations are cropped, so perturbations can be seen in them).
> The goal of purification defenses, as outlined in Equation 2, is to restore the attacked image and IQA metric score to the original image, whose MOS is known. Without subjective assessments, the exact quality of attacked and defended images is unknown; but attacks introduce additional noise to the source image and, by nature, cannot improve perceptual quality. Thus, we assume that the quality of defended images after two transformations (attack and defense) is slightly lower than the original. Since our attacks aim to increase metric scores, we consider the original score a reasonable proxy for the defended image's target quality. For researchers developing new defenses, having the original score as a reference can offer a practical baseline for comparing defense effectiveness, even in the absence of subjective evaluations.
> 2. Thank you for addressing this issue. This is a common challenge for all defenses adapted from classification to regression tasks. Since classifiers focus primarily on semantic features, defenses designed for them typically modify high-frequency information to remove adversarial perturbations.
> In our study, adversarial attacks are increasing the attacked IQA score. Therefore, the smoothed version (e.g., using a voting mechanism or the median of scores for blurred images) should align with or remain lower than the original score for the unperturbed image. Our goal is to achieve a defended quality metric that maintains strong SROCC and PLCC performance while demonstrating high robustness against attacks. When applying median smoothing, a denoising step is applied to preserve SROCC and PLCC. As shown in Table 4, the $SROCC_{clear}$ for DMS (Denoised Median Smoothing) is the highest among all evaluated methods. To sum up, certified defenses for IQA metrics can be understood as methods that aim primarily to put back (lower) the attacked IQA score. Without denoising, these defenses will significantly reduce performance (SRCC), as our experiments proved.
> 3. We apologize for making it unclear. The attack we applied from Zhang et al. has a slightly modified objective compared to its original paper. Zhang et al. maximized the absolute difference between IQA scores of original and adversarial images (making the score higher or lower), while our adaptation focuses on increasing scores. This difference in objective accounts for the discrepancy in correlation coefficients on attacked images. In our formulation, the attacked score is consistently higher than the original, which results in a smaller drop in SRCC coefficients than reported in the original paper.
> Our approach prioritizes score increases because of their relevance in practical applications, such as inflating quality scores in streaming environments and manipulating benchmarks. We will clarify these differences in the revised paper to explain more clearly how the choice of attack objective impacts observed effectiveness.
> 4. Thank you for the suggestion. In fact, we evaluated two recent metrics published in 2024: CLIP-IQA+ and TOPIQ. CLIP-IQA+, in particular, integrates both visual and language features for quality assessment, aligning with the latest advancements in NR-IQA. We will clarify this in the paper to ensure it addresses this point directly. We agree that NR-IQA models incorporating visual and language components are a valuable addition to the field. Additionally, we plan to expand our evaluation to include more NR-IQA models, incorporating the models you suggested in future versions of the study.
> 5. While our study is empirical, it addresses critical challenges in machine learning. It contributes to advancing the understanding and development of robust models, particularly in the neural network-based Image Quality Assessment (IQA) metrics domain. Benchmarks are a cornerstone of ML research, frequently published at top A* conferences, including ICLR (e.g. [1], [2], and many more). Our work introduces the first comprehensive benchmark for evaluating adversarial defenses on IQA metrics, including 1) A publicly available dataset of adversarial images. 2) A leaderboard for reproducibility and comparison of defense methods.
> This benchmark provides a standardized framework for testing and improving the robustness of ML-based IQA metrics.
>
> [1] Q-Bench (ICLR 2024 spotlight) https://openreview.net/forum?id=0V5TVt9bk0
>
> [2] ViLMA (ICLR 2024) https://openreview.net/forum?id=liuqDwmbQJ

---

> > ### Comment · Reviewer_vZ76 · 2024-11-29
> > **Additional comments**
> >
> > Thank the authors for responding to the comments; the reviewer highly appreciates it.
> >
> > Regarding Point 1: The reviewer appears to have overlooked that the attack is solely directed toward increasing the predicted quality score (i.e., larger scores indicate better-predicted quality). Under this assumption, the attack becomes a specific case of Zhang et al., where $\lambda = \infty$, and can be made unbounded and therefore trivial: one could simply compute $x^\star =\arg\max_x q(x)$. The resulting image would likely contain additional visible noise, but the predicted quality score would still be maximized. All NR-IQA models would be inherently vulnerable to this type of attack.
> >
> > Regarding Point 2: The reviewer suggests that the key distinction does not lie between classification and regression. Instead, it still boils down to whether the added perturbation preserves the label or not. If the perturbation is label-preserving, ensemble methods (e.g., averaging or smoothing predictions) can be applied to enhance robustness against adversarial attacks.
> >
> > Regarding Point 3: Thank the authors for the clarification. These details should be incorporated into the main text for better clarity.
> >
> > In light of the additional comments, the reviewer has decided to maintain the current rating.

---

> > > ### Author Response · Authors · 2024-12-01
> > > **Additional clarification**
> > >
> > > __Regarding point 1__: We respectfully disagree with the reviewer’s interpretation. In our formulation, the attack is designed to meet two key criteria: (1) maximize the predicted IQA score, and (2) ensure that perturbations remain imperceptible to human observers. Trivial unbounded attacks fail to meet the second criterion, as they generate visible distortions, making them impractical for real-world applications where imperceptibility is essential. To ensure imperceptibility, we employ a parameter tuning methodology, detailed in Figure 2. To ensure that attacks are aiming to increase IQA scores, we made some minor modifications to the objective function. For example:
> > > * Original objective function for Zhang et al. attack: $L = - D(x_{clean}, x_{adv}) + \lambda * (f(x_{adv}) - f(x_{clean}))^2 $
> > > * Our slightly modified objective function: $L = - D(x_{clean}, x_{adv}) + \lambda * f(x_{adv}) $
> > >
> > > where $D$ — is a proxy full-reference IQA metric (e.g., LPIPS, DISTS, or SSIM in our experiments), and $f(\cdot)$ is the NR-IQA model being attacked. The original function maximized the absolute difference between the IQA scores of clean and adversarial images, while our modification focuses directly on increasing $f(x_{adv})$, balancing this increase with maintaining imperceptibility.
> > > Our approach addresses the reviewer’s concern about trivial unbounded attacks, ensuring their practical relevance while highlighting the vulnerability of NR-IQA models to adversarial attacks.
> > >
> > > __Regarding point 2__: Yes, this is true; adding noise to an image can impact its quality. However, we can view the process of adding noise followed by a denoising step as a quality-preserving transformation in relation to image quality metrics. Importantly, all theoretical guarantees that apply to classification also hold for quality metrics since these guarantees are based solely on the noise model and the score aggregation process. We do not alter these components of the algorithm(for the RS, DRS, DDRS, and DP methods, we discretize the regression metric values into N classes before applying the methods; for the MS and DMS methods, we apply them directly without any modifications). The main issue is that the performance of these models, in terms of accuracy of quality scores or correlation with subjective scores, may degrade more than in classification or detection tasks. Nevertheless, our experimental results demonstrate that the decline in accuracy is relatively minor and comparable to that observed with other defense mechanisms. Therefore, we believe that certified defenses based on smoothing represent a promising direction for image quality assessment.
> > >
> > > We hope this addresses your concerns, and we remain open to additional feedback

---

### Official Review · Reviewer_sRtC · 2024-11-01

**Soundness:** 3
**Presentation:** 2
**Contribution:** 2
**Rating:** 5
**Confidence:** 5

**Summary:**

This paper systematically evaluates the effectiveness of various defense strategies for NR-IQA models, including adversarial purification, adversarial training, and certified robustness. It also examines these defenses under both adaptive and non-adaptive attack scenarios. Overall, the experiments in this paper are thorough and comprehensive, but the paper's readability could still be enhanced.

**Strengths:**

1. The research topic of this paper is interesting and promising.
2. The experiments in this paper are comprehensive and detailed.

**Weaknesses:**

1. Most of the strategies are directly borrowed from classification tasks, with no new approaches tailored to the specific characteristics of IQA tasks.
2. In addition to PSNR and SSIM, incorporating additional quantitative indicators like L_2 and L_\infty would provide a more intuitive understanding of the differences between the original image and the purified image.
3. The article is less readable, and many tables contain abbreviations that are not defined.

**Questions:**

1. The definition of attack in Equation 1 focuses solely on increasing the model's score. However, if an image already possesses the highest quality, what is the purpose of the attack on it? Why was the idea of decreasing the score of high-quality images/videos not considered?
2. Currently, the attack methods employed are those typically used in classification problems. It would be beneficial to consider incorporating some of the attack strategies for IQA that have been proposed in recent years.
3. Table 3 shows that many adversarial purification defenses exhibit strong defensive effects. However, these methods should be analyzed more thoroughly. For instance, purification techniques that modify the entire image, such as color quantization and median blur, should include more detailed indicators (L_2 and L_\infty) to better reflect the extent of image modification.
4. Most of the analysis in this paper primarily describes the data presented in the tables. It would be beneficial to include an in-depth analysis of the characteristics and connections among these various types of defense strategies, providing great guidance for future research.
5. The abbreviations in the table should be added with full spelling in the caption to help readers understand and prevent misunderstandings.
6. Equation 2 misses the variable x’.

---

> ### Author Response · Authors · 2024-11-18
>
> We sincerely thank the reviewers for their thorough evaluation and constructive feedback, which have helped us improve the quality and clarity of our manuscript. We address your concerns below:
> ### Weaknesses
> 1. Indeed, IQA metrics are also neural networks as classification methods. Thus, the existing defenses with minimal changes may apply to IQA metrics. However, we compare them in a different setting, i.e. we raise the importance of measuring discrepancy and quality of images, which makes the IQA metrics more difficult to defend. Currently, few defenses that satisfy these criteria exist. For example, we included FCN filter [1] introduced by Guschin et al. as a purification approach. Currently, we are computing the Gradient Norm Regularization defense [2] and will update the paper with its results in a few days.
> 2. Thank you for your suggestion. Following your request, we have updated Table 3 in the revised paper to include $L_{inf}$ and MSE indicators as additional measures of quality (line 324). According to our findings, these metrics align closely with PSNR and SSIM, confirming consistent trends across different evaluation methods.
> 3. Thank you for raising the concern regarding the readability. We have addressed this issue by adding definitions for all abbreviations in Table 4 (line 394) and providing a description of the appendix structure to help readers navigate the supplementary material more easily (line 760).
> ### Questions
> 1. From an adversarial perspective, decreasing IQA scores can be achieved similarly to increasing them, requiring only a sign change in the optimization step of each attack. We focused on increasing scores, as this has more practical applications in real-world misuse cases: for instance, IQA metrics are used in video streaming, where artificially raising perceived quality scores can lead to increased bitrate after transcoding, creating stress for network resources. Additionally, inflating IQA scores can be exploited to mislead benchmarks that influence project investments, especially in resource-intensive areas. Modern codecs even include optimization modes that target specific metrics (e.g., Google’s libaom encoder with its --tune-vmaf option). Given these prevalent scenarios, as mentioned in the problem formulation, we prioritized score increases in our approach.
> 2. In our study, we included 4 recent attacks designed for IQA: FACPA (Shumitskaya et al. "Fast adversarial cnn-based perturbation attack on no-reference image- and video-quality metrics"), Optimized-UAP (Shumitskaya et al. Towards adversarial robustness verification of no-reference image-and video-quality metrics), Korhonen et al. (J. Korhonen and J. You, "Adversarial attacks against blind image quality assessment models"), and three variants of Zhang et al. (Zhang et al., "Perceptual attacks of no-reference image quality models with human-in-the-loop"). These were selected to represent a range of techniques relevant to IQA model vulnerabilities. We acknowledge that new IQA-specific attacks are emerging, and we plan to expand our benchmark in the future to include additional methods as they are developed. Adversarial attacks on IQA models represent a relatively new and growing area of research, with only a limited number of attacks tailored specifically for IQA tasks.
> 3. As mentioned in our comment to weakness #2 above, we added results and conclusions for $L_{inf}$ and $L_2$ (MSE), line 324 in the revised paper.
> 4. We added more analysis regarding differences in performances on KonIQ1k and other datasets (line 482) and FPR poor results (line 474).
> 5. We updated the caption of Table 4 in the revised version of the paper (line 394)
> 6. Thank you for pointing to this issue, we added the missing variable (line 146).
>
> [1] Gushchin et al., “Adversarial purification for no-reference image-quality metrics: applicability study and new methods,” 2024
>
> [2] Defense Against Adversarial Attacks on No-Reference Image Quality Models with Gradient Norm Regularization, IEEE Conference on Computer Vision and Pattern Recognition (CVPR), Seattle, WA, USA, June 2024

---

> > ### Comment · Reviewer_sRtC · 2024-11-25
> > **Core concerns are not addressed**
> >
> > Thank the authors for responses. My most concerns are not addressed well. Why can decreasing IQA scores be achieved similarly to increasing them, requiring only a sign change in the optimization step of each attack? I think it is not straightforward. For the definition of adversarial attack, only increasing the score of low-quality video is only one half, missing another half.

---

> > > ### Author Response · Authors · 2024-11-27
> > > **Addressing decreasing IQA scores and comparing it with increasing**
> > >
> > > We apologize for the lack of clarity in our original explanation.
> > > Generally, to increase the IQA score, white-box attacks perform gradient ascending with some modification (spatial maps for Korhonen et al., proxy IQA metric for Zhang et al., etc.). To adapt these attacks to decrease IQA scores, the optimization step is reformulated to perform gradient descent. For instance, the IFGSM update step changes as follows:
> > > * Increase IQA score: $x_t = x_{t-1} - \alpha * sign(\nabla(Loss))$
> > > * Decrease IQA  score: $x_t = x_{t-1} + \alpha * sign(\nabla(Loss))$
> > >
> > > This adjustment effectively decreases IQA scores using methodologies similar to those used to increase them.
> > >
> > > To evaluate the effectiveness of these approaches, we attacked 1,000 images from the KADID dataset using both settings (increasing and decreasing IQA scores). The table below summarizes the results, showing SSIM scores and $D_{score}$ percentages for decreasing and increasing IQA values across various attacks and models. The first value in each column corresponds to decreasing IQA scores, while values in parentheses indicate increasing scores.
> > >
> > > | Attack      | IQA model | SSIM | $D_{score}$
> > > | -----------   | ----------- | ----------- | ----------- |
> > > IFGSM|CLIP-IQA+|0.977(0.977)|-82.4%(66.74%)|
> > > Korhonen et al.|CLIP-IQA+|0.957(0.957)|-80.3%(64.18%)|
> > > Zhang et al.|CLIP-IQA+|0.949(0.949)|-87.7%(65.69%)|
> > > IFGSM|Koncept|0.960(0.953)|-126%(141.5%)|
> > > Korhonen et al.|Koncept|0.960(0.960)|-112%(104.4%)|
> > > Zhang et al.|Koncept|0.949(0.948)|-118%(111.5%)|
> > > IFGSM|MANIQA|0.961(0.952)|-98.2%(109.6%)|
> > > Korhonen et al.|MANIQA|0.965(0.963)|-96.7%(94.81%)|
> > > Zhang et al.|MANIQA|0.952(0.952)|-97.0%(104.8%)|
> > > IFGSM|SPAQ|0.965(0.962)|-72.4%(82.9%)|
> > > Korhonen et al.|SPAQ|0.955(0.951)|-63.2%(78.97%)|
> > > Zhang et al.|SPAQ|0.944(0.941)|-68.7%(89.2%)|
> > > IFGSM|TOPIQ|0.974(0.972)|-103.%(108.8%)|
> > > Korhonen et al.|TOPIQ|0.958(0.958)|-100%(91.72%)|
> > > Zhang et al.|TOPIQ|0.949(0.952)|-107%(92.9%)|
> > >
> > > The results demonstrate that the success rates for increasing and decreasing IQA scores are largely comparable across models and attacks. SSIM values are almost identical and $D_{score}$ are aligned, highlighting that attacks work very similarly in both cases. Given the comparable effectiveness and the limited practical applications of decreasing IQA scores, we decided to concentrate on attacks only increasing IQA scores, as was done in other studies [1, 2].
> > >
> > > [1] Shumitskaya et al., “IOI: Invisible One-Iteration Adversarial Attack on No-Reference Image-and Video-Quality Metrics,” In Proceedings of the 41st International Conference on Machine Learning (ICML), pages 45329–45352. PMLR, 2024
> > >
> > > [2] Antsiferova et al., “Comparing the robustness of modern no-reference image- and video-quality metrics to adversarial attacks,” in Proceedings of the 2024 AAAI Conference on Artificial Intelligence, doi:10.1609/aaai.v38i2.27827

---

> ### Comment · Reviewer_sRtC · 2024-12-02
> **No new insight to adversarial defense**
>
> Overall, this work does not provide new insight for adversarial defense on IQA. It synthesizes existing adversarial attacks on IQA and ensembles some testing results, lacking enough novelty to reach the conference bar. I keep my rating.

---

### Official Review · Reviewer_RZ8e · 2024-11-03

**Soundness:** 3
**Presentation:** 1
**Contribution:** 2
**Rating:** 6
**Confidence:** 3

**Summary:**

The paper attempts to set up benchmarking for adversarial attacks and defenses in the context of No-Reference Image Quality Assessment algorithms. The coverage of the work seems to be good—29 defense strategies, 14 attack methods, and 9 IQA methods. Lots of experiments (as needed) and results are provided, as expected from a benchmarking paper.

**Strengths:**

Major strengths :

1. The paper presents the first comprehensive benchmark of 29 defense methods,14 attack methods, and 9 IQA methods for NR-IQA
2. Multi-dimensional approach to evaluation: robustness, preserving correlation with human perception (SRCC/PLCC), and quality of the image with respect to original (PSNR, SSIM)
3. Practical Contribution to Research and Industry -> Good work on setting up a public benchmark.

**Weaknesses:**

Major weaknesses :

1. The paper is a valuable resource for the IQA community. But in terms of technical novelty, it would have been nice to have an attack/defense method with some motivation for the IQA task. I feel this paper would have been more suitable for a Benchmarking Track, but I understand ICLR lacks such a track and had to submit it to the main conference track
2. The appendix provides many results, but it is very difficult to connect them to the main text, which points to the paper's poor organization.
3. The authors should add LPIPS to the results in Table 3 (and other similar tables) along with PSNR and SSIM. PSNR is not a perceptual metric and can be reliable, leaving SSIM as the only metric. It is better to report both SSIM and LPIPS scores.

Minor :

1. Paper formatting needs to improve, and content organization can also be better. For example, Fig 1 does not discuss page 8.
2. Section 3.1 under Adversarial defenses: It is better to divide the paragraphs into different types. This will make reading the paper much easier.

**Questions:**

1. In section 3.2, "clustering the KonIQA dataset by spatiotemporal complexity." - Could you please explain the temporal aspect of images?
2. Given the high computational demands of certified defenses and 10 images being used? How would you expect the results to vary as you sample different sets of 10 images?
3. Logistics questions on the leaderboard :
   How do you plan to maintain the leaderboard, and will there be mechanisms for incorporating new defense/attack techniques over time?

---

> ### Author Response · Authors · 2024-11-18
>
> Thank you for your valuable suggestions and thoughtful feedback. We address your concerns below:
> ### Weaknesses
> 1. Our primary goal was to create a benchmark that highlights the need for IQA-specific adversarial methods. We agree that methods tailored specifically for IQA would be valuable. While our current focus is on establishing this benchmark, we plan to pursue developing IQA-motivated attacks and defenses in future work. We also want to highlight that we did evaluate 4 IQA-specific attacks, 1 defense (FCN filter) and 1 defense in progress (Gradient Norm Regularization)
> 2. We refer to the figures and tables located in the Appendix from the main text. for example: (lines 140, 205, 275, etc.).  We agree that the Appendix needs to be easier to follow, so we added a brief summary of the appendix structure in section A0 (lines 760-780 in the revised paper), highlighting how each section connects to specific parts of the main text. We hope this makes it easier to navigate the results and align them with our main findings.
> 3. Thank you for your suggestion. Our motivation for using PSNR and SSIM was based on the simple structure of these metrics. LPIPS itself is a NN-based metric, and is known to be vulnerable to adversarial attacks ([1],[2]). We assume that LPIPS can be used as a perceptual metric in our study, however,  it requires a robustness analysis against transferable attacks on NR IQA models. This step is essential to ensure its reliability and consistency in the context of adversarial evaluation. We also have included additional metrics such as L_inf and L_2 alongside PSNR and SSIM to better capture image quality, particularly in terms of structural and perceptual features.
> 4. Thank you for the suggestion. In the revised version, we have reorganized Section 3.1 by dividing the paragraphs according to defense types to enhance readability.
> ### Questions
> 1. We apologize for typo. The correct term is "spatial complexity," and we have made the necessary corrections in the updated version of the paper (line 204). Section A3 contains a correct description.
> 2. Thank you for raising this point. To address this, we will conduct additional experiments to evaluate the stability of our results across multiple, randomly sampled sets of 10 images. This analysis will help to assess the consistency of our findings and determine if the results remain robust across different image selections. We will report these additional results in the updated version of the paper in a few days.
> 3. Our benchmark and leaderboard will be published online, where we will openly accept submissions of new defense methods. As mentioned in Section 3.6, we have implemented an automated pipeline to compute results for all new submissions efficiently. To ensure consistency, we require submissions to adhere to a specified PyTorch interface. Regarding submission for new attack methods, they can also be submitted and we will design a separate leaderboard with a comparison of these attacks in the future.
>
> [1] https://arxiv.org/pdf/1906.03973
>
> [2] https://arxiv.org/pdf/2307.15157

---

> > ### Comment · Reviewer_RZ8e · 2024-11-25
> >
> > Thanks for the reply. It would be good to see results related to Q#2.

---

> > > ### Author Response · Authors · 2024-11-27
> > >
> > > We have conducted additional experiments to evaluate the impact of sampling different sets of 10 images on certified defenses. We have sampled 10 different sets of images from KONIQ-10k following the sampling strategy described in the paper, evaluated certified defenses for the PAQ2PIQ IQA model and calculated mean and 95% confidence intervals for $D_{score}$, $R_{score}$ and Cert.R./Cert.RD. These results are summarized in the Table below:
> > > | Certified Defense| Robustness ($R_{score}\uparrow$) |  $D^{{(D)}}_{{score}}$$\downarrow$ | $Cert. R \uparrow$ / $Cert. RD \downarrow$
> > > | :-| :- | --------: | -: |
> > > RS   | $5.54 \pm 0.23$          | $5.44  \pm 1.57$  | $0.179 \pm 0.01 / \infty$
> > > DRS  | $1.67 \pm 0.13$           | $14.9  \pm 4.20$  | $0.172 \pm 0.01 / \infty$
> > > DDRS | $5.71 \pm 0.25$           | $2.63  \pm 0.59$  | $0.165 \pm 0.01 / \infty$
> > > DP   | $5.60 \pm 0.26$           | $2.49  \pm 0.54$  | $0.161 \pm 0.01 / \infty$
> > > MS   | $1.33 \pm 0.17$           | $6.97  \pm 1.38$  | $0 / 2.83 \pm 0.36$
> > > DMS  | $1.41 \pm 0.13$           | $7.49  \pm 1.41$  | $0 / 2.28 \pm 0.33$
> > >
> > > The results align closely with those presented in the main paper. For example, RS and DRS perform best in Cert.R, while RS, DDRS and DP show the best $R_{score}$. The consistency across samples is further supported by narrow confidence intervals, which suggest minimal variability between different sets of sampled images. Given the current results, we consider the findings in the main paper representative across different image subsets.

---

### Official Review · Reviewer_RxLB · 2024-11-03

**Soundness:** 3
**Presentation:** 3
**Contribution:** 2
**Rating:** 5
**Confidence:** 4

**Summary:**

The paper aims to benchmark and evaluate the robustness of 30 different adversarial defense methods against 14 adversarial attacks regarding IQA metrics. It emphasizes the need for effective defenses due to the unique challenges posed by preserving image quality while defending against adversarial perturbations. It presents a comprehensive analysis of the efficiency of various adversarial defense methods for image quality assessment (IQA) metric.

**Strengths:**

(1) This paper gives a comprehensive comparison of multiple defense methods against IQA under a variety of attacks, and draws a few conclusions under different scenarios.
(2) The detailed analysis of the trade-offs between performance and robustness in various defense strategies offers practical guidance for researchers and developers.
(3) The inclusion of statistical tests and evaluations of quality scores adds robustness to the findings.

**Weaknesses:**

(1)	Although the paper considers 30 different defense methods, it ignores some defense methods which are tailored for IQA methods specifically such as [1]. These methods should be discussed and compared in the paper, as the goal of this paper is to discuss the defense of IQA.
(2)	The paper has evaluated different defense methods under different indicators, showing a lot of charts, but it lacks the in-depth analysis about what is the reason behind the effectiveness of different defense methods which is important. For example, the defense performance on the KonIQ-1k dataset on the right of Figure 3 exceeds the other two datasets in multiple defense methods. What is the reason? Why do many attack methods in Figure 4 achieve the worst defense performance on FPR, in terms of R robustness?
(3)  More experimental details and analysis are expected. For example, in line 227, 50 images are selected from 1k images for attack, do different selections of attack images affect the performance of attack and defense? Does it affect the conclusion?

[1] Defense Against Adversarial Attacks on No-Reference Image Quality Models with Gradient Norm Regularization, IEEE Conference on Computer Vision and Pattern Recognition (CVPR), Seattle, WA, USA, June 2024

**Questions:**

Please see Weakness.

---

> ### Author Response · Authors · 2024-11-18
>
> Thank you for your thoughtful and detailed feedback. We appreciate the recognition of our study and analysis. We address some of the weaknesses below:
>
> (1) and (3):
> Among IQA-specific defenses, we tested the FCN filter [1], specifically tailored for IQA methods. Additionally, we are conducting experiments regarding the proposed defense (Gradient Norm Regularization[2]) and sampling strategy. We will update the paper with these results in a few days.
> Regarding (2):
> 1. Figure 3 shows SROCCadv results for each defense method, which highly depends on SROCCclear. Table 24 in the Appendix shows SROCCclear for each dataset. KonIQ-1k consistently demonstrates higher SROCCclear values compared to AGIQA3k and KADID datasets. This can be due to two factors: a) Several IQA models (e.g., TOPIQ and CLIP-IQA+) were trained on the KonIQ-10k dataset or its subsets, giving them a natural advantage on KonIQ-1k. b) Certain IQA models, such as MetaIQA and PAQ2PIQ, generally achieve higher correlation values on KonIQ-10k, as reported in their respective studies, suggesting an inherent dataset bias.
> 2. Figure 4 (a) shows the R score, with higher scores indicating more robust IQA metrics. FPR's low R score indicates that FPR is the most vulnerable model among evaluated models. These results correlate with the previous research (https://videoprocessing.ai/benchmarks/metrics-robustness.html). This vulnerability is likely caused by its atypical architecture for the NR-IQA task, which includes a Siamese network and an attempt to ”hallucinate” the features of the pseudo-reference image from a distorted one.
> 3. These insights are integrated into the revised version.
>
> [1] Gushchin et al., “Adversarial purification for no-reference image-quality metrics: applicability study and new methods,” 2024
> [2] Defense Against Adversarial Attacks on No-Reference Image Quality Models with Gradient Norm Regularization, IEEE Conference on Computer Vision and Pattern Recognition (CVPR), Seattle, WA, USA, June 2024

---

> > ### Comment · Reviewer_RxLB · 2024-11-26
> >
> > Thanks for the authors' reply and their efforts for conducting new experiments.  The reply to Weakness 1 and 3 is fine and addresses my concerns. But the reply to Weakness 2 only partially addresses the concern. More in-depth analysis about the effectiveness of different defense methods are expected.

---

> > > ### Author Response · Authors · 2024-12-01
> > >
> > > To further address your concern, we have added Section A.7 in the Appendix, where we provide additional analysis and insights into the effectiveness of different defense methods. Below, we summarize the key findings:
> > >
> > > * Role of Geometric Transformations and Distribution Awareness:
> > >
> > > Defenses that incorporate geometric transformations or account for the differences between clean and attacked image distributions perform significantly better than those that do not. For instance, transformations like Random Rotate introduce variability that helps mitigate adaptive attacks, while static transformations are less effective due to their predictability.
> > >
> > > * Effectiveness of Compression Techniques:
> > >
> > > Compression-based methods are highly effective as they remove high-frequency noise while preserving the underlying image structure. This explains their strong performance against attacks that exploit fine-grained perturbations.
> > >
> > > * Performance of Denoising Methods:
> > >
> > > Methods like MPRNet and Real-ESRGAN demonstrate moderate effectiveness. However, their limitations stem from being trained on simpler noise types, highlighting the need for fine-tuning on adversarial perturbations to improve robustness.
> > >
> > > * Diffusion-Based Models:
> > >
> > > While diffusion-based models offer tunable strengths and have shown success in classification tasks, they face challenges in quality assessment tasks due to the introduction of their own artifacts, which are degrade perceived quality and worsen correlations.
> > >
> > > * Adaptive Attack Mitigation:
> > >
> > > High-randomness approaches combined with geometric transformations, such as Random Rotate, are particularly effective at mitigating adaptive attacks. In contrast, methods relying on static transformations are less adaptive and thus less robust.
> > >
> > > These observations provide a foundation for future research into improving defense strategies. Section A.7 expands on these findings and includes potential directions for further investigation, such as fine-tuning denoising models and combining multiple defenses to leverage their complementary strengths. We hope this additional analysis addresses your concerns and welcome further feedback.

---

### Author Response · Authors · 2024-11-24
**Overall Response**

Dear Reviewers,
We sincerely thank you all for your thorough and constructive feedback. We appreciate the time spent reviewing our work.
We are grateful for the positive reception of our work. Multiple reviewers highlighted our study’s practical contribution to research and industry and its detailed and comprehensive analysis.

To address the main concerns raised in the reviews, we have made several improvements.
* Following the _RxLB_'s suggestion, we have evaluated defense method [1] and included its results in Figure 1 and Table 4.
* We have conducted additional experiments to evaluate the impact of sampling 50 images from the dataset. The results of these experiments are included in Section A.2 in the Appendix.
* Following the _sRtC_'s suggestion, we have added $MSE$ and $L_{inf}$ metrics to Table 3 to better measure effects on image quality.
* To enhance the readability of the Appendix, we have added section A.0 with a brief Appendix structure.
* We also have corrected some mistakes and typos pointed out by reviewers.
Thank you again for your valuable feedback!

[1] Defense Against Adversarial Attacks on No-Reference Image Quality Models with Gradient Norm Regularization, IEEE Conference on Computer Vision and Pattern Recognition (CVPR), Seattle, WA, USA, June 2024

---

### Meta-Review · Area_Chair_yTzj · 2024-12-16

**Metareview:**

The authors present a comprehensive study benchmarking the robustness of 29 defense methods for 14 adversarial attacks against 9 different image quality assessment (IQA) metrics. While the study is comprehensive and of value to the IQA community, there is too little new insight generated to be a good fit for ICLR.

It is recommended to consider sending the work to an image processing journal or to a benchmarking track of a major conference.

**Additional Comments On Reviewer Discussion:**

During the review phase, authors addressed concerns around legibility and organization of the paper, and added additional missing experiments. However, all reviewers cited the lack of new insights as one of the main weaknesses of the paper, which was not addressed satisfactorily by the authors.

---

### Decision · Program_Chairs · 2025-01-22

Reject